# Selective Omniprediction and Fair Abstention

**Sílvia Casacuberta**
Department of Computer Science
Stanford University
scasac@stanford.edu

**Varun Kanade**
Department of Computer Science
University of Oxford
varun.kanade@cs.ox.ac.uk

## Abstract

We propose new learning algorithms for building selective classifiers, which are predictors that are allowed to abstain on some fraction of the domain. We study the model where a classifier may abstain from predicting at a fixed cost. Building on the recent framework on multigroup fairness and omniprediction, given a pre-specified class of loss functions, we provide an algorithm for building a single classifier that learns abstentions and predictions optimally for every loss in the entire class, where the abstentions are decided efficiently for each specific loss function by applying a fixed post-processing function. Our algorithm and theoretical guarantees generalize the previously-known algorithms for learning selective classifiers in formal learning-theoretic models.

We then extend the traditional multigroup fairness algorithms to the selective classification setting and show that we can use a calibrated and multiaccurate predictor to efficiently build selective classifiers that abstain optimally not only globally but also locally within each of the groups in any pre-specified collection of possibly intersecting subgroups of the domain, and are also accurate when they do not abstain. We show how our abstention algorithms can be used as conformal prediction methods in the binary classification setting to achieve both marginal and group-conditional coverage guarantees for an intersecting collection of groups. We provide empirical evaluations for all of our theoretical results, demonstrating the practicality of our learning algorithms for abstaining optimally and fairly.

## 1 Introduction

Selective classification has long been proposed as a way of achieving higher accuracy at the cost of abstaining from making predictions on certain points[12, 19, 43, 41, 42, 23]. Such behavior is often desirable, e.g. when stakes of mistakes (of one type or another) can be very costly, such as medical diagnoses. There has been a wide range of theoretical and empirical work in this area. The most relevant one to our work is the learning-theoretic *reliable learning framework* [43] of Kalai, Kanade, and Mansour, where they introduce a framework that gives guarantees on error and abstention rate in an agnostic learning framework. Their work focuses on binary classification problems using only the *zero-one* loss function. A more general learning framework for selective classification is to consider the Chow loss [12] and its generalizations, where one can consider loss functions for predictions as well as a fixed penalty for abstention, and a weighted combination of these should be minimized.

Separately, recent work has studied the question of whether training from scratch is required for every loss function. Gopalan et al. introduce the notion of a $(\mathcal{L}, \mathcal{C})$-*omnipredictor* for a class of loss functions $\mathcal{L}$ and a concept class $\mathcal{C}$, which is a single classifier that can be efficiently post-processed to compete with a class of models for the large family $\mathcal{L}$ [25]. Omnipredictors turn out to be efficiently constructable from a multigroup fairness notion called *multicalibration* [32, 50, 16]. Here predictors are required to be calibrated, even conditioned on group membership from a large, possibly-intersecting, collection of groups. While (multi)calibration is a desirable property, these notions

39th Conference on Neural Information Processing Systems (NeurIPS 2025).

do not imply anything about accuracy, and rely upon the base predictor (which is then modified to satisfy multicalibration) being accurate to begin with. As in the case of selective classification more generally, it may be desirable to abstain in order to guarantee high accuracy when predicting.

In this work, we apply the rich line of work on omniprediction and multigroup fairness to the problem of selective classification. First, we introduce the notion of a *selective classifier*, which is able to simultaneously minimize generalized Chow losses from a rich family of loss functions. We show how this generalizes the previous results on reliable agnostic learning beyond the 0-1 loss. Second, we introduce the notion of *multigroup selective classification*, where the predictions need to be highly accurate, but in addition it is required that the rate of abstention conditioned on any group is no worse than the abstention rate of an optimal classifier for that class from a base class of abstaining classifiers. We show that such a predictor can be obtained starting from a slightly weaker notion of *calibrated multiaccuracy* (rather than multicalibration) and access to a weak agnostic learner for a suitable class. When the number of groups is small, we show that this can be constructed efficiently with only access to a reliable learner for the base class (which is believed to be a weaker notion than weak agnostic learning [45]). The main results and the focus of our work is theoretical. We do however also provide an empirical evaluation on synthetic data that shows that our algorithms are easy to implement and achieve the desired outcomes. All proofs of our claims are deferred to the appendix.

## 1.1 Related work

We summarize the most relevant work to ours here; further related work is discussed in the appendix.

**Selective classification.** The idea of building an algorithm that abstains on certain predictions is not new and has traditionally been called *selective classification* [19]. A selective classifier is allowed to also return ? as an output (which indicates abstention), along with the usual numerical values. The works [43, 45] study this in the context of agnostic learning and give bounds on both the error and abstention rate (competitive with respect to a base class). Selective classification has also been studied within the context of a Chow loss framework, where there is a fixed cost of abstention in addition to a loss function on predictors and the goal is to minimize the overall cost [12]. Abstention has also been used to provide bounds on efficient learning in the presence of covariate shifts [41, 42, 23]. Although somewhat different, a related notion is that of conformal prediction [66, 3, 39], where the classifier is allowed to output a set and the goal is that the true label should be in the predicted set with high probability. In binary classification case, considering abstention as predicting the set $\{0, 1\}$ relates these two notions tightly.

**Fairness.** Recently, in the setting of fairness, some works have studied the effect of selectively abstaining. Jones et al. find that certain forms of selective classification can magnify disparities across groups [37], and follow-up works try to restrict this type of disparities [52, 62, 67]. Some recent works have put forth the necessity of accounting for uncertainty in the setting of algorithmic fairness [2, 5, 40, 54, 49, 53, 13], but this question remains largely understudied. This question is also related to the problem of *model multiplicity*, which has recently drawn a lot of attention [7, 8, 13]. Outside of the selective classification problem, the multigroup fairness framework has recently developed various techniques for ensuring that desired properties of the predictor (e.g., accuracy, calibration) hold even when conditioned on any of the possibly intersecting groups in a rich collection [32, 29, 48, 51, 61].

## 2 Notation & Preliminaries

We let $\mathcal{X}$ denote the domain, $\mathcal{Y} = \{0, 1\}$ the set of labels, $\mathcal{D}$ a distribution over $\mathcal{X} \times \mathcal{Y}$, and $\mathcal{C}$ a concept class over $\mathcal{X}$ of concepts $c$ (their range is specified in each application). We assume that $\mathcal{C}$ is closed under complement and contains the constant functions $\mathbf{0}, \mathbf{1}$. We denote the marginal distribution over $\mathcal{X}$ by $\mathcal{D}_{\mathcal{X}}$. A loss function $\ell$ takes a label $y \in \mathcal{Y}$ and an action $t \in \mathbb{R}$ and returns a loss value $\ell(y, t)$. Examples include the $\ell_p$ losses $\ell_p(y, t) = |y - t|^p$, the logistic loss $\ell(y, t) = \log(1 + \exp(-yt))$, and binary classification with different false-positive/negative costs $\ell(y, t) = c_y|y - t|$. We let $\mathcal{L} = \{\ell : \mathcal{Y} \times \mathbb{R} \to \mathbb{R}\}$ denote a collection of loss functions. We are interested in minimizing loss functions: we want to find a hypothesis $h : \mathcal{X} \to \mathbb{R}$ that makes the expected loss $\ell_{\mathcal{D}}(h) := \mathbb{E}_{\mathcal{D}}[\ell(y, h(x))]$ small. We work in the agnostic setting, where we want the expected loss of our hypothesis $h$ to be at most epsilon higher than the best concept in $\mathcal{C}$ for that loss function. Importantly, the optimal concept in $\mathcal{C}$ depends on the choice of $\ell$. We denote the ground truth predictor by $f^*(\mathbf{x}) = \mathbb{E}_{\mathcal{D}}[\mathbf{y}|\mathbf{x}]$.

## 2.1 Multigroup fairness notions

The *multigroup* framework was introduced as a way to bridge individual and group fairness notions [32, 46]. Given a collection of subgroups $\mathcal{G} = \{g : \mathcal{X} \to \{0, 1\}\}$ that can intersect arbitrarily, we want to ensure a property of interest (accuracy, calibration) within *each* of the subgroups in $\mathcal{G}$.

**Definition 2.1** (Multiaccuracy, calibrated multiaccuracy, and multicalibration [32, 26])**.** We say that a predictor $h$ is $(\mathcal{G}, \epsilon)$-*multiaccurate* for a distribution $\mathcal{D}$ if

$$\max_{g \in \mathcal{G}} \left| \mathbb{E}_{\mathcal{D}}[g(\mathbf{x})(\mathbf{y} - h(\mathbf{x}))] \right| \leq \epsilon.$$

We say the predictor $h$ is $\epsilon$-*calibrated* for $\mathcal{D}$ if $\mathbb{E}[|\mathbb{E}[\mathbf{y}|h(\mathbf{x})] - h(\mathbf{x})|] \leq \epsilon$. We say that $h$ is $(\mathcal{G}, \epsilon)$-*multiaccurate and calibrated* if it is both $(\mathcal{G}, \epsilon)$-multiaccurate and $\epsilon$-calibrated. Further, we say that $h$ is $(\mathcal{G}, \epsilon)$-*multicalibrated* if

$$\max_{g \in \mathcal{G}} \mathbb{E}[|g(\mathbf{x})(\mathbb{E}[\mathbf{y}|p(\mathbf{x})] - p(\mathbf{x}))|] \leq \epsilon.$$

We can efficiently construct predictors satisfying these increasingly demanding notions if we have access to the learning primitive of a weak agnostic learner for the class $\mathcal{G}$: we require $O(1/\epsilon^2)$ calls to the weak agnostic learner in the case of $(\mathcal{G}, \epsilon)$-multiaccuracy [32] and of $(\mathcal{G}, \epsilon)$-calibrated multiaccuracy [26], and $O(1/\epsilon^6)$ calls in the case of $(\mathcal{G}, \epsilon)$-multicalibration [32, 25]. The line of work on multigroup fairness has proven to be extremely rich in recent years [24, 27, 10, 16, 18, 17].

## 2.2 Omnipredictors

One of the most successful applications of the multigroup fairness framework has arguably been in learning theory, where Gopalan et al. used it to propose a new indistinguishability-based learning framework [25]. This framework has been applied in many follow up works [35, 28, 27, 59, 24]. Specifically, the usual learning paradigm first chooses a loss of interest (e.g., $\ell_1$ or $\ell_2$), and then trains a model to minimize it. But what if we do not know the specific loss at the time of training, or if we want to change it at a later time without having the re-train from scratch? We could instead hope to construct the following object, which they called a $(\mathcal{C}, \mathcal{L})$-*omnipredictor*:

**Definition 2.2** (Omniprediction [25])**.** Given a class of loss functions $\mathcal{L}$ and a concept class $\mathcal{C}$ of concepts $c : \mathcal{X} \to \mathbb{R}$, a predictor $h : \mathcal{X} \to [0, 1]$ is an $(\mathcal{L}, \mathcal{C}, \epsilon)$-*omnipredictor* if for every $\ell \in \mathcal{L}$ there exists a function $k_\ell : [0, 1] \to \mathbb{R}$ so that

$$\ell_{\mathcal{D}}(k_\ell \circ h) \leq \min_{c \in \mathcal{C}} \ell_{\mathcal{D}}(c) + \epsilon.$$

That is, for every loss $\ell \in \mathcal{L}$, there exists a simple (univariate) transformation $k_\ell$ of the predictions of $h$ (chosen tailored to $\ell$) such that $k_\ell \circ f$ has loss comparable to the best hypothesis $c \in \mathcal{C}$, which is chosen dependent on $\ell$. That is, we can train a *single* predictor $h$ that is able to do as well as the best hypothesis in $\mathcal{C}$ separately for every loss function in $\mathcal{L}$. This realizes a very strong learning guarantee. Note that for every $\mathcal{C}, \mathcal{L}$, the ground truth predictor $f^*$ is an $(\mathcal{L}, \mathcal{C}, 0)$-omnipredictor. As shown in [25], the right post-processing function $k_\ell$ turns out to be the minimizer of the expected loss under the Bernoulli distribution, a fact we use to show the optimality of our selective predictor. In their main result, Gopalan et al. show that we can construct omnipredictors efficiently using the technique of multicalibration:

**Theorem 2.1** (Building omnipredictors from multicalibration [25])**.** *Let $\mathcal{D}$ be a distribution on $\mathcal{X} \times \{0, 1\}$, $\mathcal{C}$ a family of real-valued functions on $\mathcal{X}$, and $\mathcal{L}$ the family of all $B$-Lipschitz, convex loss functions. Then, a $(\mathcal{C}, \epsilon)$-multicalibrated predictor $h$ is an $(\mathcal{L}, \mathcal{C}, 2\epsilon B)$-omnipredictor.*

This result can be extended beyond convex Lipschitz loss functions, including to the exponential loss, GLM losses, 1-Lipschitz losses, proper losses, and bounded variation losses [25, 59].

## 2.3 Reliable agnostic learning

In our setting, we allow predictors to output an abstention ?, and so we consider triplets of loss functions $(\ell_+, \ell_-, \ell_?)$. If a predictor is allowed to abstain and thus ? is in its support, we denote it with an abstention sign in the subscript.

**Losses** $(\ell_+, \ell_-, \ell_?)$. We further specify loss functions depending on the value of $y \in \{0, 1\}$. Given a loss function $\ell : \mathcal{Y} \times \{\mathbb{R} \cup \{?\}\} \to \mathbb{R}$, we decompose $\ell = (\ell_+, \ell_-, \ell_?)$ as follows:

1. **Negative labels.** For inputs $(y, t)$ where $t \neq ?$ and $y = 0$, we write $\ell_+(t)$ for $\ell_{\mathcal{D}}(0, t)$.

2. **Positive labels.** For inputs $(y, t)$ where $t \neq ?$ and $y = 1$, we write $\ell_-(t)$ for $\ell_{\mathcal{D}}(1, t)$.

3. **Abstentions.** For inputs $(y, t)$ where $t = ?$, we write $\ell_?(y)$ for $\ell(y, t)$. In turn to separate the cases $y = 1$ and $y = 0$, we write $\ell_?(1) = \alpha_+$ and $\ell_?(0) = \alpha_-$. Whenever the predictor cannot be uniquely inferred from the context, we still write $\ell_?(y, t)$.

We drop $\mathcal{D}$ if it can be directly inferred. For example, in the specific case of the 0-1 loss, $\ell_+(0, t) = |0 - t|$ and $\ell_-(1, t) = |1 - t|$, and so the expected loss $\mathbb{E}[\ell_+(\mathbf{y}, h(\mathbf{x}))]$ is equal to the rate of false positives and $\mathbb{E}[\ell_-(\mathbf{y}, h(\mathbf{x}))]$ to the rate of false negatives. The sum of losses $\ell_+ + \ell_-$ corresponds to the usual definition of *error* of the predictor. In turn, $\ell_?$ generalizes the definition of the abstention rate of a predictor $\mathbb{E}_{\mathcal{D}}[\mathbb{1}[h(\mathbf{x}) = ?]]$. The case where $\ell_?(y) = \alpha$ for any constant $\alpha > 0$ corresponds to the traditional Chow model [12, 41]. Note that we allow different abstention costs depending on whether the corresponding label is $y = 0$ or $y = 1$.

**Definition 2.3** (Triplet of loss functions). Given a family of loss functions $\mathcal{L}$, each loss function $\ell \in \mathcal{L}, \ell : \mathcal{Y} \times \{\mathbb{R} \cup \{?\}\} \to \mathbb{R}$ induces the triplet of loss functions $(\ell_+, \ell_-, \ell_?)$, where $\ell_+(t) = \ell_{\mathcal{D}}(0, t)$ and $\ell_-(t) = \ell_{\mathcal{D}}(1, t)$ for all $t \neq ?$, and $\ell_?(\mathbf{y}) = \ell_{\mathcal{D}}(\mathbf{y}, t)$ for all $t = ?$.

We directly use the notation $(\ell_+, \ell_-, \ell_?) \in \mathcal{L}$. When we consider selective omnipredictors, we will further associate weights $(\lambda, \mu, \nu)$ with the triplet of loss functions $(\ell_+, \ell_-, \ell_?)$, so that the total loss incurred by a predictor is equal to $\lambda \ell_+ + \mu \ell_- + \nu \ell_?$.

Allowing abstentions increases the reliability of non '?' predictions. The work most closely related to ours is the learning-theoretic framework of *reliable agnostic learning*, first proposed by Kalai, Kanade, and Mansour [43], which adapts the usual agnostic framework. While they introduced their definitions only for the case of the 0-1 loss, in our results we generalize their results to many more loss functions, and so we directly introduce the more general versions of their original definitions. Let $\mathsf{EX}(\mathcal{D})$ denote the example oracle which when queried returns $(\mathbf{x}, \mathbf{y}) \sim \mathcal{D}$. Given a concept class $\mathcal{C}$ of Boolean concepts $c : \mathcal{X} \to \{0, 1\}$ and a distribution $\mathcal{D}$, we further define the following concept classes from $\mathcal{C}$ [43]:

$$\mathcal{C}^+ = \{c \in \mathcal{C} \mid \ell_+(c, \mathcal{D}) = 0\}, \qquad \mathcal{C}^- = \{c \in \mathcal{C} \mid \ell_-(c, \mathcal{D}) = 0\}.$$

**Definition 2.4** (PRL for a family of loss functions [43, 45]). A concept class $\mathcal{C}$ of Boolean concepts is $\mathcal{L}$-*positively reliably learnable* if there exists a learning algorithm that for any distribution $\mathcal{D}$ over $\mathcal{X} \times \{0, 1\}$, any $\ell_+, \ell_- \in \mathcal{L}$, and any $\epsilon, \delta > 0$, when given access to the example oracle $\mathsf{EX}(\mathcal{D})$, outputs a hypothesis $h : \mathcal{X} \to [0, 1]$ that satisfies the following with probability at least $1 - \delta$:

1. $\mathbb{E}_{(\mathbf{x}, \mathbf{y}) \sim \mathcal{D}}[\ell_+(h(\mathbf{x}))] \leq \epsilon$,

2. $\mathbb{E}_{(\mathbf{x}, \mathbf{y}) \sim \mathcal{D}}[\ell_-(h(\mathbf{x}))] \leq \min_{c_+ \in \mathcal{C}^+} \mathbb{E}_{(\mathbf{x}, \mathbf{y}) \sim \mathcal{D}}[\ell_-(c_+(\mathbf{x}))] + \epsilon$.

The notion of $\mathcal{L}$-*negative reliable learning* (NRL) is defined analogously, by switching the positives and the negatives (see the appendix for the full definitions).

Hence, in the case of the 0-1 loss, a positive reliable classifier is one that almost never produces false positives, while simultaneously minimizing false negative errors, attaining a rate comparable to the false negative error rate of the best classifier $c_+ \in \mathcal{C}^+$ [45]. Symmetrically, a negative reliable classifier is one that almost never produces false negatives while simultaneously minimizing false positive errors, attaining a rate comparable to the false positive error rate of the best $c_- \in \mathcal{C}^-$.

Given any Boolean $c_+ \in \mathcal{C}^+$ and $c_- \in \mathcal{C}^-$, we ensemble them to construct a selective classifier $c_? = (c_+, c_-)$ as follows:

$$c_?(x) = \begin{cases} 1 & \text{if } c_+(x) = c_-(x) = 1, \\ 0 & \text{if } c_+(x) = c_-(x) = 0, \\ ? & \text{if } c_+(x) \neq c_-(x). \end{cases}$$

We then let $\mathsf{SC}(\mathcal{C}) = \{(c_+, c_-) \mid c_+ \in \mathcal{C}^+, c- \in \mathcal{C}^-\}$ (for "selective clasifiers"). This ensembling provides a natural way for adding abstentions to the base class $\mathcal{C}$. We can now define *fully reliable learning* using the base class $\mathsf{SC}(\mathcal{C})$:

**Definition 2.5** (FRL for a family of loss functions). A concept class $\mathcal{C}$ of Boolean concepts is $\mathcal{L}$-*fully reliably learnable* if there exists a learning algorithm that for any distribution $\mathcal{D}$ over $\mathcal{X} \times \{0, 1\}$, any $\ell_+, \ell_-, \ell_? \in \mathcal{L}$, and any $\epsilon, \delta > 0$, when given access to the example oracle $\mathsf{EX}(\mathcal{D})$, outputs a hypothesis $h_? : \mathcal{X} \to [0, 1] \cup \{?\}$ that satisfies:

1. $\mathbb{E}_{(\mathbf{x}, \mathbf{y}) \sim \mathcal{D}}[\ell_+(h_?(\mathbf{x})) + \ell_-(h_?(\mathbf{x}))] \leq \epsilon$,

2. $\mathbb{E}_{(\mathbf{x}, \mathbf{y}) \sim \mathcal{D}}[\ell_?(\mathbf{y}, h_?(\mathbf{x}))] \leq \min_{c_? \in \mathsf{SC}(\mathcal{C})} \mathbb{E}_{(\mathbf{x}, \mathbf{y}) \sim \mathcal{D}}[\ell_?(\mathbf{y}, c_?(\mathbf{x}))] + \epsilon$.

For the specific case where $\mathcal{L}$ corresponds only to the 0-1 loss, Kalai, Kanade, and Mansour showed that if $\mathcal{C}$ is efficiently agnostically learnable, then $\mathcal{C}$ is also efficiently $\mathcal{L}$-reliable learnable, all for PRL, NRL, and FRL [43].

## 2.4 Generalized Chow model

In the original Chow abstention model, the predictor can choose to abstain at a fixed cost of $\alpha \geq 0$ [12, 41]. The goal of the predictor is to learn how to choose real-valued predictions and abstentions as to minimize the total loss. We generalize and formalize this model with what we call the *generalized Chow loss function*:

**Definition 2.6** (Generalized Chow loss). Given a triplet of loss functions $(\ell_+, \ell_-, \ell_?)$ induced by $\ell : \mathcal{Y} \times \{\mathbb{R} \cup \{?\}\} \to \mathbb{R}$ with associated weights $(\lambda, \mu, \nu)$, where $\lambda + \mu + \nu \leq 1$, and a selective classifier $h_? : \mathcal{X} \to \mathbb{R} \cup \{?\}$, the *generalized Chow loss* incurred by $h$ is equal to

$$\ell_{\mathrm{GC}, \mathcal{D}}(h_?; \lambda, \mu, \nu) = \mathop{\mathbb{E}}_{(\mathbf{x}, \mathbf{y}) \sim \mathcal{D}}[\lambda \ell_+(h(\mathbf{x})) + \mu \ell_-(h(\mathbf{x})) + \nu \ell_?(\mathbf{y})].$$

One can include randomized predictors $h$ to this definition (where $h$ assigns a *probability* of abstaining to each $x \in \mathcal{X}$); however, as we show in the appendix (Section C.1), randomization does not help in minimizing the generalized Chow loss function.

## 2.5 Selective omniprediction

In our work, we extend the omniprediction framework to the setting of selective classification.

In the next section, we show that we can efficiently build a $(\mathcal{L}, \mathcal{C}, \epsilon)$-multigroup selective omnipredictor that optimally minimizes the generalized Chow loss of *any* triplet of loss functions $(\ell_+, \ell_-, \ell_?)$ induced by any $\ell \in \mathcal{L}$. We then use this result to show how to efficiently build $\mathcal{L}$-FRL classifiers, generalizing the main result of [43].

**Definition 2.7** (Selective omniprediction). Given a concept class $\mathcal{C}$ on $\mathcal{X}$, distribution $\mathcal{D}$, $\epsilon > 0$, and a class of loss functions $\mathcal{L}$, we say that a predictor $h : \mathcal{X} \to [0, 1]$ is a $(\mathcal{L}, \mathcal{C}, \epsilon)$-*selective omnipredictor* if for every $(\ell_+, \ell_-, \ell_?) \in \mathcal{L}$ and any associated weights $(\lambda, \mu, \nu)$, there exists a function $k^*_{\ell_\pm, \ell_?} : [0, 1] \to \mathbb{R} \cup \{?\}$ such that for any post-processing function $k : [0, 1] \to \mathbb{R} \cup \{?\}$,

$$\ell_{\mathrm{GC}, \mathcal{D}}(k^*_{\ell_\pm, \ell_?} \circ h; \lambda, \mu, \nu) \leq \min_{c_? \in k \circ \mathcal{C}} \ell_{\mathrm{GC}, \mathcal{D}}(c_?; \lambda, \mu, \nu) + \epsilon.$$

Importantly, as in the original omniprediction framework, the optimal classifier $c_?$ in $k \circ \mathcal{C}$ is tailored to the specific triplet of loss functions, whereas $h$ is a *single* classifier for all of $\mathcal{L}$ and $(\lambda, \mu, \nu)$.

# 3 Building Selective Omnipredictors

The key idea in the omniprediction learning framework is to first learn a model $f$ that is $(\mathcal{C}, \epsilon)$-computationally indistinguishable from $f^*$, which is accomplished through the technique of multicalibration, and then apply a post-processing function $k^*_\ell$ once a loss function $\ell \in \mathcal{L}$ has been fixed. In our setting of selective classification, we similarly use the following post-processing function that minimizes expected loss under the Bernoulli distribution, which we show yields an optimal final loss:

**Definition 3.1.** Given loss functions $(\ell_+, \ell_-, \ell_?)$ and corresponding weights $(\lambda, \mu, \nu)$, let the function $k^*_{\ell_\pm, \ell_?} : [0, 1] \to \mathbb{R} \cup \{?\}$ be defined as

$$k^*_{\ell_\pm, \ell_?}(p) = \arg\min_{t \in \mathbb{R} \cup \{?\}} \mathop{\mathbb{E}}_{\mathbf{y} \sim \mathrm{Bern}(p)}[\lambda \ell_+(t) + \mu \ell_-(t) + \nu \ell_?(\mathbf{y})]$$

$$= \arg\min_{t \in \mathbb{R} \cup \{?\}} p \cdot \left(\mu \ell_-(t) + \nu \alpha_+\right) + (1 - p) \cdot \left(\lambda \ell_+(t) + \nu \alpha_-\right).$$

Our main result in this section is the feasibility of efficiently constructing selective classifiers:

**Theorem 3.1** (Constructing selective omnipredictors)**.** *Let $\mathcal{C}$ be a concept class of concepts $c : \mathcal{X} \to [-M, M]$, $\mathcal{D}$ a distribution on $\mathcal{X} \times \{0, 1\}$, $\epsilon > 0$, and $\mathcal{L}$ a family loss functions with associated weights $(\lambda, \mu, \nu)$ with $\lambda + \mu + \nu \leq 1$, such that all $\ell \in \mathcal{L}$ are B-Lipschitz. Then, a $(\mathcal{C}, \epsilon)$-multicalibrated predictor is a $(\mathcal{L}, \mathcal{C}, 4\epsilon\beta + \epsilon B)$-selective omnipredictor, where $\beta$ is an absolute bound on $\ell_+, \ell_-, \ell_?$.*

The full proof of Theorem 3.1 is deferred to the appendix; here we provide a proof sketch.

*Proof sketch of Theorem 3.1.* Our algorithm first constructs a $(\mathcal{C}, \epsilon)$-multicalibrated predictor $h$. For each specific choice of $\ell = (\ell_+, \ell_-, \ell_?) \in \mathcal{L}$ and weights $(\lambda, \mu, \nu)$, we apply the post-processing function $k^*_{\ell_\pm, \ell_?}$ to $h$. We show that the generalized Chow loss incurred by $k^*_{\ell_\pm, \ell_?} \circ h$ is no more (within an $\epsilon$ slack) than that incurred by the best classifier in $k \circ \mathcal{C}$, where $k : [0, 1] \to \mathbb{R} \cup \{?\}$ is any post-processing function adding abstentions. We outline why this is an optimal and efficient strategy.

Following the generalized Chow loss expression (Definition 2.6), for each $t \in \mathbb{R}$, we either pay the cost of predicting, whose expected value we denote by $\kappa_{\text{pred}}$, or the cost of abstaining, whose expected value we denote by $\kappa_{\text{abs}}$:

$$\kappa_{\text{pred}}(p) = \min_{t \in \mathbb{R}} p \cdot \mu\ell_-(t) + (1-p) \cdot \lambda\ell_+(t),$$

$$\kappa_{\text{abs}}(p) = \nu\big(p \cdot \ell_?(1) + (1-p) \cdot \ell_?(0)\big) = \nu\big(p \cdot \alpha_+ + (1-p) \cdot \alpha_-\big).$$

For each point $p$, in order to minimize the expected generalized Chow loss under the Bernoulli distribution, we proceed in two steps: (1) Find the value $t^*(p)$ that minimizes the value of $\kappa_{\text{pred}}(p)$ given a fixed value of $p$. This depends only on the choice of $\lambda, \ell_+, \mu, \ell_-$ and not on the underlying data. Specifically, this corresponds to finding the value $t^*(p)$ that minimizes the function $k^*_\ell(p)$, where $k^*_\ell(p) = \text{argmin}_{t \in \mathbb{R}} \mathbb{E}_{\mathbf{y} \sim \text{Bern}(p)}[\lambda\ell_+(\mathbf{y}, t) + \mu\ell_-(\mathbf{y}, t)]$. (2) For each predicted value $h(x) = p$, we then map it to either $t^*(p)$ or to '?' depending on whether it is cheaper to predict or to abstain; that is, depending on the value of $\min\{\kappa_{\text{pred}}(t^*(p)), \kappa_{\text{abs}}(p)\}$.

In other words, we can re-write our post-processing function $k^*_{\ell_\pm, \ell_?}$ as $k_{\text{abs}} \circ k^*_\ell$, where

$$k_{\text{abs}} = \begin{cases} t^* & \text{if } \kappa_{\text{pred}} \leq \kappa_{\text{abs}}, \\ ? & \text{if } \kappa_{\text{pred}} > \kappa_{\text{abs}}. \end{cases}$$

Then, the value of $k^*_{\ell_\pm, \ell_?}$ can be computed efficiently because (1) computing the value of $\kappa_{\text{pred}}(p)$ corresponds to solving a one-dimensional minimization problem, and (2) the value of $k^*_{\ell_\pm, \ell_?}$ is then fully determined from the values of $\kappa_{\text{pred}}(t^*(p))$ and of $\kappa_{\text{abs}}(p)$. To show optimality, we consider the level set $\mathcal{X}_{(p, \gamma)} = \{x \in \mathcal{X} \mid h(x) = p, c = \gamma\}$ for each $p \in \text{range}(h)$ and each $\gamma$ in the range of $c$. For each set $\mathcal{X}_{(p, \gamma)}$, we split the proof into 4 cases, depending on whether $h$ and $c$ abstain or predict as per their respective decision rules. The key idea is that, even though the concept $c$ abstains or predicts using an arbitrary decision rule, multicalibration ensures that the expected value of $y$ on $\mathcal{X}_{(p, \gamma)}$ is approximately $p$, which allows us to show that $k^*_{\ell_\pm, \ell_?}(h)$ incurs no more loss than $k \circ c$. $\qquad \square$

**Interval of abstention.** We further show the following about the points where $k^*_{\ell_\pm, \ell_?} \circ h$ abstains:

**Lemma 3.2.** *Given any loss functions $\ell_+, \ell_-, \ell_? \in \mathcal{L}$, the points $x \in \mathbb{R}$ such that $k^*_{\ell_\pm, \ell_?}(x) = ?$ form a contiguous interval, which we denote by $I_{\text{abs}}$.*

This follows from the fact that $\kappa_{\text{pred}}(p)$ is concave as a function of $p$, and that $\kappa_{\text{abs}}(p)$ is affine in $p$. Lemma 3.2 does not require the loss functions to be convex; only affine in $p$. Importantly, note that we can determine $I_{\text{abs}}$ directly from the chosen triplet of functions $(\ell_+, \ell_-, \ell_?)$, without any dependence on the underlying data. This is why our method is highly efficient: once we run the multicalibration algorithm, we directly apply our off-the-shelf post-processing function $k^*_{\ell_\pm, \ell_?}$.

## 3.1 Selective omniprediction in action

While our results are of theoretical nature, we provide experiments to demonstrate the feasibility of selective omniprediction in practice and to provide some concrete examples of the function $k^*_{\ell_\pm, \ell_?}$ for specific choices of triplets of loss functions $(\ell_+, \ell_-, \ell_?)$.

We generate synthetic data to create a binary classification problem with $n = 10{,}000$ samples and implement the multicalibration algorithm to baseline predictions to obtain a $\mathcal{C}$-multicalibrated predictor $h$, where we set $\mathcal{C}$ to be the concept class $\mathcal{C}$ of decision trees of depth 3. For various choices of loss functions $\ell = (\ell_+, \ell_-, \ell_?)$, we compare the coverage (i.e., the fraction of the points in the domain on which the predictor does not abstain) and total loss of our post-processed $k^*_{\ell_\pm, \ell_?} \circ h$ predictor (where $k^*_{\ell_\pm, \ell_?}$ is chosen for each $\ell$ but $h$ is the same predictor across all loss functions) with that of a predictor optimized specifically to minimize the generalized Chow loss function, using the same base concept class $\mathcal{C}$ of decision trees of depth 3. The bar plots show the points $p \in [0, 1]$ where $k^*_{\ell_\pm, \ell_?} \circ h$ abstains for different triplets of loss functions, indicated in the caption of each subfigure. We compute $I_{\mathrm{abs}}$ theoretically for each of these triplets, independently of the data (see calculations in the appendix); in Figure 1 one can see that these are indeed the regions where our algorithm abstains in practice. Table 1 shows the final loss and coverage incurred by our single selective omnipredictor (post-processed accordingly) compared to those obtained by each of the decision trees, which are trained separately for each of the triplet of loss functions. The full details of the experimental set-up along with further examples and repetitions can be found in the appendix.

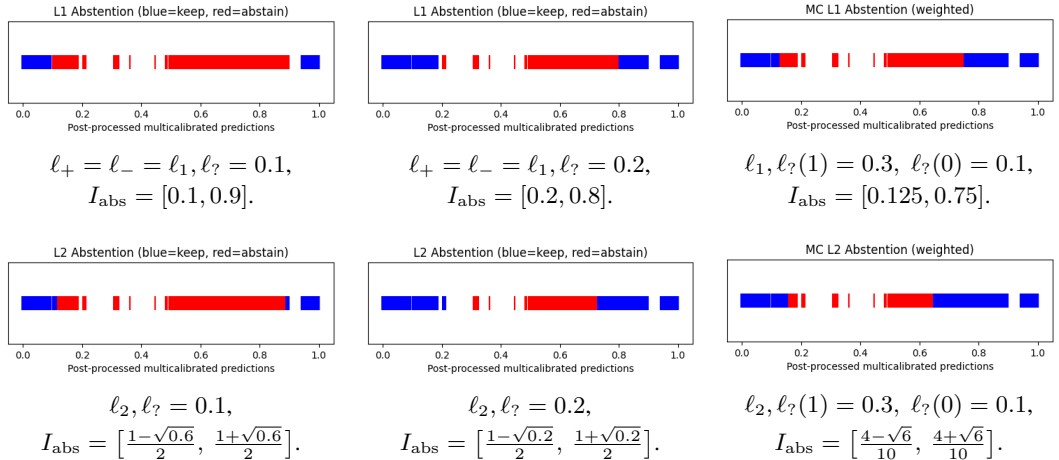

Figure 1: Final loss and coverage incurred by our single selective omnipredictor, post-processed with $k^*_{\ell_\pm, \ell_?}$ for each of the specific triplets indicated in the subfigure caption. Red: abstain. Blue: predict.

Table 1: Comparison of coverage and losses. "Cov" stands for *coverage*, "Pred" for *predicted loss* (i.e., over the non-abstaining region), and "Total" for the total generalized Chow loss.

| | $\ell_? = 0.1$ | | | $\ell_? = 0.2$ | | | $\ell_?(1) = 0.3, \ \ell_?(0) = 0.1$ | |
| | $k^*_{\ell_\pm, \ell_?} \circ h$ | Abst. DT | | $k^*_{\ell_\pm, \ell_?} \circ h$ | Abst. DT | | $k^*_{\ell_\pm, \ell_?} \circ h$ | Abst. DT |
|---|---|---|---|---|---|---|---|---|
| $\ell_1$ | Cov: 22%
Pred: 0.045
Total: 0.088 | Cov: 2.3%
Pred: 0.001
Total: 0.098 | $\ell_1$ | Cov: 52.70%
Pred: 0.083
Total: 0.139 | Cov: 3.8%
Pred: 0.002
Total: 0.194 | $\ell_1$ | Cov: 42.90%
Pred: 0.058
Total: 0.144 | Cov: 2.5%
Pred: 0.002
Total: 0.198 |
| $\ell_2$ | Cov: 24.15%
Pred: 0.046
Total: 0.087 | Cov: 3.8%
Pred: 0.001
Total: 0.098 | $\ell_2$ | Cov: 61.15%
Pred: 0.078
Total: 0.125 | Cov: 13.2%
Pred: 0.010
Total: 0.183 | $\ell_2$ | Cov: 64.6%
Pred: 0.082
Total: 0.122 | Cov: 13%
Pred: 0.0017
Total: 0.193 |

## 3.2 Building general reliable agnostic learners & conformal prediction

We show how we can use our efficient construction of selective omnipredictors to recover and generalize the results on reliable agnostic learning obtained by [43] from 0-1 loss to an entire family of loss functions (as defined in Section 2.3).

**Theorem 3.3.** *Let $\mathcal{C}$ be a concept class of Boolean concepts, $\mathcal{L}$ any class of B-Lipschitz loss functions, and $\mathcal{D}$ a distribution on $\mathcal{X} \times \mathcal{Y}$. If $\mathcal{C}$ is agnostically learnable under $\mathcal{D}$ in time $T(\epsilon, \delta)$, then $\mathcal{C}$ is $\mathcal{L}$-fully reliably learnable under $\mathcal{D}$ in time $T(\epsilon^2/6, \delta)$.*

**Conformal prediction from FRL.** Lastly, we demonstrate a useful application of the reliable agnostic learning framework. Specifically, we show that one can view FRL as a conformal prediction method for the case of binary classification, where we view the $\{0, 1\}$ prediction set as '?'.

In the classification setting, the goal of conformal prediction is to construct a *prediction set* of the possible labels $\mathcal{S}(x) \subseteq \mathcal{Y}$ for each point $x \in \mathcal{X}$ such that $\Pr_{(\mathbf{x},\mathbf{y})\sim\mathcal{D}}[\mathbf{y} \in \mathcal{S}(\mathbf{x})] \geq 1 - \epsilon$ for a chosen error rate $\epsilon \in (0, 1)$. This is known as the $\epsilon$-*marginal coverage guarantee*. In the case of binary classification, the possible prediction sets can only be $\{0\}, \{1\}, \{0, 1\}$. We show the following:

**Lemma 3.4.** *Let $h_? : \mathcal{X} \to \{0, 1, ?\}$ be an FRL predictor for $\mathcal{C}, \mathcal{D}, \epsilon$ and the 0-1 loss. Then, the prediction sets induced by the level sets of h, where we map $0 \mapsto \{0\}$, $1 \mapsto \{1\}$, and $? \mapsto \{0, 1\}$, satisfy the $\epsilon$-marginal coverage guarantee.*

**Remark 3.1.** While any conformal prediction method must definitionally satisfy the $\epsilon$-marginal coverage guarantee, typical algorithms offer no theoretical bounds on the size of the prediction sets. In the case of binary classification, this means that we could have an arbitrarily large number of points mapped to $\{0, 1\}$ (even the entire domain, which is a trivial way of satisfying the marginal coverage guarantee). However, FRL does provide provable abstention guarantees with respect to a base concept class $\mathcal{C}$ of our choice. Let $h_? : \mathcal{X} \to \{0, 1, ?\}$ be an FRL predictor for the class $\mathcal{C}$, 0-1 loss, and $\epsilon > 0$, and let $\{0\}, \{1\}, \{0, 1\}$ be its induced prediction sets as a conformal prediction method. Then,

$$\Pr_{(\mathbf{x},\mathbf{y})\sim\mathcal{D}}[h_?(\mathbf{x}) = \{0, 1\}] \leq \min_{c_? \in \mathtt{SC}(\mathcal{C})} \Pr_{(\mathbf{x},\mathbf{y})\sim\mathcal{D}}[c_?(\mathbf{x}) = ?] + \epsilon.$$

We defer the proof in the appendix, where we also run experiments implementing FRL as a conformal prediction method.

## 4 Learning Abstentions Fairly

So far, we have considered the question of how to learn abstentions optimally, where we measure optimality in the agnostic sense, with respect to a base concept class $\mathcal{C}$ and a generalized Chow loss function $\ell_{\mathrm{GC}}$. The motivation for abstaining is to be able to make almost no errors when predicting, which we can accomplish by mapping the uncertain points to '?' instead. However, we might additionally have fairness concerns: suppose that we have a collection $\mathcal{G} = \{g : \mathcal{X} \to \{0, 1\}\}$ of subgroups of interest of the domain. Besides achieving high accuracy over the points where we do predict a numerical value, we would also like to abstain fairly on *each* of the groups $g \in \mathcal{G}$, so that we avoid achieving high global accuracy at the expense of overly abstaining on some subgroups. Motivated by the multigroup fairness framework, we also want the groups in $\mathcal{G}$ to be able to intersect.

How can we measure how an optimal abstention rate looks like within each of the groups $g \in \mathcal{G}$? Motivated by the reliable agnostic learning framework [43], we do so by requiring our predictor to abstain no more than the optimal selective classifier $c_?^g \in \mathtt{SC}(\mathcal{C})$ (with an $\epsilon$ slack) on *each* group $g \in \mathcal{G}$, where $c_?^g$ can naturally be different for each group. That is, similar to our notion of a selective omnipredictor, we want to construct a single classifier simultaneously for all groups, but its abstention rate competes with that of a $c_?^g$ that is chosen optimally in each group. Formally, we introduce the following definition, which corresponds to the multigroup version of the original notion of realiable agnostic learning:

**Definition 4.1** (($\mathcal{C}, \mathcal{G}$)–multigroup selective classification). Given a collection of subgroups $\mathcal{G}$ of $\mathcal{X}$, a concept class $\mathcal{C}$, distribution $\mathcal{D}$, and $\epsilon > 0$, we say that a predictor $h_? : \mathcal{X} \to [0, 1] \cup \{?\}$ is a ($\mathcal{C}, \mathcal{G}, \epsilon$)-*multigroup selective classifier* if the following two conditions are satisfied:

1. **Global accuracy.** $\mathrm{err}_{\mathcal{D}}(h_?) := \mathbb{E}_{(\mathbf{x},\mathbf{y})\sim\mathcal{D}}[|\mathbf{y} - h_?(\mathbf{x})| \cdot \mathbb{1}[h_?(\mathbf{x}) \neq ?]] \leq \epsilon.$

2. **Optimal local abstention rate.** For every $g \in \mathcal{G}$,

$$\Pr_{(\mathbf{x},\mathbf{y})\sim\mathcal{D}}\left[g(\mathbf{x}) \cdot \mathbb{1}[h_?(\mathbf{x}) = ?]\right] \leq \min_{c_? \in \mathtt{SC}(\mathcal{C})} \Pr_{(\mathbf{x},\mathbf{y})\sim\mathcal{D}}\left[g(\mathbf{x}) \cdot \mathbb{1}[c_?(\mathbf{x}) = ?]\right] + \epsilon.$$

Note that Condition 2 automatically implies an optimal global abstention rate as well, by applying the local condition with $g = \mathbf{1}$ (which we can assume is always contained in $\mathcal{G}$). We remark that the multigroup fairness and omniprediction literature had so far always taken $\mathcal{C} = \mathcal{G}$; our notion demonstrates why it is useful to separate the base concept class $\mathcal{C}$ from the collection of groups $\mathcal{G}$.

We show that we can efficiently construct a $(\mathcal{C}, \mathcal{G}, \epsilon)$-multigroup selective classifier for any $\mathcal{C}, \mathcal{G}$ from a multiaccurate predictor for the class $\mathcal{C} \cdot \mathcal{G} = \{cg \mid c \in \mathcal{C}, g \in \mathcal{G}\}$ that is also globally calibrated:

**Theorem 4.1.** *Given access to a $(\mathcal{C} \cdot \mathcal{G}, \epsilon^2/8)$-multiaccurate and calibrated predictor, we can efficiently construct a $(\mathcal{C}, \mathcal{G}, \epsilon)$-multigroup selective classifier in time* $\mathrm{poly}(1/\epsilon)$.

The key idea in our proof is to convert a $(\mathcal{C}\cdot\mathcal{G}, \epsilon)$-multiaccurate and calibrated predictor $h : \mathcal{X} \to [0, 1]$ into a selective classifier by mapping all $x$ such that $h(x) \in (\epsilon, 1 - \epsilon)$ to '?'. For each group $g$, let $c_?^g = (c_+^g, c_-^g)$ be an optimal selective classifier in $\mathrm{SC}(\mathcal{C})$ within $g$. Since $c_+^g \in \mathcal{C}^+$, it follows that whenever $c_+^g = 1$, the true label $y$ on that point is also 1. By the multiaccuracy guarantee for $c_+^g$ and $g$ (which is in $\mathcal{C} \cdot \mathcal{G}$), we obtain that $\mathbb{E}_{\mathcal{D}}[\mathbf{y}] \approx \mathbb{E}_{\mathcal{D}}[h(\mathbf{x})] \approx 1$ in the region where $c_+^g(x) = 1, g(x) = 1$. A symmetric argument holds with $c_-^g$ and $g$. We use the global calibration condition to ensure that our thresholded $h_?$ remains accurate in the entire domain.

It is natural to ask whether we can efficiently construct $(\mathcal{C}, \mathcal{G})$-multigroup selective classifiers starting from a weaker learning primitive than a weak agnostic learner for $\mathcal{C} \cdot \mathcal{G}$. We answer this question in the positive in the case where $|\mathcal{G}|$ is small:

**Lemma 4.2.** *If the class $\mathcal{C}$ is fully reliably learnable for the $\ell_1$ loss and $\epsilon > 0$, we can construct a $(\mathcal{C}, \mathcal{G}, \epsilon)$-multigroup selective classifier in time* $\mathrm{poly}(|\mathcal{G}|, 1/\epsilon)$ *with oracle access to the full reliable learner for $\mathcal{C}$.*

Answering this question in generality (i.e., where $\mathcal{G}$ can be arbitrarily large) appears to be a very interesting open question.

*Calibrated multiaccuracy and multicalibration.* We make a further remark about how the various multigroup fairness definitions relate to selective classification (which we show in the appendix).

(a) If we have the stronger primitive of a $(\mathcal{C} \cdot \mathcal{G})$-multicalibrated predictor (which implies a $(\mathcal{C} \cdot \mathcal{G})$-multiaccurate calibrated predictor), then we can have a non-selective predictor which would also give local agnostic guarantees. This follows from the works of [25, 10, 27].

(b) Given a $(\mathcal{C} \cdot \mathcal{G}, \epsilon)$-multiaccurate and calibrated predictor, we can directly obtain FRL predictors for the class $\mathcal{C}$ for the $\ell_1$ loss by thresholding the predictor as we do in the proof of Theorem 4.1. Given that calibrated multiaccuracy implies agnostic learning [11], it is already implied by Theorem 4.1 that we can achieve reliable agnostic learning from calibrated multiaccuracy. However, this approach gives a direct reduction. Reliable agnostic learning is believed to be a weaker learning primitive than agnostic learning [45].

## 4.1 Conformal prediction from $(\mathcal{C}, \mathcal{G})$-multigroup selective classification

Similar to Section 3, we can view $(\mathcal{C}, \mathcal{G})$-multigroup selective classification as a conformal prediction method in the case of binary classification. Besides the marginal coverage guarantee, now that we have a collection $\mathcal{G}$ of groups one can also hope to satisfy a conditional version of coverage. Namely, for every $g \in \mathcal{G}$, we want to satisfy $\Pr_{(\mathbf{x},\mathbf{y}) \sim \mathcal{D}}[\mathbf{y} \in \mathcal{S}(\mathbf{x}) \mid g(\mathbf{x}) = 1] \geq 1 - \epsilon$. This property is known as the $(\mathcal{G}, \epsilon)$-*group conditional coverage guarantee* [39]. We show that $(\mathcal{C}, \mathcal{G})$-multigroup selective classifiers do indeed satisfy this conditional guarantee:

**Lemma 4.3.** *Let $h_? : \mathcal{X} \to [0, 1] \cup \{?\}$ be a $(\mathcal{C}, \mathcal{G}, \epsilon)$-multigroup selective classifier. Then, the prediction sets induced by the level sets of $h$, where we map $[0, \epsilon] \mapsto \{0\}, [1 - \epsilon, 1] \mapsto \{1\}$, and $(\epsilon, 1 - \epsilon) \mapsto \{0, 1\}$ satisfy $\Pr_{(\mathbf{x},\mathbf{y}) \sim \mathcal{D}}[\mathbf{y} \in \mathcal{S}(x) \mid g(\mathbf{x}) = 1] \geq 1 - \frac{\epsilon}{\Pr_{\mathcal{D}}[g(\mathbf{x})=1]}$ for all $g \in \mathcal{G}$.*

**Remark 4.1.** As pointed out in Section 3, typical conformal prediction methods offer no theoretical bounds on the size of the prediction sets. Through our framework of $(\mathcal{C}, \mathcal{G})$-multigroup selective classification, however, we do obtain provable abstention guarantees for each $g \in \mathcal{G}$ with respect to a base concept class $\mathcal{C}$ of our choice. Specifically, for each $g \in \mathcal{G}$, the prediction sets of our $(\mathcal{C}, \mathcal{G})$-multigroup selective classifier $h_?$ as specified in Lemma 4.3 satisfy

$$\Pr_{(\mathbf{x},\mathbf{y}) \sim \mathcal{D}} \left[ g(\mathbf{x}) \cdot \mathbb{1}[h_?(\mathbf{x}) = \{0, 1\}] \right] \leq \min_{c_? \in \mathrm{SC}(\mathcal{C})} \Pr_{(\mathbf{x},\mathbf{y}) \sim \mathcal{D}}[c_?(\mathbf{x}) = ?] + \epsilon.$$

Importantly, $(\mathcal{C}, \mathcal{G})$-multigroup selective classification as a conformal prediction method ensures group conditional coverage and a provable abstention bound on each group *even when these intersect*.

In the appendix we implement $(\mathcal{C}, \mathcal{G})$-multigroup selective classifiers by adapting the multicalibration algorithm and demonstrate its utility as a conformal prediction method in practice.

# 5    Conclusion and Future Work

We conclude by providing some directions for future work.

**The complexity of reliable agnostic learning.** The first is concerned with the dependence on $\epsilon$ when obtaining a fully reliable learner from agnostic learning. In the case of [44], they are able to learn PRL, NRL, and FRL predictors for a concept class $\mathcal{C}$ in time $T(O(\epsilon^2))$ using an agnostic learner for $\mathcal{C}$ that runs in time $T(\epsilon)$. In our case, when we obtain fully reliable learners from selective omnipredictors in Theorem 3.3, in order to obtain a $\mathcal{L}$-fully reliable learner with error $\epsilon$ we require a selective omnipredictor with error $\epsilon^2$. It appears that the nature of the two constraints in the definition of reliable agnostic learning (unlike the case of the generalized Chow loss formulation) induces this overhead, but it is unclear whether it is unavoidable.

**Building selective omnipredictors.** For our construction of selective omnipredictors, we require the multigroup fairness primitive of multicalibration. For the case of regular omniprediction (i.e., without abstentions), recent works have shown that we can construct omnipredictors from the weaker primitive of calibrated multiaccuracy, and even with weaker notions of global calibration [26, 59]. Our proof of selective omniprediction seems to require the full power of multicalibration; it is unclear whether we can relax it to calibrated multiaccuracy, or whether we can have a direct reduction from omniprediction.

**Weak agnostic learner for $\mathcal{C} \cdot \mathcal{G}$.** Our construction of a $(\mathcal{C}, \mathcal{G})$-multigroup selective classifier in Section 4 requires access to a $(\mathcal{C} \cdot \mathcal{G})$-multiaccurate and calibrated predictor. From the works on multigroup fairness [26, 11], this in requires access to a weak agnostic learner for the class $\mathcal{C} \cdot \mathcal{G}$. We do not know whether it is possible to construct $(\mathcal{C}, \mathcal{G})$-multigroup selective classifiers having only access to separate weak agnostic learners for $\mathcal{C}$ and $\mathcal{G}$, without requiring a weak agnostic learner for their intersection. This can be seen as a broader question about the learnability of intersections of concept classes.

**Building selective classifiers.** In Lemma 4.2 we show that we can construct a $(\mathcal{C}, \mathcal{G})$-multigroup selective classifier from a fully reliable learner, which is believed to be a weaker primitive than (weak) agnostic learning [45]. However, we are only able to show this for classes $\mathcal{G}$ that are small in size, given that we need to call the FRL oracle $|\mathcal{G}|$ times. Hence the question of whether we can build selective classifiers from a weaker primitive than agnostic learning for a general class $\mathcal{G}$ remains open.

**Conformal prediction & model multiplicity.** Lastly, in light of our connections between reliable agnostic learning and $(\mathcal{C}, \mathcal{G})$-multigroup selective classification with conformal prediction, it would be interesting to develop this connection further, particularly focusing on the ability of these methods to provide provable guarantees on the sizes of the prediction sets beyond the setting of binary classification. It also appears fruitful to study how our framework of learning with abstentions, where a predictor is able to measure its own reliability, relates to the recent works on model multiplicity.

## Acknowledgments

We thank Michael P. Kim and Inbal Livni Navon for helpful pointers. We are grateful to the participants of the Workshop on Predictions and Uncertainty at COLT 2025 for valuable feedback. During the development of this work, SC was supported by a Rhodes scholarship.

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

# A  Related Work

In recent years, there has been an increasingly close examination of algorithmic predictions, particularly when used in high-stakes settings. Most of these close examinations have focused on studying the potential biases present in these predictions, such as in the development of individual and group fairness notions [15, 30]. Another prominent example is the recent line of work on *multigroup fairness*, which aims to detect and avoid unwanted forms of bias on the outputs of a predictor $h$ that maps individuals in a domain $\mathcal{X}$ to values in $[0, 1]$, where the biases are measured with respect to a class $\mathcal{G}$ of subgroups of the population that can intersect [32, 50].

However, we also need to study the reliabilty of the predictions made by $h$. Some recent works have put forth the necessity of accounting for uncertainty in the setting of algorithmic fairness [2, 5, 40, 54, 49, 53, 13], but this question remains largely understudied. A major application of these notions is in learning theory through the new learning paradigm of omniprediction [24, 27, 16, 18, 17]. Multigroup fairnes notions can be understood as providing computational indistinguishability guarantees, formalized through the Outcome Indistinguishability framework [16] and follow-up works [10, 56, 11], which helps provide intuition for why they are well-suited as a method to add abstentions, as we do in our work. Recent work on calibration uses higher order calibration as a way to decompose the predictive uncertainty of a model into aleatoric and epistemic components [1].

We remark that other works on multicalibration and omniprediction do not separate $\mathcal{C}$ and $\mathcal{G}$ classes, which we do in this paper.

**Selective classification.** The idea of building an algorithm that abstains on certain individuals is not new and has traditionally been called *selective classification* [19]. A selective classifier is allowed to also return ? as an output (which indicates abstention), along with the usual numerical values. In the setting of fairness, some works have studied the effect of selectively abstaining. Jones et al. find that certain forms of selective classification can magnify disparities across groups [37], and follow-up works try to restrict this type of disparities [52, 62, 67]. Abstentions in learning have also been used to provide bounds on efficient learning in the presence of arbitrary covariate shifts [41, 42, 23, 22] and in the introduction of *partial* hypotheses classes [55, 34].

**Bayesian approaches & Conformal prediction.** Another major approach for the task of uncertainty quantification is that of Bayesian inference. These methods place distributions over the model parameters, and build a model by iteratively updating the prior with new data in order to obtain the posterior through an application of Bayes rule [9]. For training neural networks in a way that also allows for quantifying uncertainty, incorporating these techniques yields the so-called *Bayesian Neural Networks* (BNNs) [21]. Some fairness works that are also concerned with incorporating uncertainty have used BNNs in the context of fair prediction [5, 40, 33, 64]. However, Bayesian methods tend to be very slow and largely intractable.

An alternative to Bayesian-based approaches that is gathering growing popularity is that of *conformal prediction*, first proposed by by Gammerman, Vovk, and Vapnik [66, 3]. Conformal prediction is a technique for determining precise levels of confidence which can be applied to any method that has already been trained on the data [63]. Instead of a point-prediction (e.g., 0.8 for individual $x$), with conformal prediction we can also return a *prediction interval* that indicates the confidence of the algorithm on that prediction. The wider the interval, the lower the confidence and the higher the uncertainty. Some recent works have extended the conformal prediction setting to provide conditional guarantees instead of only marginal guarantees by adapting the multicalibration algorithm [38, 39].

**Reliable learning.** Still, none of the previous works studies abstention from a theoretical perspective. In the formal setting of learning theory, the study of selective classification was initiated in 2009 by Kalai, Kanade, and Mansour, who called it "reliable learning" [43]. How can we come up with a formal model of classifiers that abstain, what does it mean to "abstain optimally", and how can we learn such classifiers?

The authors answer this by adapting the original agnostic learning framework [31, 47] to what they call *agnostic reliable learning*. Here, the goal is to output a selective classifier whose accuracy nearly matches the accuracy of the best selective classifier from a pre-specified concept class. [43] show that if a concept class $\mathcal{C}$ is agnostically learnable, then it is also agnostic reliable learnable. In the other direction, due to follow-up work by Kanade and Thaler it is widely believed that reliable agnostic learning is easier than agnostic learning [45].

**PQ-learning and Chow's model of abstention.** In 2020, Goldwasser, Kalai, Kalai, and Montasser presented a model of learning meant to tackle the covariate shift problem, in which the training data is distributed according to $P$ and the test data according to $Q$, where $P$ and $Q$ can be arbitrary distributions over the domain $\mathcal{X}$ [23]. This form of learning is not possible to achieve in general, given that $P$ and $Q$ might not even overlap.

To make this problem tractable, Goldwasser et al. introduce the model of *PQ-learning*, where the learner has access to unlabeled test examples from $Q$ and the option to abstain on any point $x \in \mathcal{X}$. We compute the rejection rate $\epsilon_1$ of the algorithm (i.e., the fraction of $\mathcal{X}$ over which the classifier abstains) and the misclassification rate $\epsilon_2$, which quantifies the error of the classifier only over the subset of the domain on which the classifier does not abstain. Goldwasser et al. give algorithms for building selective classifiers in the PQ-learning model which guarantee low test error rate and low rejection rate with respect to $P$ for concept classes of bounded VC dimension [23]. Their algorithm is efficient if we have access to an Empirical Risk Minimizer (ERM) for $\mathcal{C}$. For classes $\mathcal{C}$ of bounded VC dimension, being able to do ERM efficiently is equivalent to proper agnostic learning [42].

In a follow-up work, Kalai and Kanade then showed that PQ-learning is equivalent to reliable learning. Moreover, they provide further evidence that the computational hardness of PQ-learning and reliable learning lies in-between PAC and agnostic learning (under the usual hardness assumptions) [42]. This separation was already shown by Kanade and Thaler, who gave an algorithm for reliably learning majorities over $\{0,1\}^d$ in time $2^{\tilde{O}(\sqrt{d})}$, whereas there are no known agnostic learning algorithms for this problem that run in time less than $2^{\Omega(d)}$ [45].

In another follow-up work, Kalai and Kanade consider a different formulation of the selective classification problem considered by Goldwasser et al., which is based on the slightly different and more general framework of Chow's abstention model [12]. Here, instead of finding a trade-off between error rates $\epsilon_1, \epsilon_2$, we have a fixed parameter $\alpha > 0$ which corresponds to the abstention cost. I.e., for each $x \in \mathcal{X}$, we either make a prediction $\hat{y}$ and suffer loss $\ell(y, \hat{y}) = |y - \hat{y}|$, or we abstain and pay a price of $\alpha$. Importantly, this is a stronger model than PQ-learning/reliable learning, given that they are able to by-pass the lower bounds shown in [23]. For this reason, in this paper we prove our main results using (a generalized version of) Chow's abstention model, given that it is the strongest of the three formal learning with abstention models.

**Multigroup fairness and omniprediction.** In recent years, the algorithmic fairness literature has developed a rich line of work on multigroup fairness notions, with applications to the covariate shift problem, complexity theory, causal inference, the model multiplicity problem, and conformal prediction, among many others [32, 29, 48, 51, 61, 14, 58, 60].

A major application of these notions is in learning theory through the new learning paradigm of omniprediction [24, 27, 17]. Multigroup fairnes notions can be understood as providing computational indistinguishability guarantees, formalized through the Outcome Indistinguishability framework [16] and follow-up works [10], which helps provide intuition for why they are very well-suited as a method to add abstentions, as we do in our work. Recent work on calibration uses higher order calibration as a way to provably decompose the predictive uncertainty of a model into aleatoric and epistemic components [1].

**Variance and predictive multiplicity.** Some works that studying the reliability of predictors focus on the *predictive multiplicity problem*, which is concerned with the following fact: for a given fixed dataset, there are multiple ways in which we can train a predictor on the dataset such that it achieves high accuracy, but these various potential and equally good predictors can then disagree on individual predictions [7, 57]. Various metrics have been proposed for quantifying the variance of the predictions within the class $\mathcal{M}$ [57], and various algorithms have been proposed for ensembling these various competing models in different ways [6, 60, 4].

Several works have studied the relationship between the variance within the class $\mathcal{M}$ and group fairness metrics [54, 53, 2, 49, 36]. Notably, following this variance approach, Cooper et al. find that we can obtain close-to-fair predictions simply by abstaining on the individuals with high variance [13]. A drawback of these variance-based methods is that they require fitting an entire class $\mathcal{M}$ of models.

# B  Deferred Definitions

## B.1  Reliable agnostic learning

The framework of reliable agnostic learning was first proposed by Kalai, Kanade, and Mansour for the case of zero-one loss [43]. In the main body, we generalized the definitions to a family of loss functions $\mathcal{L}$. Here, we state the original definitions for the 0-1 loss, which help grasp the notions of PRL, NRL, and FRL. We first recall the usual definitions of the error, false positive rate, and false negative rate of a predictor $h$ in the case of 0-1 loss and for Boolean labels $\mathcal{Y} = \{0, 1\}$.

$$\mathrm{err}(h, \mathcal{D}) = \Pr_{(\mathbf{x},\mathbf{y})\sim\mathcal{D}}[h(\mathbf{x}) \neq \mathbf{y}],$$

$$\mathrm{false}_+(h, \mathcal{D}) = \Pr_{(\mathbf{x},\mathbf{y})\sim\mathcal{D}}[c(\mathbf{x}) = 1 \wedge \mathbf{y} = 0],$$

$$\mathrm{false}_-(h, \mathcal{D}) = \Pr_{(\mathbf{x},\mathbf{y})\sim\mathcal{D}}[c(\mathbf{x}) = 0 \wedge \mathbf{y} = 1].$$

Note that $\mathrm{err}(h, \mathcal{D}) = \mathrm{false}_+(h, \mathcal{D}) + \mathrm{false}_-(h, \mathcal{D})$. We drop the distribution $\mathcal{D}$ when it can be inferred from context, and sometimes write $\mathrm{err}_{\mathcal{D}}(h)$ for $\mathrm{err}(h, \mathcal{D})$.

**Definition B.1** (Positive Reliable Learning (PRL) [43]). A concept class $\mathcal{C}$ of Boolean concepts is *positive reliably learnable* if there exists a learning algorithm that for any distribution $\mathcal{D}$ over $\mathcal{X} \times \{0, 1\}$, any $\epsilon, \delta > 0$, with access to the example oracle $\mathsf{EX}(\mathcal{D})$, outputs a hypothesis $h : \mathcal{X} \to \{0, 1\}$ that satisfies the following with probability at least $1 - \delta$,

1. $\mathrm{false}_+(h, D) \leq \epsilon$,

2. $\mathrm{false}_-(h, D) \leq \min_{c_+ \in \mathcal{C}^+} \mathrm{false}_-(c_+, D) + \epsilon$, where $\mathcal{C}^+ = \{c \in \mathcal{C} \mid \mathrm{false}_+(c, \mathcal{D}) = 0\}$.

Symmetrically, we have that:

**Definition B.2** (Negative Reliable Learning [43]). A concept class $\mathcal{C}$ of Boolean concepts is *negative reliably learnable* if there exists a learning algorithm that for any distribution $\mathcal{D}$ over $\mathcal{X} \times \{0, 1\}$, any $\epsilon, \delta > 0$, with access to the example oracle $\mathsf{EX}(\mathcal{D})$, outputs a hypothesis $h : \mathcal{X} \to \{0, 1\}$ that satisfies the following with probability at least $1 - \delta$,

1. $\mathrm{false}_-(h, \mathcal{D}) \leq \epsilon$, and

2. $\mathrm{false}_+(h, \mathcal{D}) \leq \min_{c_- \in \mathcal{C}^-(\mathcal{D})} \mathrm{false}_+(c_-, \mathcal{D}) + \epsilon$, where $\mathcal{C}^- = \{c \in \mathcal{C} \mid \mathrm{false}_-(c, \mathcal{D}) = 0\}$.

**Full reliability.** Both PRL and NRL are non-selective classifiers; the hypothesis $h$ in the definitions of PRL and NRL map from $\mathcal{X}$ to $\{0, 1\}$. But if we want *both* the positive and the negative rates to be low (i.e., for the total error to be low), then this is not possible unless we allow for ? to be in the range of $h$ as well. Recall the definition of the class $\mathsf{SC}(\mathcal{C})$ derived from the concept class $\mathcal{C}$ (Section 2.3). By the definitions of $\mathcal{C}^+$ and $\mathcal{C}^-$, note that all concepts in $\mathsf{SC}$ have 0 error over the non-abstaining region.

We further define the uncertainty $?(h_?, \mathcal{D})$ and accuracy $\mathrm{acc}(h_?, \mathcal{D})$ of a Boolean selective classifier $h_?$ as follows:

$$?(h_?, \mathcal{D}) = \mathbb{E}_{(\mathbf{x},\mathbf{y})\sim\mathcal{D}}[h(\mathbf{x}) = ?].$$

$$\mathrm{acc}(h_?, \mathcal{D}) = \mathbb{E}_{(\mathbf{x},\mathbf{y})\sim\mathcal{D}}[\mathbb{1}[h_?(\mathbf{x}) = 0] \cdot (1 - \mathbf{y}) + \mathbb{1}[h_?(\mathbf{x}) = 1] \cdot \mathbf{y}],$$

One can see that $\mathrm{err}(h_?, \mathcal{D}) + ?(h_?, \mathcal{D}) + \mathrm{acc}(h_?, \mathcal{D}) = 1$.

Full reliable learning is then defined as follows:

**Definition B.3** (Full Reliable Learning (FRL) [43]). A concept class $\mathcal{C}$ of Boolean concepts is *fully reliably learnable* if there exists a learning algorithm that for any distribution $\mathcal{D}$ over $\mathcal{X} \times \{0, 1\}$, any $\epsilon, \delta > 0$, with access to the example oracle $\mathsf{EX}(\mathcal{D})$, outputs a selective classifier $h_? : \mathcal{X} \to \{0, 1, ?\}$ that satisfies the following with probability at least $1 - \delta$,

1. $\mathrm{err}(h_?, \mathcal{D}) = \mathrm{false}_+(h_?, \mathcal{D}) + \mathrm{false}_-(h_?, \mathcal{D}) \leq \epsilon$,

2. $\mathrm{acc}(h_?, \mathcal{D}) \geq \max_{c_? \in \mathsf{SC}(\mathcal{C})} \mathrm{acc}(c_?, \mathcal{D}) - \epsilon$.

Alternatively, using the fact that $\mathrm{err}(h_?, \mathcal{D}) + ?(h_?, \mathcal{D}) + \mathrm{acc}(h_?, \mathcal{D}) = 1$, we can write the FRL definition with the formulation used in [45], where Condition 2 is expressed in terms of the abstention rate rather than in terms of accuracy. Indeed, we can equivalently define FRL as follows:

**Definition B.4** (Full Reliable Learning (FRL) [43]). A concept class $\mathcal{C}$ of Boolean concepts is *fully reliably learnable* if there exists a learning algorithm that for any distribution $\mathcal{D}$ over $\mathcal{X} \times \{0, 1\}$, any $\epsilon, \delta > 0$, with access to the example oracle $\mathsf{EX}(\mathcal{D})$, outputs a selective classifier $h : \mathcal{X} \to \{0, 1, ?\}$, that satisfies the following with probability at least $1 - \delta$,

1. $\mathrm{err}(h_?, \mathcal{D}) = \mathrm{false}_+(h_?, \mathcal{D}) + \mathrm{false}_-(h_?, \mathcal{D}) \leq \epsilon$,

2. $?(h_?, \mathcal{D}) \leq \min_{c_? \in \mathsf{sc}(\mathcal{C})} ?(c_?, \mathcal{D}) + \epsilon$.

When instantiating our generalized definitions (Section 2.3) to the case of the 0-1 loss, $\ell_+$ corresponds to $\mathrm{false}_+$, $\ell_-$ corresponds to $\mathrm{false}_-$, and $\ell_?$ to $?$.

The original definitions of PRL, NRL, and FRL were introduced only for the case of the 0-1 loss [43]. For this specific choice of loss function, Kalai, Kanade, and Mansour showed the following:

**Theorem B.1** ([43]). *Let $\mathcal{L}$ contain only the 0-1 loss. If a concept class $\mathcal{C}$ is agnostically learnable under distribution $\mathcal{D}$ in time $T(\epsilon, \delta)$, then $\mathcal{C}$ is $\mathcal{L}$-positively reliably learnable and $\mathcal{L}$-negative reliably learnable, both in time $T(\epsilon^2/2, \delta)$. Then, $\mathcal{C}$ is also $\mathcal{L}$-full reliably learnable, in time $2T(\epsilon^2/8, \delta/2)$.*

Lastly, in the main body we deferred the definition of $\mathcal{L}$-NRL to the appendix, which we include for completeness:

**Definition B.5** (NRL for a family of loss functions [43, 45]). A concept class $\mathcal{C}$ of Boolean concepts is $\mathcal{L}$-*negatively reliably learnable* if there exists a learning algorithm that for any distribution $\mathcal{D}$ over $\mathcal{X} \times \{0, 1\}$, any $\ell_+, \ell_- \in \mathcal{L}$, and any $\epsilon, \delta > 0$, when given access to the example oracle $\mathsf{EX}(\mathcal{D})$, outputs a hypothesis $h : \mathcal{X} \to [0, 1]$ that satisfies the following with probability at least $1 - \delta$:

1. $\mathbb{E}_{(\mathbf{x}, \mathbf{y}) \sim \mathcal{D}}[\ell_-(h(\mathbf{x}))] \leq \epsilon$,

2. $\mathbb{E}_{(\mathbf{x}, \mathbf{y}) \sim \mathcal{D}}[\ell_+(h(\mathbf{x}))] \leq \min_{c_- \in \mathcal{C}^-} \mathbb{E}_{(\mathbf{x}, \mathbf{y}) \sim \mathcal{D}}[\ell_+(c_-(\mathbf{x}))] + \epsilon$.

## B.2 Agnostic learning

**Definition B.6** (Weak agnostic learning). Given a concept class $\mathcal{C}$ on $\mathcal{X}$ and a distribution $\mathcal{D}$ on $\mathcal{X} \times \mathcal{Y}$, a $(\alpha, \gamma)$-*weak agnostic learner* for $\mathcal{C}$, denoted $\mathrm{WAL}_{\mathcal{C}}$, is an algorithm that satisfies the following promise problem. Given a collection of labeled samples $(\mathbf{x}, \mathbf{y}) \sim \mathcal{D}$, if there is some $c \in \mathcal{C}$ such that $\mathrm{err}_{\mathcal{D}}(c) \leq \alpha$, then $\mathrm{WAL}_{\mathcal{C}}$ returns a hypothesis $h : \mathcal{X} \to [0, 1]$ such that $\mathrm{err}_{\mathcal{D}}(h) \leq \gamma$ with probability at least $1 - \delta$.

**Definition B.7** (Agnostic learning). Given a concept class $\mathcal{C}$ on $\mathcal{X}$ and a distribution $\mathcal{D}$ on $\mathcal{X} \times \mathcal{Y}$, a *(strong) agnostic learner* for $\mathcal{C}$ is an algorithm that, given a collection of labeled samples $(\mathbf{x}, \mathbf{y}) \sim \mathcal{D}$ and an error parameter $\epsilon > 0$, returns a hypothesis $h : \mathcal{X} \to [0, 1]$ satisfying, with probability at least $1 - \delta$,

$$\mathrm{err}_{\mathcal{D}}(h) \leq \min_{c \in \mathcal{C}} \mathrm{err}_{\mathcal{D}}(c) + \epsilon.$$

From the results of [44, 20] on agnostic boosting, we know that strong agnostic learning reduces to weak agnostic learning for any concept class $\mathcal{C}$.

Throughout the statements and proofs, by "efficiently" we mean that the algorithm runs in polynomial time in the appropriate parameters. We drop the failure probability $\delta$ from the statements.

# C Deferred Proofs from Section 3

## C.1 Selective omnipredictors

We emphasize that the selective omniprediction learning paradigm can be extremely useful in practice: once we have obtained a $(\mathcal{L}, \mathcal{C})$-selective omnipredictor, we can choose any loss function in $\mathcal{L}$ at a later time. For example, this allows us to change the abstention costs over time, or to change the cost of false positives or false negatives over time. One can envision many practical settings in which

this flexibility is highly desirable; e.g., if not catching patients with a specific illness becomes more dangerous over time. Note that our framework allows us to separate the costs of both prediction and abstention for the cases of $y = 1$ and $y = 0$.

**Theorem 3.1** (Constructing selective omnipredictors). *Let $\mathcal{C}$ be a concept class of concepts $c : \mathcal{X} \to [-M, M]$, $\mathcal{D}$ a distribution on $\mathcal{X} \times \{0, 1\}$, $\epsilon > 0$, and $\mathcal{L}$ a family loss functions with associated weights $(\lambda, \mu, \nu)$ with $\lambda + \mu + \nu \leq 1$, such that all $\ell \in \mathcal{L}$ are B-Lipschitz. Then, a $(\mathcal{C}, \epsilon)$-multicalibrated predictor is a $(\mathcal{L}, \mathcal{C}, 4\epsilon\beta + \epsilon B)$-selective omnipredictor, where $\beta$ is an absolute bound on $\ell_+, \ell_-, \ell_?$.*

**Remark C.1.** The condition $\lambda + \mu + \nu \leq 1$ is to ensure that the additive error term does not scale up; equivalently we could just get $2B(\lambda + \mu + \nu)\epsilon$ as the additive error.

*Proof.* Given the concept class $\mathcal{C}$ and parameter $\epsilon$, we discretize each $c \in \mathcal{C}$ to precision $\epsilon$ (i.e., into $\lceil 1/\epsilon \rceil$ many buckets). We denote these discretized concepts by $\hat{c}$ and the corresponding concept class by $\hat{\mathcal{C}}$. Because all loss functions are $B$-Lipschitz, discretizing the concepts to precision $\epsilon$ incurs an additive error of at most $\epsilon B$.

We begin by calling the multicalibration theorem of [32, 26] with $\mathcal{X}, \mathcal{D}, \epsilon$, and $\hat{C}$ to obtain a $(\hat{\mathcal{C}}, \epsilon)$-multicalibrated predictor $h$. For any fixed loss function $\ell = (\ell_+, \ell_-, \ell_?) \in \mathcal{L}$, we claim that $k^*_{\ell_\pm, \ell_?} \circ h$, where $k^*_{\ell_\pm, \ell_?}$ is the post-processing function defined in Definition 3.1, is a selective omnipredictor.

By the definition of a selective omnipredictor, we want to show that generalized Chow loss incurred by $k^*_{\ell_\pm, \ell_?} \circ h$ is upper-bounded by the generalized Chow loss incurred by $k \circ c$ for every $c \in \mathcal{C}$, where $k : [0, 1] \to \mathbb{R} \cup \{?\}$ is an arbitrary post-processing function. By definition of $k^*_{\ell_\pm, \ell_?}$, recall that

$$k^*_{\ell_\pm, \ell_?}(p) = \mathrm{argmin}_{t \in \mathbb{R} \cup \{?\}} \, p \cdot \Big(\mu\ell_-(t) + \nu\ell_?(1)\Big) + (1 - p) \cdot \Big(\lambda\ell_+(t) + \nu\ell_?(0)\Big).$$

Following this generalized Chow loss expression (Definition 2.6), we decompose it into the expected cost of predicting, which we denote by $\kappa_{\mathrm{pred}}$, and the expected cost of abstaining, which we denote by $\kappa_{\mathrm{abs}}$, both under the Bernoulli distribution (i.e., for $\mathbf{y} \sim \mathrm{Bern}(p(\mathbf{x}))$ for each $p \in \mathrm{range}(h)$):

$$\kappa_{\mathrm{pred}}(p) = \min_t p \cdot \mu\ell_-(t) + (1 - p) \cdot \lambda\ell_+(t),$$

$$\kappa_{\mathrm{abs}}(p) = \nu\big(p \cdot \ell_?(1) + (1 - p) \cdot \ell_?(0)\big) = \nu\big(p \cdot \alpha_+ + (1 - p) \cdot \alpha_-\big).$$

Note that the prediction cost depends on the value of $t$, whereas the abstention cost is independent of $t$. Then, for each point $p$, in order to minimize the expected generalized Chow loss under the Bernoulli distribution $\mathrm{Bern}(p(\mathbf{x}))$, we proceed in two steps:

- Find the value $t^*(p) \in [-M, M]$ that minimizes the value of $\kappa_{\mathrm{pred}}(p)$ given a fixed value of $p$. This depends only on the choice of $\mu, \ell_-, \lambda, \ell_+$ and not on the underlying data. This corresponds to finding the $t^*(p)$ value that minimizes the value of $k^*_\ell(p)$.

- For each predicted value $h(x) = p$, we map it to either $t^*(p)$ or ? depending on whether it is cheaper to predict or to abstain; that is, depending on the value of $\min\{\kappa_{\mathrm{pred}}(t^*(p)), \kappa_{\mathrm{abs}}\}$.

In other words, we can re-write $k^*_{\ell_\pm, \ell_?}$ as $k_{\mathrm{abs}} \circ k^*_\ell$, where

$$k_{\mathrm{abs}} = \begin{cases} t^* & \text{if } \kappa_{\mathrm{pred}} \leq \kappa_{\mathrm{abs}} \\ ? & \text{if } \kappa_{\mathrm{pred}} > \kappa_{\mathrm{abs}} \end{cases}$$

The key idea is the following: the true labels are distributed as $\mathbf{y} \sim \mathrm{Bern}(p^*(\mathbf{x}))$. So if we had access to the true $p^*(x)$ value for each $x$, then the optimal prediction/abstention decision rule (i.e., the post-processing function applied to the $p^*$ value that yields the minimum total generalized Chow loss) is precisely $k_{\mathrm{abs}}$.

We can write the cost function incurred by the post-processing function $k^*_{\ell_\pm, \ell_?} = k_{\mathrm{abs}} \circ k^*_\ell$ as:

$$\kappa(t, p) = \min\{p \cdot \mu\ell_-(t) + (1 - p) \cdot \lambda\ell_+(t), p \cdot \ell_?(1) + (1 - p) \cdot \ell_?(0)\}.$$

An important point to remark is that, while the prediction/abstention decision rule is decided with the $p$-values (to which we have access to, since they correspond to the predictions of the multicalibrated predictor $h$), the actual cost that we incur is computed with the true values $p^*$. Multicalibration

precisely allows us to bridge this gap: the predictor $h$ believes that the labels are distributed according to $\mathrm{Bern}(p(\mathbf{x}))$, and so it uses the function $k_{\mathrm{abs}} \circ k_{\ell}^*$ as post-processing, which yields the optimal cost under the distribution $\mathbf{y} \sim \mathrm{Bern}(p(\mathbf{x}))$. In order to bridge the "simulated" labels $\mathbf{y} \sim \mathrm{Bern}(p(\mathbf{x}))$, which are used in our decision rule, and the "true" labels $\mathbf{y} \sim \mathrm{Bern}(p^*(\mathbf{x}))$, which yield the actual cost that we pay, we use the fact that $h$ is $\mathcal{C}$-multicalibrated, which ensures that

$$\mathbb{E}[\mathbf{y} \mid h = p, \hat{c} = \gamma] \approx \mathbb{E}[p \mid h = p, c = \gamma]$$

for each $\gamma$ in the range of $\hat{c}$. The RHS can equivalently be written as $\mathbb{E}[h \mid h = p, \hat{c} = \gamma]$. This "bridging" enabled by the multicalibration property satisfied by the predictor $h$ is what allows us to show that our selective omnipredictor (namely, the predictor $k_{\ell_\pm, \ell_?}^*$) incurs optimal loss with respect to the class $\hat{C}$.

We have discussed how the prediction/abstention decision rule for the multicalibrated predictor is given by $\kappa(t^*(p), p)$. As per the definition of a selective omnipredictor, we need to show that the generalized Chow loss incurred by $k_{\ell_\pm, \ell_?}^*$ is upper-bounded by the generalized Chow loss incurred by any of the selective concepts $c_?$, where $c \in \mathcal{C}$ and $k : [0, 1] \to \mathbb{R} \cup \{?\}$ is *any* post-processing function that adds abstentions to the concepts $c \in \mathcal{C}$. The only natural restriction on $k$ is that it is a function of the values $\hat{c}(\cdot)$, and cannot be a function of $x$. E.g., if $\hat{c}(x_1) = \gamma = \hat{c}(x_2)$, then it must be that $\hat{c}_?(x_1) = \hat{c}_?(x_2)$. Hence, in the case of the concepts $\hat{c} \in \hat{C}$, we cannot directly assume that the prediction/abstention decision rule corresponds to $k_{\ell_\pm, \ell_?}^*$ as well.

However, we can use the multicalibration condition to reason about the loss incurred by the concepts $\hat{c}$. Specifically, for each concept $\hat{c} \in \hat{\mathcal{C}}$, we define the sets $\mathcal{X}_{(p,\gamma)} = \{x \in \mathcal{X} \mid h(x) = p, \hat{c}(x) = \gamma\}$ for each $p \in \mathrm{range}(h)$ and each $\gamma \in \mathrm{range}(\hat{c})$. Moreover, we let

$$\phi_{(p,\gamma)} = \mathbb{E}[\mathbf{y} | \mathbf{x} \in \mathcal{X}_{(p,\gamma)}].$$

The fact that $h$ is $(\hat{C}, \epsilon)$-multicalibrated implies that

$$\phi_{(p,\gamma)} = \mathbb{E}[\mathbf{y} \mid h(\mathbf{x}) = p, \hat{c}(\mathbf{x}) = \gamma] \approx_\epsilon \mathbb{E}[h(\mathbf{x}) \mid h(\mathbf{x}) = p, \hat{c}(\mathbf{x}) = \gamma] = p \implies \phi_{(p,\gamma)} \approx_\epsilon p.$$

In practice, the multicalibration condition applies on expectation over the level sets $\mathcal{X}_{(p,\gamma)}$ for all $p, \gamma$. We fix a level set $h(x) = p$ of $h$ and a concept $\hat{c} \in \hat{C}$. We want to compare the loss incurred by $k_{\ell_\pm, \ell_?}^*(h)$ with the loss incurred by $k \circ c$, where $k : [0, 1] \to \mathbb{R} \cup \{?\}$ is any post-processing function. Within the level set $h(x) = p$, all of the values $k_{\ell_\pm, \ell_?}^*(h)$ are the same, and so $k_{\ell_\pm, \ell_?}^*(h)$ is either predicting the value $t^*(p)$ on all of the points in the level set $h(x) = p$, or abstaining in all of the points in the level set, as determined by the cost function $\kappa(t^*(p), p)$. Within the level set $h(x) = p$, we further partition it according to the level sets of $\hat{c}$. That is, we consider the partition of $\mathcal{X}_p$ into the sets $\mathcal{X}_{(p,\gamma)}$ for each $\gamma \in \mathrm{range}(\hat{c})$. For each $x \in \mathcal{X}_{(p,\gamma)}$, $\hat{c}$ either predicts or abstains, using a decision rule $k$ that is allowed to be arbitrary. For each set $\mathcal{X}_{(p,\gamma)}$, we split the proof into 4 cases, depending on whether $h$ and $\hat{c}$ decide to abstain or predict as per their respective decision rules.

Throughout, we let $\beta$ denote a bound on the absolute values of $\ell_+, \ell_-, \ell_?$. Moreover, we can write the loss function as

$$\ell_{\mathrm{GC}, \mathcal{D}}(k_{\ell_\pm, \ell_?}^* \circ h; \lambda, \mu, \nu) = \underset{p \in \mathrm{range}(h)}{\mathbb{E}} \; \underset{\gamma \in \mathrm{range}(\hat{c})}{\mathbb{E}} \; \underset{\mathbf{x} \sim \mathcal{D}|_{\mathcal{X}_{(p,\gamma)}}}{\mathbb{E}} [\ell_{\mathrm{GC}, \mathcal{D}}(\mathbf{y}, k_{\ell_\pm, \ell_?}^*(h(\mathbf{x})))],$$

and similarly for $k \circ \hat{c}$.

Having fixed the values $h = p$ and $\hat{c} = \gamma$, we argue about the expected loss incurred by the post-processed multicalibrated predictor versus the expected loss incurred by the concept on the level set $\mathcal{X}_{(p,\gamma)}$.

**1.** $k_{\ell_\pm, \ell_?}^* \circ h$ **predicts &** $k \circ \hat{c}$ **predicts.** We begin by swapping $\phi_{(p,\gamma)}$ for $p$ in the following expression:

$$\phi_{(p,\gamma)} \cdot \mu\ell_-(t^*(p)) + (1 - \phi_{(p,\gamma)}) \cdot \lambda\ell_+(t^*(p)) = p \cdot \mu\ell_-(t^*(p)) + (1 - p) \cdot \lambda\ell_+(t^*(p))$$
$$+ (\phi_{(p,\gamma)} - p)(\mu\ell_-(t^*(p)) - \lambda\ell_+(t^*(p))).$$

By definition of $t^*(p)$, it follows that $t^*(p)$ is the minimizer of $\kappa_{\mathrm{pred}}(p)$ in $[-M, M]$ given a fixed value of $p$. Hence,

$$p \cdot \mu\ell_-(t^*(p)) + (1 - p) \cdot \lambda\ell_+(t^*(p)) \leq p \cdot \mu\ell_-(\gamma) + (1 - p) \cdot \lambda\ell_+(\gamma).$$

Swapping $p$ for $\phi_{(p,\gamma)}$ again, we get that

$$p \cdot \mu\ell_-(\gamma) + (1-p) \cdot \lambda\ell_+(\gamma) + (\phi_{(p,\gamma)} - p)(\mu\ell_-(t^*(p)) - \lambda\ell_+(t^*(p)))$$

$$= \phi_{(p,\gamma)} \cdot \mu\ell_-(\gamma) + (1-\phi_{(p,\gamma)}) \cdot \lambda\ell_+(\gamma) + (\phi_{(p,\gamma)} - p)\left[\left(\mu\ell_-(t^*(p)) - \lambda\ell_+(t^*(p))\right) - \left(\mu\ell_-(\gamma) - \lambda\ell_+(\gamma)\right)\right].$$

Putting everything together, we get that

$$\phi_{(p,\gamma)} \cdot \mu\ell_-(t^*(p)) + (1-\phi_{(p,\gamma)}) \cdot \lambda\ell_+(t^*(p))$$

$$= \phi_{(p,\gamma)} \cdot \mu\ell_-(\gamma) + (1-\phi_{(p,\gamma)}) \cdot \lambda\ell_+(\gamma) + (\phi_{(p,\gamma)} - p)\left[\left(\mu\ell_-(t^*(p)) - \ell\lambda\ell_+(t^*(p))\right) - \left(\mu\ell_-(\gamma) - \lambda\ell_+(\gamma)\right)\right].$$

By the $\beta$-bound on the loss functions, and given that $\lambda + \mu + \nu \leq 1$, it follows that

$$(\phi_{(p,\gamma)} - p)\left[\left(\mu\ell_-(t^*(p)) - \ell\lambda\ell_+(t^*(p))\right) - \left(\mu\ell_-(\gamma) - \lambda\ell_+(\gamma)\right)\right] \leq |\phi_{(p,\gamma)} - p| \cdot 4\beta.$$

Therefore, over $\mathcal{X}_{(p,\gamma)}$, where $k^*_{\ell_\pm,\ell_?} \circ h$ is predicting $p$ and $k \circ \hat{c}$ is predicting $\gamma$, the expected generalized Chow losses compare as follows:

$$\ell_{\mathrm{GC},\mathcal{D}}(k^*_{\ell_\pm,\ell_?} \circ h; \lambda, \mu, \nu \mid \mathbf{x} \in \mathcal{X}_{(p,\gamma)}) \leq \ell_{\mathrm{GC},\mathcal{D}}(k \circ \hat{c}; \lambda, \mu, \nu \mid \mathbf{x} \in \mathcal{X}_{(p,\gamma)}) + |\phi_{(p,\gamma)} - p| \cdot 4\beta.$$

**2. $k^*_{\ell_\pm,\ell_?} \circ h$ predicts & $k \circ \hat{c}$ abstains.** As in the previous case, we swap $\rho_{(p,\gamma)}$ by p:

$$\phi_{(p,\gamma)} \cdot \mu\ell_-(t^*(p)) + (1-\phi_{(p,\gamma)}) \cdot \lambda\ell_+(t^*(p)) = p \cdot \mu\ell_-(t^*(p)) + (1-p) \cdot \lambda\ell_+(t^*(p))$$
$$+ (\phi_{(p,\gamma)} - p)(\mu\ell_-(t^*(p)) - \lambda\ell_+(t^*(p))).$$

By the decision rule for $h$ determined by the value of $\kappa(t^*(p), p)$, if $k^*_{\ell_\pm,\ell_?} \circ h$ predicts on $X_{(p,\gamma)}$ this implies that

$$p \cdot \mu\ell_-(t^*(p)) + (1-p) \cdot \lambda\ell_+(t^*(p)) \leq p \cdot \nu\ell_?(1) + (1-p) \cdot \nu\ell_?(0).$$

Again swapping $p$ for $\phi_{(p,\gamma)}$, we obtain:

$$p \cdot \nu\ell_?(1) + (1-p) \cdot \nu\ell_?(0) + (\phi_{(p,\gamma)} - p)(\mu\ell_-(t^*(p)) - \lambda\ell_+(t^*(p)))$$

$$= \phi_{(p,\gamma)} \cdot \nu\ell_?(1) + (1-\phi_{(p,\gamma)}) \cdot \nu\ell_?(0) + (\phi_{(p,\gamma)} - p)\left[\left(\mu\ell_-(t^*(p)) - \lambda\ell_+(t^*(p))\right) - \left(\nu\ell_?(1) - \nu\ell_?(0)\right)\right].$$

By the $\beta$-bound on the loss functions, and given that $\lambda + \mu + \nu \leq 1$, it follows that

$$(\phi_{(p,\gamma)} - p)\left[\left(\mu\ell_-(t^*(p)) - \lambda\ell_+(t^*(p))\right) - \left(\nu\ell_?(1) - \nu\ell_?(0)\right)\right] \leq |\phi_{(p,\gamma)} - p| \cdot 4\beta.$$

Therefore, putting everything together, we obtain that

$$\ell_{\mathrm{GC},\mathcal{D}}(k^*_{\ell_\pm,\ell_?} \circ h; \lambda, \mu, \nu \mid \mathbf{x} \in \mathcal{X}_{(p,\gamma)}) \leq \ell_{\mathrm{GC},\mathcal{D}}(k \circ \hat{c}; \lambda, \mu, \nu \mid \mathbf{x} \in \mathcal{X}_{(p,\gamma)}) + |\phi_{(p,\gamma)} - p| \cdot 4\beta.$$

**3. $k^*_{\ell_\pm,\ell_?} \circ h$ abstains & $k \circ \hat{c}$ predicts.** We start by swapping $\rho$ for $p$ as in the previous two cases:

$$\phi_{(p,\gamma)} \cdot \nu\ell_?(1) + (1-\phi_{(p,\gamma)}) \cdot \nu\ell_?(0) = p \cdot \nu\ell_?(1) + (1-p) \cdot \nu\ell_?(0) + (\phi_{(p,\gamma)} - p)(\nu\ell_?(1) - \nu\ell_?(0)).$$

Because $k^*_{\ell_\pm,\ell_?} \circ h$ abstains on this level set, it must be that

$$p \cdot \nu\ell_?(1) + (1-p) \cdot \nu\ell_?(0) \leq p \cdot \mu\ell_-(t^*(p)) + (1-p) \cdot \lambda\ell_+(t^*(p)).$$

By definition of $t^*(p)$ as the minimizer of $\kappa_{\mathrm{pred}}$, for any value of $\gamma$ we have that

$$p \cdot \mu\ell_-(t^*(p)) + (1-p) \cdot \lambda\ell_+(t^*(p)) \leq p \cdot \mu\ell_-(\gamma) + (1-p) \cdot \lambda\ell_+(\gamma).$$

We again switch $p$ back to $\phi_{(p,\gamma)}$:

$$p \cdot \mu\ell_-(\gamma) + (1-p) \cdot \lambda\ell_+(\gamma) + (\phi_{(p,\gamma)} - p)(\nu\ell_?(1) - \nu\ell_?(0))$$

$$= \phi_{(p,\gamma)} \cdot \mu\ell_-(\gamma) + (1-\phi_{(p,\gamma)})) \cdot \lambda\ell_+(\gamma) + (\phi_{(p,\gamma)} - p)\left[\left(\nu\ell_?(1) - \nu\ell_?(0)\right) - \left(\mu\ell_-(\gamma) + \lambda\ell_+(\gamma)\right)\right]$$

By the $\beta$-bound on the loss functions, and given that $\lambda + \mu + \nu \leq 1$, it follows that

$$(\phi_{(p,\gamma)} - p)\left[\left(\nu\ell_?(1) - \nu\ell_?(0)\right) - \left(\mu\ell_-(\gamma) + \lambda\ell_+(\gamma)\right)\right] \leq |\phi_{(p,\gamma)} - p| \cdot 4\beta.$$

Putting everything together, we obtain that

$$\ell_{\text{GC},\mathcal{D}}(k^*_{\ell_\pm,\ell_?} \circ h; \lambda, \mu, \nu \mid \mathbf{x} \in \mathcal{X}_{(p,\gamma)}) \leq \ell_{\text{GC},\mathcal{D}}(k \circ \hat{c}; \lambda, \mu, \nu \mid \mathbf{x} \in \mathcal{X}_{(p,\gamma)}) + |\phi_{(p,\gamma)} - p| \cdot 4\beta.$$

**4.** $k^*_{\ell_\pm,\ell_?} \circ h$ **abstains &** $k \circ \hat{c}$ **abstains.** In this case, given that the values of $\ell_?(1)$ and $\ell_?(0)$ are independent of $t$, it directly follows that both $k^*_{\ell_\pm,\ell_?} \circ h$ and $k \circ \hat{c}$ incur the exact same generalized Chow loss on $\mathcal{X}_{(p,\gamma)}$.

Putting these four cases together, and by taking the expected value over all level sets $\mathcal{X}_{(p,\gamma)}$, for all $p$ in the range of $h$ and $\gamma$ in the range of $\hat{c}$, and by thus applying the multicalibration guarantee on $\mathbb{E}[|\phi_{(p,\gamma)} - p|]$ (i.e., which guarantees that $\mathbb{E}[|\phi_{(p,\gamma)} - p|] \leq \epsilon$), we obtain that

$$\ell_{\text{GC},\mathcal{D}}(k^*_{\ell_\pm,\ell_?} \circ h; \lambda, \mu, \nu) \leq \ell_{\text{GC},\mathcal{D}}(k \circ c; \lambda, \mu, \nu) + 4\epsilon\beta.$$

Because these four cases are exhaustive and hold for all values of $p$ and $\gamma$, we conclude that

$$\ell_{\text{GC},\mathcal{D}}(k^*_{\ell_\pm,\ell_?} \circ h; \lambda, \mu, \nu) = \underset{p \in \text{range}(h)}{\mathbb{E}} \underset{\gamma \in \text{range}(\hat{c})}{\mathbb{E}} \underset{\mathbf{x} \sim \mathcal{D}|_{\mathcal{X}_{(\phi,\gamma)}}}{\mathbb{E}} [\ell_{\text{GC},\mathcal{D}}(\mathbf{y}, k^*_{\ell_\pm,\ell_?}(h(\mathbf{x})))]$$

$$\leq \underset{p \in \text{range}(h)}{\mathbb{E}} \underset{\gamma \in \text{range}(\hat{c})}{\mathbb{E}} \underset{\mathbf{x} \sim \mathcal{D}|_{\mathcal{X}_{(\phi,\gamma)}}}{\mathbb{E}} [\ell_{\text{GC},\mathcal{D}}(\mathbf{y}, k(c(\mathbf{x})))] + 4\epsilon\beta$$

$$= \ell_{\text{GC},\mathcal{D}}(k \circ c; \lambda, \mu, \nu) + 4\epsilon\beta.$$

Therefore, for the non-discretized concept class $\mathcal{C}$, and accounting for $\epsilon B$ loss incurred in the clippings of each of $c$ and $t^*(p)$, we conclude that

$$\ell_{\text{GC},\mathcal{D}}(k^*_{\ell_\pm,\ell_?} \circ h; \lambda, \mu, \nu) \leq \ell_{\text{GC},\mathcal{D}}(k \circ c; \lambda, \mu, \nu) + 4\epsilon\beta + \epsilon B$$

for all loss functions in $\mathcal{L}$, and hence $h$ is a selective omnipredictor, as we wanted to show.

Then, the value of $k^*_{\ell_\pm,\ell_?}$ can be computed efficiently because (1) computing the value of $\kappa_{\text{pred}}(p)$ corresponds to solving a one-dimensional minimization problem, and (2) the value of $k^*_{\ell_\pm,\ell_?}$ is then fully determined from the values $\kappa_{\text{pred}}(p)$ with the optimal $t = t^*(p)$ and of $\kappa_{\text{abs}}$, independent from the data.

Lastly, we show that allowing for randomized selective predictors does not help in minimizing the generalized Chow loss function. Suppose that the selective predictor was randomized, such that for each point $x \in \mathcal{X}$ it would predict a value $a \in [0,1]$ indicating the probability of abstention on $x$. Then, for a fixed $t$ we can write our post-processing function as

$$k^*_{\ell_\pm,\ell_?}(p) = \text{argmin}_{t,a}(1-a)\kappa_{\text{pred}} + a\kappa_{\text{abs}} = (\kappa_{\text{abs}} - \kappa_{\text{pred}}) \cdot a + \kappa_{\text{pred}}.$$

For a fixed $t \in \mathbb{R} \cup \{?\}$ (and once the loss functions have been fixed), the total loss only depends on $\kappa_{\text{pred}}$ and $\kappa_{\text{abs}}$, which in turn only depend on $p$. Note that after fixing $t$, $k^*_{\ell,a}$ is a linear function on $\kappa_{\text{pred}}, \kappa_{\text{abs}}$. This implies that the total generalized Chow loss is minimized at either $a = 0$ or $a = 1$, so no fractional abstention is required. $\qquad\square$

**Remark C.2.** If the concepts $c \in \mathcal{C}$ are not bounded a priori, then we can clip them into an interval $[-M, M]$ for some finite value $M$. Given any convex loss function $\ell$ and parameter $\epsilon$, we find $-M$ and $M$ by determining the values of $t \in \mathbb{R}$ such that $\ell(0, t) = \epsilon$ and $\ell(1, t) = \epsilon$, respectively; these correspond to the values of $-M$ and $M$. This clipping is either helpful (prevents the loss from getting too large) or incurs at most $\epsilon$ additional loss; for example, for the logistic loss, we set $M = \log(1/\epsilon), -M = -\log(1/\epsilon)$.

**Interval of abstention.** Having shown that we can efficiently build selective omnipredictors, we further show that all points $p$ that are set to ? by $k^*_{\ell_\pm,\ell_?}$ are in a contiguous interval. This follows from the affinity of the loss functions $\ell_+, \ell_- \in \mathcal{L}$.

We can also write $\kappa_{\text{pred}}(p)$ as $\kappa_{\text{pred}}(p, t^*)$ to indicate that the value of $t^*$ has been set to the optimal value for each prediction $p$. More generally, we let $\kappa_{\text{pred}}(p, t) = p \cdot \mu\ell_-(t) + (1-p) \cdot \lambda\ell_+(t)$.

**Lemma 3.2.** *Given any loss functions* $\ell_+, \ell_-, \ell_? \in \mathcal{L}$, *the points* $x \in \mathbb{R}$ *such that* $k^*_{\ell_\pm,\ell_?}(x) = ?$ *form a contiguous interval, which we denote by* $I_{\text{abs}}$.

To prove Lemma 3.2, we first show the following intermediate lemma:

**Lemma C.1.** *The function $\kappa_{\mathrm{pred}}(p, t^*)$ is concave as a function of $p$.*

*Proof.* Recall that $\kappa_{\mathrm{pred}}(p, t^*) = \min_t p \cdot \mu\ell_-(t) + (1 - p) \cdot \lambda\ell_+(t)$, where $\kappa_{\mathrm{pred}}(p, t^*) = \min_{t \in \mathbb{R}} \kappa_{\mathrm{pred}}(p, t)$.[1] For every fixed $t_0 \in \mathbb{R}$, the function $\kappa_{\mathrm{pred}}(p, t_0)$ is affine in $p$:

$$\kappa_{\mathrm{pred}}(p, t_0) = \big(\mu\ell_-(t_0) - \lambda\ell_+(t_0)\big) \cdot p + \lambda\ell_+(t_0).$$

Affine functions are convex and concave, and so $\kappa_{\mathrm{pred}}(p) = \min_{t_0 \in \mathbb{R}} \kappa_{\mathrm{pred}}(p, t_0)$ is equal to the pointwise infimum of a family of affine functions in $p$. It is a known fact in analysis that the pointwise infimum of affine functions is concave, and so $\kappa_{\mathrm{pred}}(p, t^*)$ is indeed concave as a function of $p$. $\square$

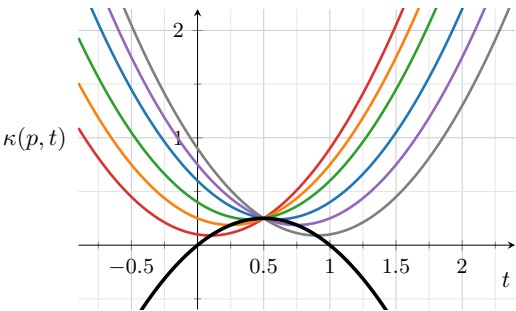 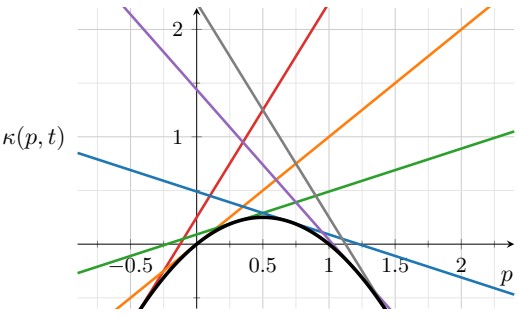

(a) Example with the $\ell_2$ function; $\kappa_{\mathrm{pred}}(p, t)$ with fixed values of $p$. For each fixed value of $p$, the resulting function is convex in $t$.

(b) Example with the $\ell_2$ function; $\kappa_{\mathrm{pred}}(p, t)$ with fixed values of $t$. For each fixed value of $t$, the resulting function is affine in $p$.

Note that for our results on $I_{\mathrm{abs}}$ we only need affinity in $p$.

*Proof of Lemma 3.2.* Recall that $k^*_{\ell_\pm, \ell_?}(p) = \min\{\kappa_{\mathrm{pred}}(p, t^*), \kappa_{\mathrm{abs}}(p)\}$.[2] Per Lemma C.1, the function $\kappa_{\mathrm{pred}}(p, t^*)$ is concave in $p$. By definition, note that $\kappa_{\mathrm{abs}}(p)$ is affine in $p$:

$$\kappa_{\mathrm{abs}}(p) = \big(\nu\alpha_+ - \alpha_-\big)p + \alpha_-.$$

Therefore, the function $\kappa_{\mathrm{pred}}(p, t^*) - \kappa_{\mathrm{abs}}(p)$ is still concave in $p$. Hence this function has at most two roots, and hence the set of points $p$ where $\kappa_{\mathrm{pred}}(p, t^*) \geq \kappa_{\mathrm{abs}}(p)$ forms a contiguous interval (which can be empty, in the case where it is always better to predict than to abstain). By the definition of our post-processing function $k^*_{\ell_\pm, \ell_?}(p)$, this interval corresponds precisely to the set of points where $k^*_{\ell_\pm, \ell_?}(p) = ?$, and hence this interval is equal to $I_{\mathrm{abs}}$. $\square$

Figure 2b illustrates the concavity of $\kappa_{\mathrm{pred}}(p, t)$ as a function of $t$ (left) and the affinity of $\kappa_{\mathrm{pred}}(p, t)$ as a function of $p$ (right), which prove that the abstentions as allocated by the selective omnipredictor occur in one contiguous (and possibly empty) interval $I_{\mathrm{abs}}$.

**Omniprediction with constraints.** Recent work by Hu, Livni-Navon, Reingold, and Yang extends the line of work on omniprediction to constrained optimization problems [35]. This allows the learner to train agnostic to the final choice of loss function as well as of constraints that will be later imposed (as long as these satisfy certain conditions). By viewing Condition 1 in the definitions of PRL, NRL, and FRL as a constraint, one could potentially adapt their results in order to construct $\mathcal{L}$-PRL and $\mathcal{L}$-NRL predictors, and then ensemble them in the usual way to obtain $\mathcal{L}$-FRL predictors. It is unclear how this approach could be used to obtain optimality in the more general framework of selective omniprediction with generalized Chow losses (Definition 2.7).

---

[1]In the proof of Theorem 3.1 we used $\kappa_{\mathrm{pred}}(p)$. Here we write $t^*$ to remind that the function has been minimized with respect to $t$.

[2]In the case of ties, we decide to predict.

## C.2 FRL from selective omniprediction

**Theorem 3.3.** *Let $\mathcal{C}$ be a concept class of Boolean concepts, $\mathcal{L}$ any class of B-Lipschitz loss functions, and $\mathcal{D}$ a distribution on $\mathcal{X} \times \mathcal{Y}$. If $\mathcal{C}$ is agnostically learnable under $\mathcal{D}$ in time $T(\epsilon, \delta)$, then $\mathcal{C}$ is $\mathcal{L}$-fully reliably learnable under $\mathcal{D}$ in time $T(\epsilon^2/6, \delta)$.*

Note that here we use our proposed notion of $\mathcal{L}$-FRL (Definition 2.5), which is a generalization of the 0-1 version proposed in [43].

*Proof.* Let $\epsilon$ be the target error parameter for fully reliable learning. We consider the parameters $\lambda = \mu = 1/3$ and $\nu = \epsilon/6$. From $\mathcal{C}$, we define the concept class $\mathcal{C}'$ of concepts $c' : \mathcal{X} \to \{0, 1/2, 1\}$ as containing the concepts $(c_1 + c_2)/2$ for every possible pairing $c_1, c_2 \in \mathcal{C}$. Defining the post-processing $k \circ \mathcal{C}'$ as mapping 0 to 0, 1 to 1, and $1/2$ to ?, we get that $\mathrm{SC}(\mathcal{C}) \subseteq k \circ \mathcal{C}'$. Let $c_?^*$ be an optimal abstaining classifier in $\mathrm{SC}(\mathcal{C})$. By definition of $\mathrm{SC}(\mathcal{C})$, it follows that all concepts in $\mathrm{SC}(\mathcal{C})$ incur 0 error over the non-abstaining region, and hence

$$\ell_{\mathrm{GC},\mathcal{D}}(c_?^*; \lambda, \mu, \nu) = \nu \ell_?(c_?^*).$$

Let $h$ be a $(\mathcal{L}, \mathcal{C}', \gamma)$-selective omnipredictor (which we can obtain for $\mathcal{C}'$ given that $\mathcal{C}'$ is agnostically learnable, as shown in Theorem 3.1). By the selective omniprediction guarantee, it follows that for any post-processing function $k : [0, 1] \to \mathbb{R} \cup \{?\}$,

$$\ell_{\mathrm{GC},\mathcal{D}}(h; \lambda, \mu, \nu) \leq \min_{c_?' \in k \circ \mathcal{C}} \ell_{\mathrm{GC},\mathcal{D}}(c_?'; \lambda, \mu, \nu) + \gamma.$$

In particular, using the post-processing $k$ that maps 0 to 0, 1 to 1, and $1/2$ to ?, this implies that

$$\ell_{\mathrm{GC},\mathcal{D}}(h; \lambda, \mu, \nu) \leq \ell_{\mathrm{GC},\mathcal{D}}(c_?^*; \lambda, \mu, \nu) + \epsilon = \nu \ell_?(c_?^*) + \gamma.$$

By setting $\gamma = \epsilon^2/6$, and using the fact that $\ell_?(c_?^*) \leq 1$, we get and since $\lambda = 1/3$, $\nu = \epsilon/6$,

$$\frac{1}{3}\ell_+(h) \leq \epsilon/6 + \epsilon^2/6,$$

and so $\ell_+(h) \leq \epsilon$. Similarly, we have that $\ell_-(h) \leq \epsilon$. Finally,

$$\gamma \ell_?(h) \leq \gamma \ell_?(c_?^*) + \epsilon^2/6,$$

since $\gamma = \epsilon/6$, it follows that,

$$\ell_?(h) \leq \ell_?(c_?^*) + \epsilon.$$

This completes the proof. □

## C.3 Conformal prediction from FRL

**Lemma 3.4.** *Let $h_? : \mathcal{X} \to \{0, 1, ?\}$ be an FRL predictor for $\mathcal{C}, \mathcal{D}, \epsilon$ and the 0-1 loss. Then, the prediction sets induced by the level sets of $h$, where we map $0 \mapsto \{0\}$, $1 \mapsto \{1\}$, and $? \mapsto \{0, 1\}$, satisfy the $\epsilon$-marginal coverage guarantee.*

*Proof.* Let $\mathrm{err}(h_?, \mathcal{D}) = \Pr_{(\mathbf{x},\mathbf{y})\sim\mathcal{D}}[h_?(\mathbf{x}) \neq \mathbf{y}]$ denote the error of the selective predictor $h_?$ and recall that

$$\mathrm{false}_+(h_?, \mathcal{D}) = \Pr_{(\mathbf{x},\mathbf{y})\sim\mathcal{D}}[h_?(\mathbf{x}) = 1 \wedge \mathbf{y} = 0] \quad \mathrm{false}_-(h_?, \mathcal{D}) = \Pr_{(\mathbf{x},\mathbf{y})\sim\mathcal{D}}[h_?(\mathbf{x}) = 0 \wedge \mathbf{y} = 1].$$

Since $\mathrm{err}(h_?, \mathcal{D}) \leq \epsilon$ by the FRL guarantee of $h_?$ and $\mathrm{err}(h_?, \mathcal{D}) = \mathrm{false}_+(h_?, \mathcal{D}) + \mathrm{false}_-(h_?, \mathcal{D})$, it follows that

$$\mathrm{false}_+(h_?, \mathcal{D}) \leq \epsilon, \qquad \mathrm{false}_-(h_?, \mathcal{D}) \leq \epsilon.$$

This implies that for all $\mathbf{y} = 1$,

$$\Pr_{(\mathbf{x},\mathbf{y})\sim\mathcal{D}}[\mathbf{y} \notin \{1\}] \leq \epsilon,$$

and symmetrically for all $\mathbf{y} = 0$. Therefore, we satisfy the marginal coverage guarantee for the prediction sets $\{0\}, \{1\}$. We also trivially satisfy it for the prediction set $\{0, 1\}$, since $\mathbf{y}$ is Boolean and so $\Pr_{(\mathbf{x},\mathbf{y})\sim\mathcal{D}}[\mathbf{y} \in \{0, 1\}] = 1$. □

Lastly, we formalize and prove Remark 3.1, which shows that FRL viewed as a conformal prediction method, besides satisfying the $\epsilon$-marginal coverage guarantee, also provides a provable upper bound on the size of the set $\{0, 1\}$, which constitutes a measure of uncertainty:

**Lemma C.2.** *Let $h_? : \mathcal{X} \to \{0, 1, ?\}$ be an FRL predictor for $\mathcal{C}, \mathcal{D}, \epsilon$, and the 0-1 loss. Then, the prediction sets induced by the level sets of $h_?$ satisfy:*

$$\Pr_{(\mathbf{x},\mathbf{y}) \sim \mathcal{D}}[h_?(\mathbf{x}) = \{0, 1\}] \leq \min_{c_? \in \mathtt{SC}(\mathcal{C})} \Pr_{(\mathbf{x},\mathbf{y}) \sim \mathcal{D}}[c_?(x) = ?] + \epsilon.$$

*Proof.* This follows directly from Condition 2 in the definition of FRL, which ensures that

$$?(h_?, \mathcal{D}) \leq \min_{c_? \in \mathtt{SC}(\mathcal{C})} ?(c_?, \mathcal{D}) + \epsilon.$$

$\square$

# D    Examples of Selective Omnipredictors

In this section, we provide concrete examples of the post-processing function $k^*_{\ell_\pm, \ell_?}$ for specific choices of loss function triplets $(\ell_+, \ell_-, \ell_?)$. We do so for the triplets of loss functions used in Figure 1, and for each triplet we theoretically derive the abstention interval $I_{\mathrm{abs}}$. Recall that $I_{\mathrm{abs}}$ only depends on the chosen generalized Chow loss, and not on the underlying data, which is what makes our selective omniprediction framework extremely efficient to adapt to many different loss functions.

$\ell_1$ **loss with different abstention costs.**    Consider letting $\ell_-, \ell_+$ correspond to the 0-1 loss, and let the abstention cost be fixed at some value $\alpha > 0$. That is:

$$\ell_-(t) = |1 - t|, \quad \ell_+(t) = |t|, \quad \ell_?(y) = \alpha.$$

Then,

$$\kappa_{\mathrm{pred}}(p) = \min_t p \cdot |1 - t| + (1 - p) \cdot |t|, \quad \kappa_{\mathrm{abs}} = \alpha.$$

The value $t^*(p)$ that minimizes $\kappa_{\mathrm{pred}}(p)$ for each $p$ corresponds to $k^*_\ell(p) = t^*(p) = \mathbb{1}[p \geq 1/2]$. Then, for each predicted value $h(x) = p$, $k^*_{\ell_\pm, \ell_?}$ will compare the expected prediction cost under the Bernoulli distribution $\mathbf{y} \sim \mathrm{Bern}(p)$, namely

$$\kappa_{\mathrm{pred}}(p) = p \cdot |1 - t^*(p)| + (1 - p) \cdot |t^*(p)| = p \cdot \left|1 - \mathbb{1}[p \geq 1/2]\right| + (1 - p) \cdot \left|\mathbb{1}[p \geq 1/2]\right|, \quad (1)$$

with the expected cost of abstention $\kappa_{\mathrm{abs}} = \alpha$. If $\kappa_{\mathrm{pred}}(p) \leq \kappa_{\mathrm{abs}}$, then $k^*_{\ell_\pm, \ell_?}(p) = t^*(p)$; otherwise, $k^*_{\ell_\pm, \ell_?}(p) = ?$.

We compute the interval $I_{\mathrm{abs}}$ for different choices of $\alpha$. For an arbitrary value of $\kappa_{\mathrm{abs}} = \alpha$, $I_{\mathrm{abs}}$ corresponds to the interval contained between the roots of the polynomial $\kappa_{\mathrm{pred}}(p) - \kappa_{\mathrm{abs}}$, which is a function in $p$. In the case of $\alpha = 0.1$, the roots of $\kappa_{\mathrm{pred}}(p) - 0.1$ (where $\kappa_{\mathrm{pred}}$ is given in Equation 1) yield $I_{\mathrm{abs}} = [0.1, 0.9]$. For $\alpha = 0.2$, the roots of $\kappa_{\mathrm{pred}}(p) - 0.2$ yield $[0.2, 0.8]$.

Lastly, we consider the case where $\ell_?(1) = \alpha_+ = 0.3$ and $\ell_?(0) = \alpha_- = 0.1$. Here, we have that

$$\kappa_{\mathrm{abs}}(p) = p \cdot \alpha_+ + (1 - p) \cdot \alpha_-.$$

Then, the roots of the polynomial $\kappa_{\mathrm{pred}}(p) - p \cdot \alpha_+ + (1 - p) \cdot \alpha_-$ for $\alpha_+ = 0.3$ and $\alpha_- = 0.1$ yield $I_{\mathrm{abs}} = [0.125, 0.75]$.

$\ell_2$ **loss with different abstention costs.**    In the case of the $\ell_2$ loss, and still with fixed abstention cost $\ell_?(y) = \alpha$, we have that

$$k^*_{\ell_\pm, \ell_?}(p, t) = p \cdot (1 - t)^2 + (1 - p) \cdot (0 - t)^2 + a\alpha.$$

Hence, if we decide to predict, we pay an expected cost of $\kappa_{\mathrm{pred}}(p) = \min_t p \cdot (1 - t)^2 + (1 - p) \cdot (0 - t)^2$. If we abstain, we pay an expected cost $\kappa_{\mathrm{abs}}(p) = \alpha$. Both of these expected costs are under the Bernoulli distribution $\mathbf{y} \sim \mathrm{Bern}(p)$. For a fixed value of $p$, the function $\kappa_{\mathrm{pred}}(p, t)$ is minimized at $t^*(p) = p$, and hence

$$\kappa_{\mathrm{pred}}(p, t^*) = p \cdot (1 - p)^2 + (1 - p) \cdot p^2.$$

Hence, the value of $k^*_{\ell_\pm,\ell_?}(p)$ is fully determined by the quantity $\min\{p \cdot (1-p)^2 + (1-p) \cdot p^2, \alpha\}$.

We can similarly compute the interval $I_{\text{abs}}$ for different choices of $\alpha$ by computing the roots of the polynomial $\kappa_{\text{pred}} - \kappa_{\text{abs}}$. In the case of $\alpha = 0.1$, the roots of $\kappa_{\text{pred}}(p) - 0.1$ yield $I_{\text{abs}} = [\frac{1-\sqrt{0.6}}{2}, \frac{1+\sqrt{0.6}}{2}]$. For $\alpha = 0.2$, the roots of $\kappa_{\text{pred}}(p) - 0.2$ yield $I_{\text{abs}} = [\frac{1-\sqrt{0.2}}{2}, \frac{1+\sqrt{0.2}}{2}]$.

Lastly, we again consider the case where $\ell_?(1) = \alpha_+ = 0.3$ and $\ell_?(0) = \alpha_- = 0.1$. The roots of the polynomial $\kappa_{\text{pred}}(p) - p \cdot \alpha_+ + (1-p) \cdot \alpha_-$ for $\alpha_+ = 0.3$ and $\alpha_- = 0.1$ yield $I_{\text{abs}} = [\frac{4-\sqrt{6}}{10}, \frac{4+\sqrt{6}}{10}]$.

All of these theoretically-derived abstention intervals $I_{\text{abs}}$ can be visualized in our experiments, as summarized in Figure 1. We further provide an illustrative example in Figure 3 to show how the abstention interval $I_{\text{abs}}$ widens when we decrease the cost of abstention from 0.2 to 0.1.

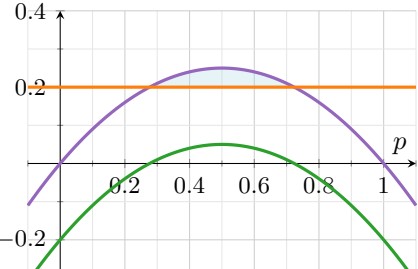 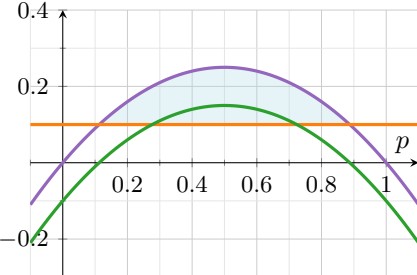

Figure 3: Example with the $\ell_2$ loss and $\ell_?(y) = 0.2$ for all $y$ (i.e., the traditional Chow abstention model) in the left subfigure and $\ell_?(y) = 0.1$ for all $y$ in the right subfigure. The purple line represents $\kappa_{\text{pred}}(p, t^*)$ and the orange line represents $\kappa_{\text{abs}}(p)$. The green line represents $\kappa_{\text{pred}}(p, t^*) - \kappa_{\text{abs}}(p)$, so its roots determine the interval $I_{\text{abs}}$ where we abstain optimally (shaded in blue). In the left example, $I_{\text{abs}} = \big[(1-\sqrt{0.2})/2, (1+\sqrt{0.2})/2\big]$, independent of the data. As we decrease the cost of abstention from $\alpha = 0.2$ to $\alpha = 0.1$ (and so the orange line moves down), our selective omnipredictor obtains an increasingly wider abstention interval $I_{\text{abs}}$ (e.g., in the right subfigure), as one would expect.

# E    Experiments for Section 3

## E.1    Selective omnipredictors

We construct selective omnipredictors from multicalibration to demonstrate the feasibility of our abstention method in practice. We remark that, as far as we are aware, none of the previous works on omniprediction included experimental evaluations. Because our experiments are a proof of concept, we use synthetic data for all of them.[3]

We first provide a full description of the experimental set-up that we used to generate Figure 1 and Table 1 in Section 3.1. Next, we provide further repetitions of our experiment showing selective omnipredictors in action.

First, we generate $10,000$ samples and $20$ features as our data using `sk-learn`'s function `make_classification` and train a random forest to obtain baseline predictions. We then implement the multicalibration algorithm from scratch using the concept class $\mathcal{C}$ of decision trees of depth 3. At each step, we check for correlation between any concept in $\mathcal{C}$ and the residuals computed with the current predictions. For the multicalibration algorithm, we use a discretization parameter of $0.1$, a learning rate of $0.01$, and $200$ maximum iterations. The multicalibration algorithm is run on the validation set ($20\%$ of the data) and we then report all of our statistics on the test set ($20\%$ of the data). This gives us a predictor $h$; note that so far we have not used any loss functions.

Next, we choose specific triplets $(\ell_+, \ell_-, \ell_?)$ and for each we apply our post-processing function $k^*_{\ell_\pm,\ell_?}$. We use the triplets that are shown in Tables 1, 2, 3, 4. This is a pre-computed function that we derive mathematically, so it is extremely efficient to post-process our $h$ in this way. For each triplet, we compute the total coverage, the total loss over the non-abstaining region (i.e., $\ell_+ + \ell_?$), and the

---

[3]The code for this paper can be found at `https://github.com/silviacasac/learning-to-abstain`.

abstention loss (i.e., $\ell_?$). Separately, we do loss-specific minimization to compare with. In order to use the same concept class $\mathcal{C}$, we implement decision trees that minimize the Chow losses for the chosen triplet $(\ell_+, \ell_-, \ell_?)$. That is, we train a different decision tree (that adds abstentions) for each of the different triplets in Tables 1, 2, 3, 4. We similarly report the total coverage, the total loss over the non-abstaining region, and the abstention loss (i.e., $\ell_?$). Further details on our experiments can be found directly in our code.

We repeat all of this process for different initializations of the synthetic data and report several runs in Tables 1, 2, 3, 4. All of them demonstrate the same pattern: our post-processed predictor $h$, even though it is a single predictor for any of the loss triplets, achieves better coverage and better loss than the loss-specific abstaining decision tree. This is because the multicalibration process helps calibrate the predictions towards 0 and 1. In contrast, the decision tree abstains significantly more (and thus obtains lower loss over the non-abstention region). Therefore, this demonstrates the utility of using selective omniprediction in practice.

Table 2: Comparison of coverage and losses. "Cov" stands for *coverage*, "Pred" for *predicted loss* (i.e., over the non-abstaining region), and "Total" for the total generalized Chow loss.

|  | $\ell_? = 0.1$ | | | $\ell_? = 0.2$ | | | $\ell_?(1) = 0.3,\ \ell_?(0) = 0.1$ | |
|  | $k^*_{\ell_\pm,\ell_?} \circ h$ | Abst. DT |  | $k^*_{\ell_\pm,\ell_?} \circ h$ | Abst. DT |  | $k^*_{\ell_\pm,\ell_?} \circ h$ | Abst. DT |
|---|---|---|---|---|---|---|---|---|
| $\ell_1$ | Cov: 45.70% 
 Pred: 0.038 
 Total: 0.072 | Cov: 1.9% 
 Pred: 0.010 
 Total: 0.108 | $\ell_1$ | Cov: 89.15% 
 Pred: 0.069 
 Total: 0.083 | Cov: 4.4% 
 Pred: 0.030 
 Total: 0.221 | $\ell_1$ | Cov: 62.60% 
 Pred: 0.065 
 Total: 0.093 | Cov: 4.8% 
 Pred: 0.005 
 Total: 0.192 |
| $\ell_2$ | Cov: 47.55% 
 Pred: 0.037 
 Total: 0.070 | Cov: 4.4% 
 Pred: 0.003 
 Total: 0.098 | $\ell_2$ | Cov: 89.65% 
 Pred: 0.066 
 Total: 0.079 | Cov: 10.5% 
 Pred: 0.007 
 Total: 0.186 | $\ell_2$ | Cov: 66.65% 
 Pred: 0.057 
 Total: 0.085 | Cov: 11.5% 
 Pred: 0.012 
 Total: 0.181 |

Table 3: Comparison of coverage and losses. "Cov" stands for *coverage*, "Pred" for *predicted loss* (i.e., over the non-abstaining region), and "Total" for the total generalized Chow loss.

|  | $\ell_? = 0.1$ | | | $\ell_? = 0.2$ | | | $\ell_?(1) = 0.3,\ \ell_?(0) = 0.1$ | |
|  | $k^*_{\ell_\pm,\ell_?} \circ h$ | Abst. DT |  | $k^*_{\ell_\pm,\ell_?} \circ h$ | Abst. DT |  | $k^*_{\ell_\pm,\ell_?} \circ h$ | Abst. DT |
|---|---|---|---|---|---|---|---|---|
| $\ell_1$ | Cov: 20.50% 
 Pred: 0.027 
 Total: 0.085 | Cov: 9.5% 
 Pred: 0.032 
 Total: 0.122 | $\ell_1$ | Cov: 43.80% 
 Pred: 0.043 
 Total: 0.131 | Cov: 12% 
 Pred: 0.176 
 Total: 0.211 | $\ell_1$ | Cov: 39% 
 Pred: 0.042 
 Total: 0.136 | Cov: 11.25% 
 Pred: 0.010 
 Total: 0.184 |
| $\ell_2$ | Cov: 22.90% 
 Pred: 0.032 
 Total:0.084 | Cov: 12% 
 Pred: 0.009 
 Total: 0.097 | $\ell_2$ | Cov: 59.20% 
 Pred: 0.064 
 Total: 0.119 | Cov: 20.3% 
 Pred: 0.015 
 Total: 0.174 | $\ell_2$ | Cov: 58.20% 
 Pred: 0.075 
 Total: 0.119 | Cov: 23.6% 
 Pred: 0.014 
 Total: 0.161 |

Table 4: Comparison of coverage and losses. "Cov" stands for *coverage*, "Pred" for *predicted loss* (i.e., over the non-abstaining region), and "Total" for the total generalized Chow loss.

|  | $\ell_? = 0.1$ | | | $\ell_? = 0.2$ | | | $\ell_?(1) = 0.3,\ \ell_?(0) = 0.1$ | |
|  | $k^*_{\ell_\pm,\ell_?} \circ h$ | Abst. DT |  | $k^*_{\ell_\pm,\ell_?} \circ h$ | Abst. DT |  | $k^*_{\ell_\pm,\ell_?} \circ h$ | Abst. DT |
|---|---|---|---|---|---|---|---|---|
| $\ell_1$ | Cov: 48.7% 
 Pred: 0.022 
 Total: 0.062 | Cov: 10.7% 
 Pred: 0.029 
 Total: 0.118 | $\ell_1$ | Cov: 74.15% 
 Pred: 0.043 
 Total: 0.084 | Cov: 19.3% 
 Pred: 0.065 
 Total: 0.226 | $\ell_1$ | Cov: 70.80% 
 Pred: 0.044 
 Total: 0.078 | Cov: 13.15% 
 Pred: 0.008 
 Total: 0.178 |
| $\ell_2$ | Cov: 50.45% 
 Pred: 0.024 
 Total: 0.062 | Cov: 19.3% 
 Pred: 0.005 
 Total: 0.086 | $\ell_2$ | Cov: 79.25% 
 Pred: 0.048 
 Total: 0.079 | Cov: 24.9% 
 Pred: 0.013 
 Total: 0.164 | $\ell_2$ | Cov: 76.85% 
 Pred: 0.048 
 Total: 0.072 | Cov: 24.6% 
 Pred: 0.020 
 Total: 0.164 |

### E.2 Conformal prediction from FRL

As shown in Lemma 3.4 and Remark 3.1 (formalized in Lemma C.2), we can view FRL as a conformal prediction method. We illustrated the feasibility of implementing FRL in practice and compare its

Table 5: Match on $\ell_1$: for each weight $w$, choose $\alpha$ minimizing the difference in $\ell_1$ error between the FRL and the conformal prediction methods.

| Weight | $\alpha$ | FRL MAE | CP MAE | FRL coverage | CP coverage |
|---:|---:|---:|---:|---:|---:|
| 1.0 | 0.25 | 0.250 | 0.260 | 100.0% | 97.0% |
| 5.0 | 0.05 | 0.139 | 0.130 | 54.5% | 42.3% |
| 10.0 | 0.05 | 0.113 | 0.130 | 40.0% | 42.3% |
| 25.0 | 0.01 | 0.078 | 0.027 | 25.5% | 11.0% |
| 50.0 | 0.01 | 0.045 | 0.027 | 17.7% | 11.0% |
| 75.0 | 0.01 | 0.062 | 0.027 | 16.0% | 11.0% |
| 150.0 | 0.01 | 0.015 | 0.027 | 13.1% | 11.0% |
| 200.0 | 0.01 | 0.000 | 0.027 | 9.4% | 11.0% |

coverage and $\ell_1$ error with that of standard conformal prediction algorithms. Similar to our other experiments in the paper, our experiments are a proof of concept for our theoretical results, but do not intend to perform an exhaustive comparison to all of the conformal prediction methods in the literature.

We generate $5,000$ data samples synthetically using sk-learn's make_blobs function, which generates isotropic Gaussian blobs for clustering. To create an FRL predictor, we do it by training a PRL and an NRL predictor separately for the same data, and then ensembling them in the usual way (i.e., if the two agree on a prediction we keep it, otherwise we abstain). To create the PRL and NRL predictors, we use random forests where we give extra weight $w$ to the label that we wish to penalize most for the predictions (so false positives in the case of PRL, and false negatives in the case of NRL). We try weights $w = [1, 5, 10, 25, 50, 75, 150, 200]$. This ensures that our trained predictors achieve the required one-sided guarantees. For the ensembled FRL, we compute its global coverage and $\ell_1$ error.

Next, we use a popular conformal prediction algorithm for binary classification to compare to. We choose the widely-used MAPIE library [65] and use the standard split conformal prediction method using the LAC method to compute the conformity score. We note that this is the only allowed conformity score method in MAPIE for binary classification, which is why we did not add other conformity score methods. We use a random forest as the base class, and train different models for different confidence levels of $\alpha = [0.01, 0.05, 0.10, 0.20, 0.25]$ (i.e., this is the parameter for the marginal coverage guarantee). When tested on the test set, the conformal prediction method returns prediction sets $\{0\}$, $\{1\}$, or $\{0, 1\}$. We view the set $\{0, 1\}$ as equivalent to an abstention ?, and then compute the global coverage and $\ell_1$ error for the conformal prediction method.

To compare both methods for all of the weights $w = [1, 5, 10, 25, 50, 75, 150, 200]$ and coverage parameter $\alpha = [0.01, 0.05, 0.10, 0.20, 0.25]$ we match them based on a) matched $\ell_1$ error, and b) matched coverage. We report the results pairs of tables as follows: a) for each weight $w$, we take the $\alpha$ value that corresponds to the closest $\ell_1$ error of the FRL predictor for this $w$, and we report the errors and coverages for this pair. We then also report the results by matching coverages instead: b) for each weight $w$, we instead take the coverage value that corresponds to the closest coverage to that of the FRL predictor for this $w$, and we report the errors and coverages for this pair.

We do several repeats of our experiment for different initializations of the synthetic data and report the pairs of tables. Tables 5 and 6 correspond to one run, Tables 7 and 8 to another, and 9 and 10 to a last run. We observe the same pattern: FRL provides similar coverage and error guarantees to the standard conformal prediction method, with FRL usually having a higher coverage and lower $\ell_1$ error.

# F Deferred Proofs from Section 4

## F.1 $(\mathcal{C}, \mathcal{G})$-multigroup selective classification

**Theorem 4.1.** *Given access to a $(\mathcal{C} \cdot \mathcal{G}, \epsilon^2/8)$-multiaccurate and calibrated predictor, we can efficiently construct a $(\mathcal{C}, \mathcal{G}, \epsilon)$-multigroup selective classifier in time $\mathrm{poly}(1/\epsilon)$.*

Table 6: Match on coverage: for each weight $w$, choose $\alpha$ minimizing the difference in coverage between the FRL and the conformal prediction methods.

| Weight | $\alpha$ | FRL $\ell_1$ | CP $\ell_1$ | FRL coverage | CP coverage |
|---:|---|---|---|---:|---:|
| 1.0 | 0.25 | 0.250 | 0.260 | 100.0% | 97.0% |
| 5.0 | 0.10 | 0.139 | 0.172 | 54.5% | 62.7% |
| 10.0 | 0.05 | 0.113 | 0.130 | 40.0% | 42.3% |
| 25.0 | 0.01 | 0.078 | 0.027 | 25.5% | 11.0% |
| 50.0 | 0.01 | 0.045 | 0.027 | 17.7% | 11.0% |
| 75.0 | 0.01 | 0.062 | 0.027 | 16.0% | 11.0% |
| 150.0 | 0.01 | 0.015 | 0.027 | 13.1% | 11.0% |
| 200.0 | 0.01 | 0.000 | 0.027 | 9.4% | 11.0% |

Table 7: Match on $\ell_1$: for each weight $w$, choose $\alpha$ minimizing the difference in $\ell_1$ error between the FRL and the conformal prediction methods.

| Weight | $\alpha$ | FRL $\ell_1$ | CP $\ell_1$ | FRL coverage | CP coverage |
|---:|---|---|---|---:|---:|
| 1.0 | 0.25 | 0.226 | 0.223 | 100.0% | 98.5% |
| 5.0 | 0.10 | 0.136 | 0.155 | 56.7% | 64.5% |
| 10.0 | 0.05 | 0.111 | 0.093 | 43.2% | 41.0% |
| 25.0 | 0.05 | 0.084 | 0.093 | 28.7% | 41.0% |
| 50.0 | 0.05 | 0.075 | 0.093 | 21.3% | 41.0% |
| 75.0 | 0.01 | 0.052 | 0.026 | 19.1% | 11.6% |
| 150.0 | 0.01 | 0.052 | 0.026 | 15.5% | 11.6% |
| 200.0 | 0.01 | 0.046 | 0.026 | 10.8% | 11.6% |

Table 8: Match on coverage: for each weight $w$, choose $\alpha$ minimizing the difference in coverage between the FRL and the conformal prediction methods.

| Weight | $\alpha$ | FRL $\ell_1$ | CP $\ell_1$ | FRL coverage | CP coverage |
|---:|---|---|---|---:|---:|
| 1.0 | 0.25 | 0.226 | 0.223 | 100.0% | 98.5% |
| 5.0 | 0.10 | 0.136 | 0.155 | 56.7% | 64.5% |
| 10.0 | 0.05 | 0.111 | 0.093 | 43.2% | 41.0% |
| 25.0 | 0.05 | 0.084 | 0.093 | 28.7% | 41.0% |
| 50.0 | 0.01 | 0.075 | 0.026 | 21.3% | 11.6% |
| 75.0 | 0.01 | 0.052 | 0.026 | 19.1% | 11.6% |
| 150.0 | 0.01 | 0.052 | 0.026 | 15.5% | 11.6% |
| 200.0 | 0.01 | 0.046 | 0.026 | 10.8% | 11.6% |

Table 9: Match on $\ell_1$: for each weight $w$, choose $\alpha$ minimizing the difference in $\ell_1$ error between the FRL and the conformal prediction methods.

| Weight | $\alpha$ | FRL $\ell_1$ | CP $\ell_1$ | FRL coverage | CP coverage |
|---:|---|---|---|---:|---:|
| 1.0 | 0.25 | 0.239 | 0.247 | 100.0% | 98.8% |
| 5.0 | 0.10 | 0.127 | 0.147 | 57.3% | 67.2% |
| 10.0 | 0.05 | 0.104 | 0.083 | 43.3% | 41.1% |
| 25.0 | 0.05 | 0.070 | 0.083 | 29.9% | 41.1% |
| 50.0 | 0.01 | 0.047 | 0.038 | 23.4% | 13.0% |
| 75.0 | 0.01 | 0.046 | 0.038 | 19.5% | 13.0% |
| 150.0 | 0.01 | 0.034 | 0.038 | 14.9% | 13.0% |
| 200.0 | 0.01 | 0.000 | 0.038 | 10.2% | 13.0% |

Table 10: Match on coverage: for each weight $w$, choose $\alpha$ minimizing the difference in coverage between the FRL and the conformal prediction methods.

| Weight | $\alpha$ | FRL $\ell_1$ | CP $\ell_1$ | FRL coverage | CP coverage |
|---|---|---|---|---|---|
| 1.0 | 0.25 | 0.239 | 0.247 | 100.0% | 98.8% |
| 5.0 | 0.10 | 0.127 | 0.147 | 57.3% | 67.2% |
| 10.0 | 0.05 | 0.104 | 0.083 | 43.3% | 41.1% |
| 25.0 | 0.05 | 0.070 | 0.083 | 29.9% | 41.1% |
| 50.0 | 0.01 | 0.047 | 0.038 | 23.4% | 13.0% |
| 75.0 | 0.01 | 0.046 | 0.038 | 19.5% | 13.0% |
| 150.0 | 0.01 | 0.034 | 0.038 | 14.9% | 13.0% |
| 200.0 | 0.01 | 0.000 | 0.038 | 10.2% | 13.0% |

*Proof.* Let $g \in \mathcal{G}$ be some group and let us focus on the part of the domain $\mathcal{X}_g = \{x \mid g(x) = 1\}$. Consider the classes $\mathcal{C}_g^+, \mathcal{C}_g^-$ derived from the Boolean concept class $\mathcal{C}$, where $\mathcal{C}_g^+(\mathcal{D}) = \{c \in \mathcal{C} \mid \Pr[c(\mathbf{x}) = 1, g(\mathbf{x}) = 1, \mathbf{y} = 0] = 0\}$ and $\mathcal{C}_g^-(\mathcal{D}) = \{c \in \mathcal{C} \mid \Pr[c(\mathbf{x}) = 0, g(\mathbf{x}) = 1, \mathbf{y} = 1] = 0\}$.

Let $h$ be a $(\mathcal{C} \cdot \mathcal{G}, \epsilon^2/8)$-multiaccurate and calibrated predictor for $\mathcal{X}, \mathcal{D}$, which we can build using the main theorem in [26]. Consider any $c_+ \in \mathcal{C}_g^+ \subseteq \mathcal{C}$. The multiaccuracy condition is ensured for all $c \cdot g \in \mathcal{C} \cdot \mathcal{G}$, and so for this particular $c_+$ and fixed group $g$ we have that

$$\left| \mathop{\mathbb{E}}_{\mathcal{D}} \left[ (g(\mathbf{x}) \cdot c_+(\mathbf{x}))(\mathbf{y} - h(\mathbf{x})) \right] \right| \leq \epsilon^2/8.$$

Given that whenever $c_+(x) = 1$ and $g(x) = 1$, we have that $y = 1$ by definition of $C_g^+$, it follows that

$$\mathop{\mathbb{E}}_{\mathcal{D}}[\mathbf{y} \mid g(\mathbf{x}) = 1, c_+(\mathbf{x}) = 1] = 1.$$

That is, within the region in $\mathcal{X}_g$ where $c_+(x) = 1$, we have that $\mathbb{E}_{\mathcal{D}}[\mathbf{y}] = 1$. Then, by the multiaccuracy guarantee on $c_+ \cdot g$, it follows that the expected value of $h$ over the same region (i.e., where $c_+(\mathbf{x}) = 1$ inside of $\mathcal{X}_g$) is also close to 1:

$$\left| \mathop{\mathbb{E}}_{\mathcal{D}} \left[ (g(\mathbf{x}) \cdot c_+(\mathbf{x}))(\mathbf{y} - h(\mathbf{x})) \right] \right| = \left| \mathop{\mathbb{E}}_{\mathcal{D}} \left[ (g(\mathbf{x}) \cdot c_+(\mathbf{x}))(1 - h(\mathbf{x})) \right] \right| \leq \epsilon^2/8,$$

and so

$$\mathop{\mathbb{E}}_{\mathcal{D}}[h(\mathbf{x}) \mid g(\mathbf{x}) = 1, c_+(\mathbf{x}) = 1] \geq 1 - \frac{\epsilon^2}{8 \, \mathbb{E}_{\mathcal{D}}[g(\mathbf{x})c_+(\mathbf{x})]}. \tag{2}$$

Now, we have two cases, either $\Pr[c_+(\mathbf{x}) = 1, g(x) = 1] = \mathbb{E}_{\mathcal{D}}[c_+(\mathbf{x})g(\mathbf{x})] \leq \epsilon/2$, in which case we trivially have,

$$\mathbb{E} \left[ \mathbb{1}[h(\mathbf{x}) \geq 1 - \epsilon] \cdot g(\mathbf{x}) \right] \geq \mathop{\mathbb{E}}_{\mathcal{D}}[c_+(\mathbf{x})g(\mathbf{x})] - \epsilon/2,$$

or, from Equation 2, we have that

$$\mathop{\mathbb{E}}_{\mathcal{D}} \left[ \mathbb{1}[h(x) \leq 1 - \epsilon] \mid g(\mathbf{x}) = 1, c_+(\mathbf{x}) = 1 \right] \leq \frac{\epsilon}{4 \, \mathbb{E}_{\mathcal{D}}[g(\mathbf{x})c_+(\mathbf{x})]}.$$

In this case, by taking the complement and multiplying both sides by $\mathbb{E}_{\mathcal{D}}[g(\mathbf{x})c_+(\mathbf{x})]$, we get,

$$\mathop{\mathbb{E}}_{\mathcal{D}} \left[ \mathbb{1}[h(\mathbf{x}) \geq 1 - \epsilon] \cdot g(\mathbf{x}) \right] \geq \mathop{\mathbb{E}}_{\mathcal{D}} \left[ \mathbb{1}[h(x) \geq 1 - \epsilon] \cdot g(\mathbf{x})c_+(\mathbf{x}) \right]$$
$$\geq \mathop{\mathbb{E}}_{\mathcal{D}}[c_+(\mathbf{x})g(\mathbf{x})] - \epsilon/4.$$

Thus, in either case we have,

$$\mathop{\mathbb{E}}_{\mathcal{D}} \left[ \mathbb{1}[h(\mathbf{x}) \geq 1 - \epsilon] \cdot g(\mathbf{x}) \right] \geq \mathop{\mathbb{E}}_{\mathcal{D}}[c_+(\mathbf{x})g(\mathbf{x})] - \epsilon/2.$$

A symmetric argument holds in the case of $c_- \in \mathcal{C}^-$ with the same group $g$. By the definition of $\mathcal{C}_g^-(\mathcal{D})$, it holds that whenever $c_-(\mathbf{x}) = 0$ and $g(\mathbf{x}) = 1$, $\mathbf{y} = 0$, and so within the region in $\mathcal{X}_g$ where $c_-(\mathbf{x}) = 0$, $\mathbb{E}_{\mathcal{D}}[\mathbf{y}] = 0$. Let $\bar{c}_-$ be the complement of $c_-$ (which we can take since $\mathcal{C}$ is closed under complement). Then, it follows that:

$$\mathop{\mathbb{E}}_{\mathcal{D}}[\mathbf{y} \mid g(\mathbf{x}) = 1, \bar{c}_-(\mathbf{x}) = 1] = 0.$$

By the multiaccuracy guarantee on $\bar{c}_- \cdot g$, we can use an identical argument above (by a case distinction on whether $\Pr[c_-(\mathbf{x}) = 0, g(\mathbf{x}) = 1] \geq \epsilon/2$) to show that

$$\mathbb{E}_{\mathcal{D}}\left[\mathbb{1}[h(\mathbf{x}) \leq \epsilon] \cdot g(\mathbf{x})\right] \geq \mathbb{E}_{\mathcal{D}}[\bar{c}_-(\mathbf{x})g(\mathbf{x})] - \epsilon/2,$$

From the $(\mathcal{C} \cdot \mathcal{G}, \epsilon^2/8)$-multiaccurate and calibrated $h$, we construct our $(\mathcal{C}, \mathcal{G}, \epsilon)$-multigroup selective classifier $h_?$ by post-processing $h$ as follows:

$$h_?(x) = \begin{cases} h(x) & \text{if } h(x) \in [0, \epsilon] \cup [1 - \epsilon, 1], \\ ? & \text{if } h(x) \in (\epsilon, 1 - \epsilon). \end{cases}$$

Observe that

$$\mathbb{E}_{\mathcal{D}}\left[\mathbb{1}[h_?(\mathbf{x}) \neq ?] \cdot g(\mathbf{x})\right] = \mathbb{E}_{\mathcal{D}}\left[\mathbb{1}[h(\mathbf{x}) \geq 1 - \epsilon] \cdot g(\mathbf{x})\right] + \mathbb{E}_{\mathcal{D}}\left[\mathbb{1}[h(\mathbf{x}) \leq \epsilon] \cdot g(\mathbf{x})\right]$$

$$\geq \mathbb{E}_{\mathcal{D}}[c_+(\mathbf{x})g(\mathbf{x})] + \mathbb{E}_{\mathcal{D}}[\bar{c}_-(\mathbf{x})g(\mathbf{x})] - \epsilon.$$

Recall that any $c_? \in \mathrm{SC}(\mathcal{C})$ is of the form $(c_+, c_-)$. By the construction of $\mathrm{SC}(\mathcal{C})$ (see Section 2.3) and given that $\bar{c}_-$ is the complement of $c_-$ it follows that

$$\mathbb{E}_{\mathcal{D}}\left[\mathbb{1}[c_?(\mathbf{x}) \neq ?] \cdot g(\mathbf{x})\right] \leq \mathbb{E}_{\mathcal{D}}[(c_+(\mathbf{x}) + \bar{c}_-(\mathbf{x}))g(\mathbf{x})]$$

Therefore, the two above equation show that

$$\mathbb{E}_{\mathcal{D}}\left[\mathbb{1}[h_?(\mathbf{x}) \neq ?] \cdot g(\mathbf{x})\right] \geq \mathbb{E}_{\mathcal{D}}\left[\mathbb{1}[c_?(\mathbf{x}) \neq ?] \cdot g(\mathbf{x})\right] - \epsilon.$$

Hence, on each $g \in \mathcal{G}$, $h_?$ does not abstain significantly more often than the optimal $c_?$ for each group.

Note that the global calibration property of $h_?$ and the fact that the predicted values of $h_?(\mathbf{x})$ are in $[0, \epsilon)$ or $(1 - \epsilon, 1]$ implies that both the false positive and negative rates of $h_?$ are bounded by $\epsilon$ thus satisfying Condition 1 in Definition 4.1. $\qquad \square$

Note that to achieve the global accuracy guarantee, we only used the global calibration condition and the fact that we abstain in the region where $h(x) \in (\epsilon, 1 - \epsilon)$ (per our definition of $h_?$ from $h$). In the other hand, to achieve the local abstention guarantee, we used the multiaccuracy guarantee and the definitions of $\mathcal{C}^+$ and $\mathcal{C}^-$. Hence, our proof clearly delineates the complementary roles played by each of the multiaccuracy and calibration. This complementarity is structurally similar to the recent result by [11] demonstrating that calibrated multiaccuracy implies agnostic learning.

It is natural to ask whether we can efficiently construct $(\mathcal{C}, \mathcal{G})$-multigroup selective classifiers starting from a weaker learning primitive than that of a calibrated multiaccurate predictor for the class $\mathcal{C} \cap \mathcal{G}$. What is the weakest learning primitive that we can build it from? In Theorem 4.2 below, we answer this question in the positive in the case where $|\mathcal{G}|$ is small. Specifically, we show that in this case, we can construct it from fully reliable learning:

**Lemma 4.2.** *If the class $\mathcal{C}$ is fully reliably learnable for the $\ell_1$ loss and $\epsilon > 0$, we can construct a $(\mathcal{C}, \mathcal{G}, \epsilon)$-multigroup selective classifier in time $\mathrm{poly}(|\mathcal{G}|, 1/\epsilon)$ with oracle access to the full reliable learner for $\mathcal{C}$.*

*Proof.* For each $g \in \mathcal{G}$, we call the FRL oracle with the restricted domain $\{x \mid g(x) = 1\}$, error parameter $\epsilon/|\mathcal{G}|$, classes $\mathcal{C}$, $\mathcal{G}$, and the 0-1 loss function. Let $h_?^g : \{x \mid g(x) = 1\} \to \{0, 1, ?\}$ denote the selective classifier that we obtain with an FRL call with the domain $\mathcal{X}_g = \{x \mid g(x) = 1\}$. We construct our final global classifier $h_? : \mathcal{X} \to \{0, 1, ?\}$ as follows: for every point $x \in \mathcal{X}$, consider the set $\mathcal{K} \subseteq \mathcal{G}$ of all groups $g \in \mathcal{C}$ such that $g(x) = 1$. Then, $h_?$ is equal to the plurality vote of the $|\mathcal{K}|$ total classifiers $h_?^g$.

By the FRL guarantee, each $h_?^g$ makes a wrong prediction on at most $\epsilon/|\mathcal{G}|$ points in $\mathcal{X}$. In the worst case, all the $|\mathcal{G}|$ error regions are disjoint and cause all the majority votes taken on the points $x$ in these regions to cause the wrong prediction. Hence,

$$\mathrm{err}_{\mathcal{D}}(h_?) = \mathbb{E}_{\mathcal{D}}[|\mathbf{y} - h_?(\mathbf{x})| \cdot \mathbb{1}[h_?(\mathbf{x}) \neq ?]] = |\mathcal{G}| \cdot \frac{\epsilon}{|\mathcal{G}|} \leq \epsilon,$$

and thus we satisfy Condition 1 in the definition of $(\mathcal{C}, \mathcal{G})$-multigroup selective classification (global accuracy).

Secondly, we have not added further abstentions in our ensembling of the $h_?^g$ predictors, and so we continue to satisfy optimal local abstention rate within each $g \in \mathcal{G}$. Namely, the FRL guarantee on each $g$ ensures that

$$\Pr_{(\mathbf{x},\mathbf{y})\sim\mathcal{D}} \left[ g(\mathbf{x}) \cdot \mathbb{1}[h_?(\mathbf{x}) = ?] \right] \leq \min_{c_? \in \mathrm{SC}(\mathcal{C})} \Pr_{(\mathbf{x},\mathbf{y})\sim\mathcal{D}} \left[ g(\mathbf{x}) \cdot \mathbb{1}[c_?(\mathbf{x}) = ?] \right] + \epsilon.$$

In fact, we are getting a better $\epsilon/|\mathcal{G}|$ error. Hence, we satisfy Condition 2 in the definition in the definition of $(\mathcal{C}, \mathcal{G})$-multigroup selective classification. Thus we have shown that $h_?$ is a $(\mathcal{C}, \mathcal{G}, \epsilon)$-multigroup selective classifier, as desired.

$\square$

## F.2 Multigroup fairness primitives

We recall that multicalibration is a stronger notion than calibrated multiaccuracy, which is in turn a stronger notion than multiaccuracy.

*(a) If we have the stronger primitive of a $(\mathcal{C} \cdot \mathcal{G})$-multicalibrated predictor (which implies a $(\mathcal{C} \cdot \mathcal{G})$-multiaccurate calibrated predictor), then we can have a non-selective predictor which would give local agnostic guarantees. This follows from the works of [25, 10, 27].*

Formalized:

**Lemma F.1.** *Given access to a $(\mathcal{C} \cap \mathcal{G}, \epsilon)$-multicalibrated predictor, we can construct a classifier $h : \mathcal{X} \to [0,1]$ in time $\mathrm{poly}(1/\epsilon)$ satisfying the following local accuracy property: for every $g \in \mathcal{G}$,*

$$\mathbb{E}_{(\mathbf{x},\mathbf{y})\sim\mathcal{D}} [|\mathbf{y} - h(\mathbf{x})| \mid g(\mathbf{x}) = 1] \leq \min_{c \in \mathcal{C}} \mathbb{E}_{(\mathbf{x},\mathbf{y})\sim\mathcal{D}} [|\mathbf{y} - c(\mathbf{x})| \mid g(\mathbf{x}) = 1] + \epsilon.$$

Note that a $(\mathcal{C}, \mathcal{G}, \epsilon)$-multigroup selective classifier achieves a local accuracy property within each $g \in \mathcal{G}$, where the error term is weighted by $1/\Pr_{\mathcal{D}}[g(\mathbf{x}) = 1]$. In Lemma F.1, however, we do so without requiring abstentions, which is why we need the stronger notion of multicalibration rather than of calibrated multiaccuracy. Note that accuracy notion in Lemma F.1 is of the agnostic form, rather than an absolute error guarantee.

*(b) Given a $(\mathcal{C} \cdot \mathcal{G}, \epsilon)$-multiaccurate and calibrated predictor, we can directly obtain FRL predictors for the class $\mathcal{C}$ for the $\ell_1$ loss by thresholding the predictor as we do in the proof of Theorem 4.1. Given that calibrated multiaccuracy implies agnostic learning [11], it is already implied by Theorem 4.1 that we can achieve reliable agnostic learning from calibrated multiaccuracy. However, this approach gives a direct reduction. Reliable agnostic learning is believed to be a weaker learning primitive than agnostic learning [45].*

Formalized:

**Lemma F.2.** *Given access to a $(\mathcal{C} \cdot \mathcal{G}, \epsilon)$-multiaccurate and calibrated predictor $h$, we can apply a post-processing function to $h$ to obtain an $\mathcal{L}$-FRL predictor for the $\ell_1$ loss.*

*Proof.* Let $h$ be a $(\mathcal{C} \cdot \mathcal{G}, \epsilon)$-multiaccurate and calibrated predictor. We first construct a $\mathcal{L}$-FRL predictor $h_?$ from $h$ by applying the same post-processing function as in the proof of Theorem 4.1. Namely, we let

$$h_?(x) = \begin{cases} h(x) & \text{if } h(x) \in [0,\epsilon] \cup [1-\epsilon, 1], \\ ? & \text{if } h(x) \in (\epsilon, 1-\epsilon). \end{cases}$$

By the definition of FRL, we need to show that $h_?$ satisfies 1) low global error, and 2) optimal abstention. This follows directly from our Theorem 4.1 by using the group $g = \mathbf{1}$. $\square$

We remark that we can extend the previous lemma to the 0-1 loss.

### F.3 Conformal prediction from $(C, G)$-multigroup selective classification

**Lemma 4.3.** *Let $h_? : \mathcal{X} \to [0, 1] \cup \{?\}$ be a $(C, G, \epsilon)$-multigroup selective classifier. Then, the prediction sets induced by the level sets of $h$, where we map $[0, \epsilon] \mapsto \{0\}, [1 - \epsilon, 1] \mapsto \{1\}$, and $(\epsilon, 1 - \epsilon) \mapsto \{0, 1\}$ satisfy $\Pr_{(\mathbf{x}, \mathbf{y}) \sim \mathcal{D}}[\mathbf{y} \in \mathcal{S}(x) \mid g(\mathbf{x}) = 1] \geq 1 - \frac{\epsilon}{\Pr_{\mathcal{D}}[g(\mathbf{x}) = 1]}$ for all $g \in \mathcal{G}$.*

*Proof.* By the definition of a $(C, G, \epsilon)$-multigroup selective classifier, it follows that $h_?$ satisfies the global accuracy guarantee with an $\epsilon$ error parameter. This directly implies a local accuracy guarantee on each $g \in \mathcal{G}$ if we weight the error parameter by the probability mass assigned by $\mathcal{D}$ to $g$; that is:

$$\mathop{\mathbb{E}}_{(\mathbf{x}, \mathbf{y}) \sim \mathcal{D}} \left[ |\mathbf{y} - h_?(\mathbf{x})| \cdot \mathbb{1}[h_?(\mathbf{x}) \neq ?] \mid g(\mathbf{x}) = 1 \right] \leq \frac{\epsilon}{\Pr_{\mathcal{D}}[g(\mathbf{x}) = 1]}.$$

Then, the result follows from the fact that for all $\mathbf{y} = 1$,

$$\Pr_{(\mathbf{x}, \mathbf{y}) \sim \mathcal{D}} [\mathbf{y} \in \{1\} \mid g(\mathbf{x}) = 1] \leq \mathop{\mathbb{E}}_{(\mathbf{x}, \mathbf{y}) \sim \mathcal{D}} \left[ |\mathbf{y} - h_?(\mathbf{x})| \cdot \mathbb{1}[h_?(\mathbf{x}) \neq ?] \mid g(\mathbf{x}) = 1 \right],$$

and symmetrically for all $\mathbf{y} = 0$ it follows that

$$\Pr_{(\mathbf{x}, \mathbf{y}) \sim \mathcal{D}} [\mathbf{y} \in \{0\} \mid g(\mathbf{x}) = 1] \leq \mathop{\mathbb{E}}_{(\mathbf{x}, \mathbf{y}) \sim \mathcal{D}} \left[ |\mathbf{y} - h_?(\mathbf{x})| \cdot \mathbb{1}[h_?(\mathbf{x}) \neq ?] \mid g(\mathbf{x}) = 1 \right].$$

$\square$

Lastly, we formalize and prove Remark 4.1, which shows that $(C, G, \epsilon)$-multigroup selective classification, when used as a conformal prediction method, besides satisfying the conditional coverage guarantee, also provides a provable upper bound on the size of the set $\{0, 1\}$ within each group $g \in \mathcal{G}$:

**Lemma F.3.** *Let $h_? : \mathcal{X} \to [0, 1] \cup ?$ be a $(C, G, \epsilon)$-multigroup selective classifier. Then, the prediction sets induced by the level sets of $h_?$ satisfy:*

$$\Pr_{(\mathbf{x}, \mathbf{y}) \sim \mathcal{D}} \left[ g(\mathbf{x}) \cdot \mathbb{1}[h_?(\mathbf{x}) = \{0, 1\}] \right] \leq \min_{c_? \in \text{SC}(C)} \Pr_{(\mathbf{x}, \mathbf{y}) \sim \mathcal{D}} \left[ g(\mathbf{x}) \cdot \mathbb{1}[c_?(\mathbf{x}) = ?] \right] + \epsilon$$

*for every group $g \in \mathcal{G}$.*

*Proof.* This follows directly from the optimal local abstention rate guarantee of a $(C, G, \epsilon)$-multigroup selective classifier, which ensures that for every $g \in \mathcal{G}$,

$$\Pr_{(\mathbf{x}, \mathbf{y}) \sim \mathcal{D}} \left[ g(\mathbf{x}) \cdot \mathbb{1}[h_?(\mathbf{x}) = ?] \right] \leq \min_{c_? \in \text{SC}(C)} \Pr_{(\mathbf{x}, \mathbf{y}) \sim \mathcal{D}} \left[ g(\mathbf{x}) \cdot \mathbb{1}[c_?(\mathbf{x}) = ?] \right] + \epsilon.$$

$\square$

## G   Experiments for Section 4

Lastly, we implement our $(C, G)$-multigroup selective ominpredictors in practice. To do so, we modify the construction our proof of Theorem 4.1 to adapt the multicalibration algorithm that we used in the experiments for Section 3. Again for our proof of concept, we generate $10,000$ samples synthetically using `sk-learn`'s `make_classification` function. For the concept class $C$, we use the same class of decision trees of depth 3. For the class of groups $G$, we generate them from the data using randomness by allowing the groups to intersect and by ensuring some correlation with the labels within each group. In these experiments, we use 10 groups, and each ends up having size of between 300 and 400 samples.

We use the same discretization parameter $0.1$ and learning rate $0.01$ in the update step as in the experiments for Section 3.1. We perform the multicalibration algorithm across all groups in $\mathcal{G}$. In each, we use the same concept class $C$ of decision trees of depth 3 to find correlation with the residuals and we cap the number of iterations at 150. We see that the $(C, G)$-multicalibration greatly reduces the $\ell_2$ (i.e., Brier score) and expected calibration error (ECE) for each of the groups in $\mathcal{G}$, as shown in Table 11. We emphasize that, as far as we are aware, our work is the first to separate the roles of the concept class $\mathcal{G}$ and the group collection $\mathcal{G}$ (in the multigroup fairness literature, one usually sets $\mathcal{C} = \mathcal{G}$).

Table 11: Group-wise metrics before and after $(\mathcal{C}, \mathcal{G})$-multicalibration.

| $G$ | $N$ | $\text{Brier}_{\text{pre}}$ | $\text{Brier}_{\text{post}}$ | $\text{ECE}_{\text{pre}}$ | $\text{ECE}_{\text{post}}$ |
|---|---|---|---|---|---|
| 0 | 414 | 0.106 | 0.004 | 0.162 | 0.036 |
| 1 | 389 | 0.122 | 0.003 | 0.139 | 0.036 |
| 2 | 419 | 0.120 | 0.005 | 0.158 | 0.046 |
| 3 | 395 | 0.119 | 0.006 | 0.164 | 0.047 |
| 4 | 416 | 0.111 | 0.005 | 0.170 | 0.047 |
| 5 | 387 | 0.111 | 0.005 | 0.178 | 0.051 |
| 6 | 410 | 0.116 | 0.009 | 0.118 | 0.054 |
| 7 | 400 | 0.120 | 0.008 | 0.152 | 0.060 |
| 8 | 388 | 0.120 | 0.011 | 0.168 | 0.063 |
| 9 | 380 | 0.116 | 0.011 | 0.156 | 0.060 |

Table 12: Group-wise metrics before and after $(\mathcal{C}, \mathcal{G})$-multicalibration.

| $G$ | $N$ | $\text{Brier}_{\text{pre}}$ | $\text{Brier}_{\text{post}}$ | $\text{ECE}_{\text{pre}}$ | $\text{ECE}_{\text{post}}$ |
|---|---|---|---|---|---|
| 0 | 381 | 0.072 | 0.002 | 0.149 | 0.024 |
| 1 | 394 | 0.084 | 0.003 | 0.134 | 0.029 |
| 2 | 400 | 0.079 | 0.006 | 0.137 | 0.033 |
| 3 | 406 | 0.081 | 0.004 | 0.142 | 0.036 |
| 4 | 380 | 0.089 | 0.008 | 0.140 | 0.049 |
| 5 | 390 | 0.095 | 0.007 | 0.136 | 0.051 |
| 6 | 400 | 0.090 | 0.009 | 0.137 | 0.045 |
| 7 | 396 | 0.090 | 0.010 | 0.141 | 0.052 |
| 8 | 416 | 0.095 | 0.010 | 0.126 | 0.053 |
| 9 | 379 | 0.082 | 0.011 | 0.158 | 0.048 |

We give another example from another run with a different initialization of synthetic data:

Next, we turn our $(\mathcal{C}, \mathcal{G})$-multicalibrated predictor into a $(\mathcal{C}, \mathcal{G})$-multigroup selective classifier, we apply the thresholding function that we repeatedly use in our proofs in Section 4: namely, for a chosen value of $\epsilon$, we map the predicted value to 0 if it is $\leq \epsilon$, to 1 if it is $\geq \epsilon$, and to ? otherwise. We then compute the coverage and $\ell_1$ error (i.e., mean absolute error) of our $(\mathcal{C}, \mathcal{G})$-multigroup selective classifier within each of the groups $g \in \mathcal{G}$ for different values of $\epsilon$.

Separately, we again use the MAPIE library to train a conformal prediction method *separately within each of the groups* $g \in \mathcal{G}$. Notably, this methodology does not technically allow the groups to intersect, but we report the statistics independently on every group. In contrast, our method yields one global predictor, instead of a predictor for each group that does not allow for intersections. This is a significant advantage of using $(\mathcal{C}, \mathcal{G})$-multigroup selective classification as a conformal prediction method. We remark that two recent works use the multicalibration algorithm to obtain group conditional coverage guarantees for an intersecting collection of groups and perform extensive evaluations [38, 39]. We once again remark that the significance of our theoretical results in Section 4 are to extend the multigroup fairness framework to learn how to abstain fairly, and that the use of our algorithms as conformal prediction methods is presented as a use case rather than the end goal.

For the MAPIE training, we use random forests as the base class and $\alpha = 0.1$ as the parameter for the coverage guarantee. After training the predictor, we obtain the prediction sets $\{0\}$, $\{1\}$, and $\{0, 1\}$ for each of the points in the test set. Viewing $\{0, 1\}$ as equivalent to ?, we compute the coverage and $\ell_1$ error of the conformal prediction method within each group (where we remark that the predictor is trained separately for every group, unlike our selective classifier). We report the coverage and $\ell_1$ errors of the selective classifier for different values of $\epsilon$ (specifically, $\epsilon = 0.2, 0.3, 0.4$) and compare it to the coverage and $\ell_1$ errors of the per-group conformal prediction method. We perform different runs with different initializations of the synthetic data and report the results in Tables 13, 14, and 15. All runs show a similar pattern: the selective classifier is competitive with the conformal prediction method within each group $g \in \mathcal{G}$, even though we have a single predictor for all groups and the conformal prediction method is trained separately on each group.

Table 13: Per-group Coverage and $\ell_1$ error for $(\mathcal{C}, \mathcal{G})$-multigroup selective classification thresholding at $\epsilon = 0.20$, $\epsilon = 0.30$, $\epsilon = 0.40$, compared with a separate conformal prediction predictor per group.

| Group | N | $\text{Cov}_{0.20}$ | $\ell_{10.20}$ | $\text{Cov}_{0.30}$ | $\ell_{10.30}$ | $\text{Cov}_{0.40}$ | $\ell_{10.40}$ | $\text{Cov}^{\mathcal{G}}$ | $\ell_1^{\mathcal{G}}$ |
|---|---|---|---|---|---|---|---|---|---|
| 0 | 421 | 0.983 | 0.104 | 0.988 | 0.106 | 1.000 | 0.107 | 0.988 | 0.108 |
| 1 | 405 | 0.983 | 0.118 | 0.985 | 0.118 | 0.990 | 0.120 | 0.938 | 0.100 |
| 2 | 408 | 0.975 | 0.131 | 0.983 | 0.135 | 0.993 | 0.141 | 0.963 | 0.115 |
| 3 | 384 | 0.977 | 0.120 | 0.979 | 0.122 | 0.992 | 0.123 | 0.958 | 0.114 |
| 4 | 410 | 0.949 | 0.118 | 0.956 | 0.117 | 0.973 | 0.128 | 0.956 | 0.128 |
| 5 | 401 | 0.960 | 0.114 | 0.970 | 0.116 | 0.980 | 0.120 | 0.963 | 0.098 |
| 6 | 388 | 0.961 | 0.123 | 0.977 | 0.129 | 0.985 | 0.134 | 0.951 | 0.108 |
| 7 | 399 | 0.952 | 0.124 | 0.967 | 0.122 | 0.987 | 0.124 | 0.962 | 0.109 |
| 8 | 400 | 0.932 | 0.123 | 0.948 | 0.127 | 0.970 | 0.139 | 0.963 | 0.109 |
| 9 | 409 | 0.914 | 0.131 | 0.917 | 0.133 | 0.961 | 0.155 | 0.941 | 0.132 |

Table 14: Per-group Coverage and $\ell_1$ error for $(\mathcal{C}, \mathcal{G})$-multigroup selective classification thresholding at $\epsilon = 0.20$, $\epsilon = 0.30$, $\epsilon = 0.40$, compared with a separate conformal prediction predictor per group.

| Group | N | $\text{Cov}_{0.20}$ | $\ell_{10.20}$ | $\text{Cov}_{0.30}$ | $\ell_{10.30}$ | $\text{Cov}_{0.40}$ | $\ell_{10.40}$ | $\text{Cov}^{\mathcal{G}}$ | $\ell_1^{\mathcal{G}}$ |
|---|---|---|---|---|---|---|---|---|---|
| 0 | 407 | 0.973 | 0.078 | 0.978 | 0.080 | 0.988 | 0.082 | 0.988 | 0.082 |
| 1 | 381 | 0.958 | 0.090 | 0.969 | 0.100 | 0.992 | 0.098 | 0.969 | 0.095 |
| 2 | 422 | 0.979 | 0.099 | 0.988 | 0.101 | 0.995 | 0.102 | 0.988 | 0.118 |
| 3 | 386 | 0.948 | 0.093 | 0.959 | 0.092 | 0.974 | 0.096 | 0.984 | 0.111 |
| 4 | 384 | 0.948 | 0.082 | 0.958 | 0.087 | 0.979 | 0.098 | 0.992 | 0.102 |
| 5 | 391 | 0.954 | 0.056 | 0.959 | 0.059 | 0.977 | 0.068 | 0.982 | 0.078 |
| 6 | 389 | 0.928 | 0.066 | 0.941 | 0.071 | 0.967 | 0.077 | 0.985 | 0.081 |
| 7 | 400 | 0.910 | 0.074 | 0.932 | 0.078 | 0.955 | 0.081 | 0.998 | 0.128 |
| 8 | 411 | 0.937 | 0.078 | 0.956 | 0.084 | 0.973 | 0.092 | 0.985 | 0.099 |
| 9 | 414 | 0.908 | 0.069 | 0.923 | 0.071 | 0.952 | 0.081 | 0.988 | 0.108 |

## G.1 Code reproducibility

We include all of the code that we use to obtain the experimental results for Sections 3 and 4 as a ZIP file in the supplementary material. These experiments were conducted locally using a system equipped with an M1 chip and 16 GB of local memory. We used ChatGPT to help debug the code and implement the abstaining decision trees, and we studied the multicalibration code provided in the Python package from the paper [29] to aid us with our implementation (which we did from scratch, given that the implementation in [29] finds correlation with the residuals using the Boolean groups $g$ in $\mathcal{G}$, whereas we want to use the real-valued concepts $c$ in the concept class $\mathcal{C}$).

Table 15: Per-group Coverage and $\ell_1$ error for $(\mathcal{C}, \mathcal{G})$-multigroup selective classification thresholding at $\epsilon = 0.20$, $\epsilon = 0.30$, $\epsilon = 0.40$, compared with a separate conformal prediction predictor per group.

| Group | N | $\text{Cov}_{0.20}$ | $\ell_{10.20}$ | $\text{Cov}_{0.30}$ | $\ell_{10.30}$ | $\text{Cov}_{0.40}$ | $\ell_{10.40}$ | $\text{Cov}^{\mathcal{G}}$ | $\ell_1^{\mathcal{G}}$ |
|---|---|---|---|---|---|---|---|---|---|
| 0 | 393 | 0.977 | 0.146 | 0.985 | 0.147 | 1.000 | 0.148 | 0.827 | 0.138 |
| 1 | 409 | 0.971 | 0.131 | 0.983 | 0.132 | 0.990 | 0.133 | 0.856 | 0.131 |
| 2 | 381 | 0.971 | 0.119 | 0.979 | 0.118 | 0.984 | 0.120 | 0.803 | 0.101 |
| 3 | 360 | 0.964 | 0.118 | 0.975 | 0.120 | 0.992 | 0.132 | 0.831 | 0.110 |
| 4 | 377 | 0.966 | 0.140 | 0.981 | 0.141 | 0.984 | 0.143 | 0.862 | 0.154 |
| 5 | 389 | 0.961 | 0.155 | 0.979 | 0.157 | 0.985 | 0.157 | 0.823 | 0.150 |
| 6 | 387 | 0.948 | 0.125 | 0.974 | 0.130 | 0.987 | 0.134 | 0.858 | 0.139 |
| 7 | 361 | 0.958 | 0.150 | 0.970 | 0.154 | 0.992 | 0.165 | 0.853 | 0.169 |
| 8 | 373 | 0.920 | 0.122 | 0.952 | 0.124 | 0.973 | 0.129 | 0.847 | 0.133 |
| 9 | 393 | 0.959 | 0.149 | 0.985 | 0.150 | 0.987 | 0.149 | 0.852 | 0.152 |

