# OpenReview forum: "Selective Omniprediction and Fair Abstention"
_NeurIPS.cc/2025/Conference — NeurIPS 2025 spotlight_

### Official Review · Reviewer_6Q23 · 2025-07-02

**Clarity:** 3
**Significance:** 3
**Originality:** 3
**Rating:** 5
**Confidence:** 4

**Summary:**

This paper proposes extending the “omniprediction” framework of Gopalan et al. to also allow predictors to abstain (a la selective prediction frameworks). The paper begins by introducing omniprediction and the multi-group fairness notions which allow achievable omniprediction (multiaccuracy, multicalibration, calibrated multiaccuracy). In section 2.3, the paper introduces the reliable agnostic learning framework. Any loss $\ell$ is decomposed into a triple of losses $(\ell_+, \ell_{-}, \ell_{?})$, which are respectively the cost of misclassifying a negative prediction, positive prediction, and the cost of abstention.

Given a concept class $\mathcal{C}$, the paper defines $\mathcal{C}^+$ as the subset of concepts in $\mathcal{C}$ which have $\ell_+(c, \mathcal{D}) = 0$, that is, the subset of hypotheses which never have false positive predictions on the distribution $\mathcal{D}$. Similarly define $\mathcal{C}^-$ as the subset of hypotheses which never have false negative predictions.

The paper then defines the (agnostic) positively reliable learning (PRL) framework of Kalai et al. and Kanade et al. In short, a concept class $\mathcal{C}$ with loss pair $(\ell_+, \ell_{-})$ is PRLearnable if the learner can output an $h$ w.h.p. Obtains both:
In expectation, $\ell_+(h(x)) \leq \epsilon $
$h$ is competitive on $\ell_{-}$ with the best $c \in \mathcal{C}^+$ for the loss $\ell_{-}$. That is, $h$ is competitive with the best hypothesis which never has false negative predictions on the distribution.
Negatively reliable learning (NRL) is defined analogously.

Finally, the paper defines the class $\textrm{SC}(\mathcal{C})$ as the set of classifiers $c_?$, where each $c_?$ only predicts something other than ? on $x$ if there exist a pair of hypotheses $c_+ \in \mathcal{C}^+$ and $c_{-} \in \mathcal{C}^{-}$ agree. Intuitively, the class SC is filled with very cautious classifiers which will only predict not ? if there are two high confidence predictors which agree. The “fully” reliable learning framework (FRL) asks for a learner to output a $h_?$ which has low error on the sum of losses $\ell_+$ and $\ell_{-}$, and also is competitive w.r.t. The best $c_?$ in $\textrm{SC}(\mathcal{C})$.

Everything up to now was introduced in previous selective classification frameworks (Kalai et al. Kanade et al.). In section 2.4, the paper introduces a generalization of the Chow loss model (triplet of losses $(\ell_+, \ell_{-}, \ell_{?})$) which allows for weighing each sub-loss by constants $\lambda, \mu, \nu$ respectively.

Finally, we are ready to define selective omniprediction. We say that a predictor $h$ is a $(\mathcal{L}, \mathcal{C}, \epsilon)$-select omnipredictor if the expected generalized chow loss of $h$ composed with a postprocessing function $k*$  (which maps from $[0,1] \to \mathbb{R} \cup {?}$ ) is almost as good as the best $c_?$ coming from the class $k \circ \mathcal{C}$. This is exactly normal omniprediction, but we allow for
1. post-processing functions to map certain regions of predictions from $[0,1] \to ?$
2. The loss function class to be the generalized chow loss, which penalizes ? differently.

Note that $k^*$ is the optimal post-processing function for a specific loss triplet for the Bayes optimal predictor, similar to the optimal post-processing function from normal omniprediction).

In theorem 3.1, the paper claims that selective omniprediction is achievable. In fact, it follows from standard omniprediction with no adjustments except for the post-processing functions $k$ which allow for abstention. Interestingly, lemma 3.2 shows that the interval where k^* abstains is a contiguous subset of $[0,1]$.

In section 3.1, a nice experiment is run which demonstrates that multicalibration + selective omniprediction achieves comparable / better generalized Chow loss in practice relative to training a depth-3 decision tree for each loss individually. This provides empirical evidence for Theorem 3.1.

Next, in theorem 3.3, the paper utilizes selective omniprediction to generalize the existing results for PRL, NRL, and FRL beyond the 0-1 loss and to an entire family of losses. Of note is that many of the results up until now extend to some important families of non-convex losses, similar to recent work in normal omniprediction. In Lemma 3.4, the paper shows how the FRL allows the predictor to have guarantees on the fraction of the distribution it abstains on (via conformal prediction sets).

Next, the paper turns towards group-fairness notions based on selective classification. In particular, the paper asks whether we can provide accurate predictions overall and not penalize certain subgroups of the population $\mathcal{G}$ by abstaining more on them than necessary. Along this vein, the paper defines a $(\mathcal{C}, \mathcal{G})$-selective classifier as one which produces both good overall error, and also achieves optimal abstention rate when compared to the best in class abstaining predictor $c_? \in \textrm{SC}(\mathcal{C})$ for each subgroup $g \in \mathcal{G}$. In Theorem 4.1, the paper shows that global calibration + multiaccuracy w.r.t. The intersection class $\mathcal{C} \cap \mathcal{G}$ suffices to achieve this notion of group sensitive selective classification. In Lemma 4.2, the paper then shows that for finite $|\mathcal{G}|$, we don’t need a weak agnostic learner for the intersection of the concepts in order to achieve group sensitive selective classification.

Finally, in section 4.1, the paper shows that $(\mathcal{C}, \mathcal{G})$-selective classifiers also satisfy group-conditional conformal coverage guarantees which have recently become popular. These state that the resulting predictor has good conformal guarantees not only overall, but also when conditioned on subgroups of the population $\mathcal{G}$.

**Questions:**

1. Coverage as a concept does not seem to be defined before it is used in Table 1. Does coverage mean that the predictor is abstaining less and predicting more? On that note, why is the coverage so different for training DTs from scratch versus selective omniprediction? Also, are there fundamental limitations to high coverage for all losses?

2. Is there a typo in the definition of 0-1 loss where it is first introduced? It is currently defined as $|y - t|$, however, for $t \in \mathbb{R}$ (as stated in line 83), this is not the 0-1 loss, this is the $\ell_1$ loss. Throughout the paper, we are working with predictions from a concept class $\mathcal{C}$ which only contains hypotheses which map to ${0,1}$, so that’s fine. However, this isn’t true when you introduce the 0-1 loss in line 82/83, where you allow $t \in \mathbb{R}$.

3. I think more discussion on the class $\textrm{SC}(\mathcal{C})$ is warranted. Why is this class interesting or important to be competitive w.r.t.? How do selective classification and selective omniprediction relate to each other? I understand that the former compares to the best in class for $\text{SC}(\mathcal{C})$, whereas the latter for the best in class $k \circ \mathcal{C}$, but how should I think about the relative strengths of the results in Theorems 4.1 and 3.1? Are they in some sense incomparable because the baselines are different? However, the calibrated multiaccuracy requirement is also a weaker condition than full multicalibration, so I would assume 3.1 is stronger. Is it strictly stronger (a la multicalibration vs. calibrated multiaccuracy)?

4. Is $\mathcal{C} \cap \mathcal{G}$ the class of all intersections of concepts from $\mathcal{C}$ and $\mathcal{G}$? Isn’t this difficult to achieve multiaccuracy for in general? It seems so, given the fact that this requires a weak-agnostic learner for the class. I understand this is why Lemma 4.2 is proposed.

5. Do we have lower bounds for omniprediction, and do these extend to selective omniprediction?

**Ethical Concerns:**

["NO or VERY MINOR ethics concerns only"]

**Final Justification:**

The authors answered all my minor questions / clarifications, and will take them into consideration in the next draft of the paper. I believe that the paper is a clear accept since it advances the ominprediction and selective prediction literature in a very intuitive and clean manner.

**Limitations:**

The authors include some limitations implicitly in lines 342-345. It may be good to point out that a WAL for the intersection class can be difficult to obtain (in theory, not sure about in practice). Perhaps one could also highlight the higher sample complexity for full multicalibration (and hence, selective omniprediction) vs. calibrated multiaccuracy.

**Quality:**

4

**Strengths And Weaknesses:**

I believe that the paper is a clear accept. It extends both the selective prediction and omniprediction frameworks, showing that they combine in a relatively straightforward way. Learning to abstain has become a popular topic, especially in fairness, and combining utilizing the omniprediction guarantees seems to naturally present some nice technical solutions to this. In addition, I really enjoy that the paper has significant numerical experiments with reasonable classes of predictors (depth 3 decision trees, random forests etc.). The paper also has some nice connections between omniprediction and conformal prediction, which I have not ran into before (although I don't keep up with the omniprediction literature, and have read only the first Gopalan et al. paper). In addition, to the best of my knowledge there is plenty of potential for future work with selective omniprediction in, for example, online settings which have become quite popular in the omniprediction community.

Overall, the paper introduces the abstention framework to the omniprediction line of work, which may prove to be fruitful for future research in this area.

Minor Weaknesses:
1. The proofs, while numerous, seem relatively straightforward given the definitions proposed in this paper, the definition of omniprediction, and the definitions of PRL, FRL, NRL, etc. However, I think this is not a major weakness. In particular, I believe that this means that the selective prediction definitions in this paper were chosen correctly if things work out so nicely.
2. In some sense, there are many results in the paper (and further experiments demoted to the appendix). Given that omniprediction papers rarely have experiments, I would perhaps recommend giving informal theorem statements for some of the later results, and trying to include more of the experimental results in the main paper. For the average neurips audience, this could be especially interesting / useful.

---

> ### Author Rebuttal · Authors · 2025-07-31
>
> We thank the reviewer for their very thorough evaluation of our paper and for the excellent questions, which are all important points that we should expand on and clarify in the final version of our paper. We answer them in order:
>
> (1) Yes, coverage refers to the fraction of the points in the domain on which the predictor does not abstain (e.g., a non-selective classifier has full coverage equal to 1). We should have defined this term formally in the paper and will add a definition. Regarding the coverage of a selective omnipredictor vs that of decision trees, it is arguably intuitive to expect the selective omnipredictor to attain higher coverage. This is because the omniprediction framework provides a much stronger learning guarantee than that of usual loss minimization. Because the labels are boolean, the values of the selective omnipredictor are "polarized" towards 0 and 1 because it is a more powerful learning algorithm. Meanwhile, we are comparing it to a method with no theoretical guarantees, so it makes sense that decision trees attain lower coverage (because the predictions are more uncertain and thus closer to $½$).
>
> Regarding the question of whether there are limitations to high coverage for all losses, the main limitation for the coverage rate is that we are working in the agnostic setting, and so we are not making any assumptions about the data generation process. Indeed, the labels could be random noise. In this case, unless the cost of abstention is extremely high, one would expect very low coverage. More generally, the rate of coverage depends on the relative cost of misprediction vs of the cost of abstention, given that higher costs of abstention lead to higher coverage (and vice-versa; if it is much cheaper to abstain than to predict, then we should expect low coverage). A theoretical guarantee of our selective omnipredictor is that the abstention interval $I_{abs}$ is fixed for each generalized Chow loss function, independent of the true labels (as we show in Lemma 3.2). So, the coverage rate corresponds to the rate of datapoints that have $k_{\ell_{\pm}, \ell_{?}} \circ h$ value outside of the abstention interval $I_{abs}$ (where $h$ is a selective omnipredictor). Again, because we work in the agnostic setting, we cannot have a formal guarantee on how many such points there are, but perhaps if we had some assumptions on the data generation process then we would be able to argue about the number of points that fall within the $I_{abs}$ interval once the loss function has been fixed. This is an interesting direction for future work.
>
> (2) Yes, sorry, there is a typo in the definition of the 0-1 loss. We will change it and stick to the $\ell_1$ loss. Thank you!
>
> (3) These are all great points. First of all, regarding the definition of the class $SC(C)$: we remark that we take the definition directly from the reliable agnostic learning framework defined by Kalai, Kanade and Mansour (they call it $PC(C)$ in their paper). In their case, they are only considering boolean concepts $c \in C$. For a selective classifier, we need to compare the abstention rate to the base class $C$, but the issue is that $C$ only contains boolean concepts. So, we need to add abstentions to $C$. Note that because the concepts in $C$ are boolean, it doesn’t make sense to consider post-processings of the form $k \circ C$, as used in our selective omniprediction definition (where the concepts are real-valued). If we want to add abstentions ? to $c: X \rightarrow \{0,1\}$ without having to bring in other concept classes (i.e., we want to get it directly as a variation of the original class $C$), and given that the motivation of reliable agnostic learning is to ensure that whenever the classifier predicts, it essentially makes no errors, then the definition of $SC(C)$ is in fact very natural. We take the concepts that make no false positives, the concepts that make no false negatives, and ensemble them as per line 177. This ensures that we add abstentions to the class such that the classifiers in this class (i.e., $SC(C)$) make no errors whatsoever whenever they predict. We agree that we should explain this intuition behind the construction of $SC(C)$ (which we remark is not our original definition) and will include it in the paper.
>
> Regarding the difference between $SC(C)$ and $k \circ C$, it is important to note that for anything reliable agnostic learning related (i.e., for the original Kalai et al. paper and our Section 4), we are dealing with boolean concepts. Instead, for $k \circ C$ and our selective omnipredictors (i.e., Section 3), we are dealing with real-valued concepts. We will make this distinction clear in the paper. $SC(C)$ and $k \circ C$ are a priori incomparable, but we can relate them by expanding the boolean class $C$ as follows: for each $c_1, c_2 \in C$, we define the class $C’$ as containing the concepts $(c_1 + c_2)/2$ (note that $C’$ is no longer a boolean class). We define the post-processing $k \circ C$ as mapping 0 to 0, 1 to 1, and $½$ to “?”. Then, we have that $SC(C) \subseteq k \circ C’$.
>
> Lastly, regarding the relative strengths of Theorems 3.1 and 4.1, they are incomparable. The above paragraph explains how we can think of relating the base concept classes, but they are still different. The objects that the theorems are dealing with are also different: there is no class $G$ in the case of Theorem 3.1, only $C$. A $C$-selective omnipredictor is also not necessarily stronger than a $(C, G)$-selective classifier: there are no guarantees on its “local” abstention rate on each $g \in G$ (there is no class $G$ to begin with). The concepts also have different ranges, as we have discussed. We view Theorem 4.1 as the “multigroup” version of fully reliable learning (as originally introduced by Kalai et. al), which is also why we continue to use $SC(C)$ in that case, and because we similarly want to ensure that whenever we predict, we make no mistakes (as per Condition 1 in the definition of $(C,G)$-selective classification). As  discussed, this base class is not appropriate for the definition of a selective omnipredictor, and there we also instead consider the total Chow loss, which considers the total loss rather than asking for no misprediction loss (unlike in the reliable agnostic setting).
>
> It is a good question whether we could get selective omnipredictors from the weaker primitive of calibrated multiaccuracy rather than from multicalibration, especially in light of new constructions of omniprediction from calibrated multiaccuracy (see “Loss Minimization Through the Lens Of Outcome Indistinguishability” by Gopalan et al. and “Near-Optimal Algorithms for Omniprediction” by Okoroafor et al.). It is a question that we would like to explore further.
>
> (4) Yes, $C \cap G$ corresponds to the class ${cg \mid c \in C, g \in G}$. We will add the formal definition to the paper. We acknowledge that $C \cap G$ is a larger class than $C$ and $G$ separately, but agnostic learning is hard in theory for most concept classes anyway. So, it is not taking the intersection of the classes what makes the problem hard in theory. In practice, we have heuristics to simulate a weak agnostic learning oracle (e.g., using random forests or neural networks). It is still useful (and common) to solve a learning problem (in this case, constructing a $(C, G)$-selective classifier) by assuming oracle access to another learning problem (in this case, constructing a $(C \cap G)$-weak agnostic learner), even if we only have heuristics to simulate this oracle access in practice. The theoretical reduction still holds in practice.
>
> Separately, is interesting to ask whether we can solve the problem of $(C \cap G)$-weak agnostic learning by using only a $C$-weak agnostic learner and a $G$-weak agnostic learner (i.e., without having to require a weak agnostic learner for the class $C \cap G$). We do know whether this is possible, but it appears to be a standalone interesting learning-theoretic question.
>
> (5) As far as we know, there are no known lower bounds for omniprediction, although there has been a lot of recent work dedicated to improving the construction of omnipredictors (see, for example, the work “Near-Optimal Algorithms for Omniprediction” by Okoroafor et al.). It would be good to think more about the precise relationship between the problem of omniprediction vs the problem of selective omniprediction, both in terms of learning primitives and of computational efficiency. Now that we have introduced the notion of selective omniprediction in our work, this is a natural next direction. More broadly, it would be great to better understand the relationship between typical learning-theoretic problems and their selective counter-parts. For example, Kanade and Thaler showed that reliable agnostic learning is weaker than (weak) agnostic learning (under standard hardness assumptions). We can ask: is this a general phenomenon when adding abstentions to a "non-selective" learning problem?
>
> Lastly, we agree that it would be useful to make some of the statements more informal for some of the results and give some more experimental details for a NeurIPS audience, and will incorporate this in the final version of our paper. We will also compare the known sample complexity differences between calibrated multiaccuracy and multicalibration. Thank you very much!

---

> > ### Comment · Reviewer_6Q23 · 2025-08-04
> >
> > Thank you for the wonderful response! I will keep my score the same.

---

### Official Review · Reviewer_AQet · 2025-07-05

**Clarity:** 2
**Significance:** 3
**Originality:** 2
**Rating:** 5
**Confidence:** 4

**Summary:**

This work extends the literature on omniprediction and multigroup fairness to the reliable learning framework. They define a selective omnipredictor for a class of loss triples (where losses may differ on negatively and positively labeled examples, as well as examples on which the classifier abstains) and concepts C to be a predictor h such that for every loss in L, and any associated weights for these losses, there exists an efficient post-processing function such that post-processing the predictions of h yields predictions with loss that is competitive with the best (potentially abstaining) post-processing of a concept in C. They prove that multicalibration is sufficient for selective omniprediction, and show that selective omnipredictors will abstain in contiguous intervals that balance the cost of abstention vs classification. They then consider the question of abstaining fairly, so that no group suffers gratuitous abstention for the sake of reliability. They show that calibrated multiaccuracy with respect to the class $C \cap G$ is sufficient for fair abstention, and also observe that their fair selective classifier can be interpreted as providing group conditional coverage in the sense of conformal prediction.

**Questions:**

See comments above

**Ethical Concerns:**

["NO or VERY MINOR ethics concerns only"]

**Final Justification:**

I have a positive appraisal of the paper and I believe the updates proposed by the authors in the rebuttal will address my concerns regarding clarity

**Limitations:**

Yes

**Paper Formatting Concerns:**

No concerns.

**Quality:**

3

**Strengths And Weaknesses:**

This paper provides a very useful contribution to our understanding of multigroup fair and reliable classification. I don’t think there are many particularly novel techniques used to prove the results of this work, but I see its contribution more as giving the right definitions of models of fair abstention to make nice observations about how the properties of multicalibration and calibrated multiaccuracy facilitate fair, flexibly loss-minimizing, and reliable learners.

I don’t think $C \cap G$ was actually defined anywhere. It can be inferred via context and proofs, but would be good to define before first used in section 4.

I think in one of the proofs it is assumed that the constant function g(x) = 1 is in G, but that assumption doesn’t appear in the theorem statement.

---

> ### Author Rebuttal · Authors · 2025-07-31
>
> We thank the reviewer for their positive comments on our paper and for providing insightful feedback. We agree that we should have formally defined the class $C \cap G$ rather than directly use it in the proofs; we will add a formal definition in the beginning of Section 4. We also agree that we should explicitly say that we assume that the constant 0 and 1 functions are in both $C$ and $G$. We say so for $C$ in the beginning of Section 2, and for $G$ in lines 326-327, but we should be more explicit about this assumption. Thank you!

---

> > ### Comment · Reviewer_AQet · 2025-08-06
> >
> > Thank you to the authors for their response, I will maintain my score.

---

### Official Review · Reviewer_snFc · 2025-07-08

**Clarity:** 3
**Significance:** 3
**Originality:** 3
**Rating:** 5
**Confidence:** 2

**Summary:**

The paper proposes selective omniprediction, a framework with the scope of training once a single predictor for a class $\\mathcal{L}$ of convex Lipschitz losses by using data samples that can later be post-processed without additional data to compete with the best predictor in a given concept class $\\mathcal{C}$ with respect to a generalization of the Chow loss induced by a given loss function $\\ell \\in \\mathcal{L}$.
The main contribution of the selective omniprediction framework, which distinguishes it from plain omniprediciton, is that it allows classifiers to abstain whenever the risk of a misprediction can be too high.
It achieves this by first building a multicalibrated predictor $h$ and then applying a data-free post-processing $k\_{\\ell}$.
The resulting abstention region is always shown to be a single contiguous interval and to allow a desirable coverage guarantee, while the method remains computationally efficient.
The construction yields positive, negative, and fully reliable learners for losses in $\\mathcal{L}$, subsuming earlier $0$-$1$ reliable learning results and enabling a simple conformal prediction scheme with bounded abstentions.
Because the approach relies only on the agnostic learnability of the base class $C$, it remains computationally practical too.
Synthetic experiments confirm that a single multicalibrated omnipredictor, after task-specific post-processing, matches or surpasses other baseline models for each loss while abstaining exactly on the theoretically predicted interval, validating the approach in practice.

**Questions:**

- At line 91, the Bayes optimal predictor has a specific form. The general definition should be something like $f^*(x) = \\arg\\min\_{t \\in \\mathbb{R}}\\mathbb{E}[\\ell(y,t) | x]$. Could you clarify how your definition is equivalent to the general one?
- In the math display below line 100, shouldn't the conditioning on "$g(x) = 1$" be in the outer expectation?

**Ethical Concerns:**

["NO or VERY MINOR ethics concerns only"]

**Final Justification:**

I keep my already positive initial evaluation, also given that the authors replied to all my concerns and no sudden issue came up during the discussion.

**Limitations:**

Yes

**Quality:**

3

**Strengths And Weaknesses:**

The paper presents an interesting framework for selective omniprediction, which is well-motivated and theoretically sound.
This is obtained by extending the omniprediction framework to allow for abstention by considering a novel generalization of the Chow loss.
On top of that, the paper provides a clear and comprehensive overview of the theoretical results with multiple insights into the properties of the proposed method.
For instance, the notions of positive, negative, and fully reliable learnability are clearly described and adopted in order to perform conformal prediction, whose objective is to output prediction sets that provide a coverage guarantee for the true labels with respect to a given error rate $\\epsilon$.
The results are further extended to consider a notion of multigroup fairness, providing a way to construct a fair selective omnipredictor by relying on the notion of calibrated multiaccuracy.
The methods and concepts adopted could also be of interest in other related areas.

In addition to the positive theoretical results, the experiments are another strong point of this work.
Although the empirical study is limited to a single synthetic dataset and a concise set of baselines, the selective omnipredictor attains significantly higher coverage than loss-specific abstaining decision trees while also achieving a lower total loss.

On the other hand, the write-up presents some typos that could affect the overall understanding of the results, since the notions handled in the arguments are somewhat delicate.
These mistakes are generally minor and should be easily fixable.

**Minor comments and typos:**
- Line 68: "that true" should be "that the true"
- Line 82: it would be clearer to specify that the marginal distribution is over $\\mathcal{X}$
- Lines 83-84: the absolute loss is called $0$-$1$ loss by the authors, which is correct if $t \\in \\{0,1\\}$ but might be a misleading name when $t \\in [0,1]$ in general
- Line 85: "$=$" should be "$\\subseteq$"
- Line 86: "$\\ell_{\\mathcal{D}}$" should be "$\\ell_{\\mathcal{D}}(h)$"
- Line 91: it would be clearer to argue why the Bayes optimal predictor is defined as shown
- Line 117: "predictions" should be "predictions of"
- Line 147: "any value of" is a bit misleading, and it would be clearer to say something like "a constant"
- Line 168: "$c(x)$" should be "$c\_{+}(x)$"
- Lines 198, 225: please, provide a more precise reference to the appendix
- From line 227, the proof sketch of Theorem 3.1 has no clear end, so it would be better to put a q.e.d. mark when it is concluded
- Line 288: in "$\\mathcal{S}(x) \\subseteq \\mathcal{Y}$", $x$ should not be bold
- Below line 301, $c\_{\?}(x)$ should have a bold $x$
- Lines 314-319: it would be clearer to make the dependence of $c\_{\?}$ on $g$ explicit, e.g., using $c\_{\?}^{g}$
- Lines 325-327: just as a remark, it would suffice to assume that $\\mathcal{G}$ satisfies $\\bigcup\_{g \\in \\mathcal{G}} g = \\mathcal{X}$
- Lines 350-360: consider making this part a more explicit and structured remark, since it contains interesting observations
- Line 368: "$[0,1] \\cup \?$" should be "$[0,1] \\cup \\{\?\\}$"
- Line 369: $[1-\\epsilon,\\epsilon]$ should be $[1-\\epsilon,1]$
- Line 370: specify the inequality holds "for all $g \\in \\mathcal{G}$"
- Lines 538-565: the instructions block can be deleted

---

> ### Author Rebuttal · Authors · 2025-07-31
>
> We thank the reviewer for their insightful comments and questions. We answer them in order:
>
> (1) Thank you very much for pointing out these typos, we will make sure to correct all of them.
>
> (2) This is a great point about the definition of the Bayes optimal predictor. We agree that we misuse this term in the paper; what we should have said in line 91 (and will say in the final version of the paper) is that $f^{\ast}$ is the ground truth predictor $f^{\ast}(x) = E[y|x]$. This corresponds to the Bayes optimal predictor in the form written by the reviewer in the case of the squared loss. More generally, all proper losses ensure that $E[\ell(y, f(x))]$ is minimized by the ground-truth predictor. (For improper losses, we can post-process the ground-truth predictor $f^{\ast}$ to minimize the loss; see the paper “Near-Optimal Algorithms for Omniprediction” by Okoroafor et al. for details.) We will be careful about this distinction and stick to the name  “ground-truth predictor” in our paper.
>
> (3) Yes, that is correct about the conditioning on $g(x)=1$. We will correct it, thank you!

---

> > ### Comment · Reviewer_snFc · 2025-08-07
> >
> > Thank you for answering my questions. I keep my positive evaluation for this submission.

---

### Official Review · Reviewer_FrNz · 2025-07-11

**Clarity:** 3
**Significance:** 2
**Originality:** 3
**Rating:** 5
**Confidence:** 3

**Summary:**

This paper extends the problem of omniprediction to the selective classification setting. That is, the goal is to learn an omnipredictor that is trained for a class of loss functions and can easily be post-processed with respect to a specific loss function at test time, while being allowed to abstain from making a prediction. The authors show that a selective omnipredictor that achieves comparable loss to the best selective classifier can be constructed by first constructing a multicalibrated predictor and then applying the post-processing function, which is taken from the standard omnipredictor setting and extended to the triple of loss functions for selective classifications. They additionally show that a similar insight can be used to build selective classifiers with fair abstentions, where fairness is defined as the local abstention rate within each subgroup being comparable to that of an optimal selective classifier.

**Questions:**

1. How was the synthetic data for experiments generated?

2. What are some example scenarios in which selective omniprediction is necessary?

3. Can this framework be easily extended beyond binary classification settings?

Other minor comments/questions:
- It would be nice to get at least some high level insights in the main body of the paper about how the framework can be generalized beyond convex and Lipchitz loss functions. For space concerns, I think the very long preliminaries could be simplified a little bit. For instance, I would suggest focusing on FRL in the main paper rather than distinguishing PRL, NRL, and FRL.
- Multicalibration: why multiply by probability of group? Is this standard definition? Less calibrated ok for minority then
- Theorem 2.1: Should $\mathcal{C}$ be a set of functions $\mathcal{X} \to \{0,1\}$ instead, for compatibility with the notion of multicalibration?
- Line 214: what is the formal definition of “$(\mathcal{C}, \epsilon)$-computationally indistinguishable”?
- Section 3.4: what do $l_1$ and $l_2$ refer to?
- Table 1 shows that the proposed method achieves much higher coverage. Is this a general phenomenon implied by the theoretical framework?
- What are the empty regions (neither blue or red) in Figure 1?

**Ethical Concerns:**

["NO or VERY MINOR ethics concerns only"]

**Final Justification:**

The authors have addressed most of my questions, and I am raising my score to a 5.

**Limitations:**

Yes

**Paper Formatting Concerns:**

No major formatting issues.

**Quality:**

3

**Strengths And Weaknesses:**

The main contributions of this paper are theoretical, showing how to build selective omnipredictors for both convex and nonconvex losses, which guarantees on the performance that seem fairly strong. The results are also technically sound as far as I can tell.

Moreover, the authors also provide experiments on a synthetic dataset demonstrating that the proposed selective omnipredictors can achieve comparable or even better final loss than selective decision trees trained for each triple of loss functions, even though the loss within non-abstaining regions are much higher, thanks to higher coverage rates. However, the details of how the synthetic dataset was generated were not shared, and the results would be stronger with additional experiments on real-world benchmark datasets.

While the theoretical framework of selective omniprediction is strong, the significance of the setting combining omniprediction and selective classification could be better motivated, especially with potential real-world application scenarios.

I am less convinced about the proposed notion of fair abstentions. Often for tasks where fairness is a concern, the data itself is biased. For instance, the labels for certain minority groups may be less accurate or noisier. In this case, even though abstention rates are locally comparable to optimal selective classifiers within each subgroup, they may still be highly disproportionate among different subgroups. I think a more natural definition that aligns with existing fairness notions might be to additionally equalize (up to epsilon) the abstention rates across subgroups.

The authors defer a lot of details to the appendix, which can somewhat hurt readability. However, I understand that this is hard to avoid to some degree due to page limits.

---

> ### Author Rebuttal · Authors · 2025-07-31
>
> We thank the reviewer for their insightful questions. We answer them in the same order:
>
> (1) We provide the details of how we generate the synthetic data for our experiments in the appendix. All the details of the experiments for selective omniprediction can be found in Section F.1. Specifically, for these experiments, we generate 10,000 samples and 20 features as our data using sk-learn’s function "make_classification".
>
> The experiments for the conformal prediction application of FRL can be found in Section F.2. Specifically, for these experiments, we generate 5,000 data samples synthetically using sk-learn’s "make_blobs" function, which generates isotropic Gaussian blobs for clustering.
>
> Lastly, the experiments for (C, G)-selective classifiers can be found in Section H. Specifically, we use the same data generation process as in the selective omnipredictors experiments. Namely, we generate 10,000 samples synthetically using sk-learn’s "make_classification" function.
>
> (2) Selective omnipredictors, given their strong learning guarantees, allow us to train without having to commit to a loss function in advance. This is very useful in scenarios where we do not know the loss function at the time of training, or when we might want to change the loss function in the future. For example, if false positives become costlier over time, or if the cost of abstention changes over time. Having this type of flexibility is very useful in any prediction setting and in real-world deployments. While this flexibility is also present in the original notion of an omnipredictor (i.e., without abstentions), being able to abstain (optimally) is also a very useful outcome option to add to a  predictor. For example, in societal applications, this allows us to identify the points on which the prediction is uncertain and defer them to further review, rather than forcing a prediction. Our framework merges the line of work on omniprediction and the line of work on selective classification, thus merging the benefits of both frameworks. Our selective omnipredictors abstain optimally and do so in this very flexible “loss agnostic” sense. So, this is useful both because we have the loss function flexibility feature and because we give the predictor the option to abstain.
>
> (3) Yes, it can be extended beyond binary classification settings, and we will make a remark in the paper. Multiple follow-up works to the original papers on multicalibration and omniprediction have provided constructions for the real-valued labels setting. (See, for example, the works “Omnipredictors for regression and the approximate rank of convex functions“ by Gopalan et al. and “Multicalibration as Boosting for Regression” by Globus-Harris et al.). Because our algorithm for selective omniprediction relies on a multicalibration algorithm, known extensions of the multicalibration algorithm to the real-valued setting apply to us as well.
>
> (4) Yes, it is standard in all works on multicalibration to multiply by the probability of the subgroup. We need to do something about the very small subgroups; otherwise, the task of building a multicalibrated predictor becomes statistically impossible (indeed, otherwise one could consider the collection G including all singletons, in which case requiring multicalibration would not be computationally feasible). Different papers deal with the very small subgroups in slightly different (but equivalent) ways: we can multiply by the probability of the subgroup (for example, as done in the original omnipredictors paper), or we can condition on the group g, but then we must discard the level sets that have probability mass less than some chosen constant gamma, so that we speak of (epsilon, gamma)-multicalibration (for example, as done in the original multicalibration paper).
>
> We remark that there are two different types of “subgroups” going on here. Multicalibration is all about ensuring calibration on subgroups of the domain, and it does so for every group that includes a constant fraction of the population, and the guarantee does hold for all of these. What we are discarding are the subgroups that are smaller than that. But, as in many things in computer science, we just cannot make any promises for probabilities that are too small; there is not a way around not having enough statistical signal.
>
> Next we address the “Other minor comments/questions”:
>
> (1) Yes, we will add an explanation in the main body regarding how we can extend the construction of selective omnipredictors beyond convex and Lipschitz loss functions. The formal proof is provided in Section E in the appendix. Essentially, the idea is to only require Lipschitzness over a large enough closed interval, rather than on all of $\mathbb{R}$. This relaxation allows us to add new loss functions to the class $\mathcal{L}$. This relaxation is used in the original omnipredictors paper, where these type of functions are called nice loss functions. Recent work has expanded the construction of omnipredictors beyond convex losses (see “Loss Minimization Through the Lens Of Outcome Indistinguishability” by Gopalan et al. and “Near-Optimal Algorithms for Omniprediction” by Okoroafor et al.).
>
> (2) While the original definition of multicalibration and the way we state it in Section 2.1 is for boolean functions, the notion of multicalibration was quickly extended to concepts $c$ taking real values on a closed interval. This is why in both the original omnipredictors theorem (Theorem 2.1) and in our selective omnipredictors theorem (Theorem 3.1) we use real-valued concepts $c$. We will clarify this in the paper so that the theorem statements are coherent with the precise multicalibration definition that we use.
>
> (3) We will include a definition of computational indistinguishability. Essentially, we just wanted to give some intuition as to why multicalibration is a useful tool for omniprediction. As formalized in the so-called “Outcome Indistinguishability” framework by Dwork et al., multicalibration can be viewed as a form of indistinguishability, in the sense traditionally used in the field of complexity theory. A thorough explanation of the omniprediction framework from the lenses of computational indistinguishability can be found in the paper “Loss Minimization through the Lens of Outcome Indistinguishability” by Gopalan et al.
>
> (4) They refer to the $\ell_1(y, t) = |y-t|$ and $\ell_2(y, t) = (y-t)^2$ loss functions, respectively.
>
> (5) While we do not formally state bounds on the coverage of our method, one would expect a selective omnipredictor to attain higher coverage than a decision tree. The reason is that omniprediction is a very strong learning notion, which provides much more stringent guarantees than usual loss minimization. Because the labels are boolean, the values of the selective omnipredictor are "polarized" to 0 and 1 because it is a more powerful learning algorithm. Meanwhile, we are comparing it to a method with no theoretical guarantees, so it makes sense that decision trees attain lower coverage (because the predictions are more uncertain and thus closer to $½$). It would be interesting to formally investigate the coverage guarantees further.
>
> (6) These empty regions are because the figure comes from our experimental results, and so our selective omnipredictor doesn’t have all values $p \in [0,1]$ in its range. So in theory, the blue and red regions are entirely the designated color; experimentally, the empty regions correspond to points $p \in [0,1]$ that are not in the range of the post-processed selective omnipredictor for the particular data at hand.
>
> Thank you for all of the comments!

---

> > ### Comment · Reviewer_FrNz · 2025-08-07
> >
> > Thank you for your response. This addressed most of my concerns.
> >
> > My apologies, but I must have made a copy-paste mistake in Strengths And Weaknesses section while editing the review. I have corrected it now. One question that remains is justification for the proposed notion of fair abstentions. While this is not a major weakness as I think the contributions to selective omniprediction are interesting and significant, and I understand that the discussion period is closing soon, but I would appreciate if the authors could comment on that.

---

> > > ### Author Response · Authors · 2025-08-08
> > > **Addressing the new paragraph of "Strength and Weaknesses" -- Part I**
> > >
> > > Thank you very much for the response! No problem, we now see the new “Strengths And Weaknesses” paragraph (it didn’t appear to us before, which is why we weren’t able to address it in our first response). We address these new points below:
> > >
> > > (1) As mentioned in our first response, we do provide the details of the synthetic dataset generation for our various experiments in the appendix. Specifically, they can be found in Sections F.1, F.2, and H. We agree that it would be good to try our experiments on real-world datasets; however, as the focus of our work is theoretical, the empirical evaluation is meant to demonstrate that our algorithms are easy to implement and achieve the desired outcomes.
> > >
> > > (2) The significance of our setting combining omniprediction and selective classification is that we can combine their respective benefits. On the omniprediction side, selective omnipredictors allow us to train our predictor “agnostic” to the specific loss function. This allows us to change the loss function at any time a posteriori after training without having to re-train from scratch, which is a very useful property to have in practice. For example, in a medical setting, we might want to change the cost of false positives over time, or we might want to change the cost of abstention over time depending on, e.g., how many resources are available to further review the patients that get a prediction of “?”.
> > >
> > > On the selective side, selective omnipredictors allow us to add abstentions to the range of the predictor, which also brings about many benefits in practice. We add abstentions so that we reduce the prediction error: we only want our predictor to predict if it’s certain that it is correct. In high-stakes scenarios, abstention is a very useful and important option for the predictor to have; otherwise, we might be forcing it to make errors that could have been prevented had it had the option to defer. Selective omnipredictors combine these two benefits: we have the flexibility to choose any loss from the loss class at any point after training (including abstention losses) and achieve optimal generalized Chow loss (in the agnostic sense), and our predictor has learnt how to abstain optimally and thus has the benefits of a predictor that can defer to predict (more so: it abstains optimally for *any* chosen Generalized Chow loss from the class of losses).

---

> > > > ### Author Response · Authors · 2025-08-08
> > > > **Addressing the new paragraph of "Strength and Weaknesses" -- Part II**
> > > >
> > > > (3) This is a great question regarding the bias-ness of the data, which often gets brought up in the context of multigroup fairness notions. The key point to keep in mind for the framework of multigroup fairness (which includes all of the papers on multiaccuracy, multicalibration, omniprediction, etc., and our paper) is that we are operating in the setting of “fairness as accuracy”. That is, we assume that there is a ground truth predictor $f^{\star}(x) = E[y|x]$, and ideally this is the predictor that we would like to build. We also assume that we are getting samples $(x, y)$, where $y$ is labeled according to the ground-truth predictor. So, we assume that there is a “correct” ground-truth answer, and that we are getting labeled examples “correctly” (i.e., without biases). The “multi” framework focuses on the following observation: within this “fairness as accuracy” point of view (where there is no bias in the data), we can still build a predictor that is globally good, but very mis-accurate/mis-calibrated on some subgroups of the domain, if we only optimize for local accuracy/calibration when training. That is, this framework is trying to correct forms of “training negligence”, where we have forgotten about “local” guarantees and only focused on “global” measures instead.
> > > >
> > > > Precisely the multigroup fairness framework was introduced as an *alternative* to the typical group fairness notions (such as the “equalizing” type measures that the reviewer is pointing out), which are not the correct goal if we are operating within this “fairness as accuracy” framework. Moreover, these traditional group fairness notions typically can only be defined on **non-intersecting** groups, whereas the multigroup fairness framework (and our notion of a fair predictor) works for a collection $G$ of groups that can be arbitrarily large and **can intersect**. This is one of the key strengths of the multigroup fairness framework. If we get correct samples from $f^{\star}$, the correct thing to do is to try to abstain compared to the best selective classifier within each group, rather than to arbitrarily equalize the abstention rates among subgroups.
> > > >
> > > > That is, one can view multigroup fairness definitions as requiring more robust versions of the typical measures, by inspecting our predictor on many subgroups of the domain. In our definition of $(C, G)$-selective classification, we mimic precisely this idea: the original notion of fully reliable learning (FRL) by Kalai, Kanade, and Mansour only required the predictor to make no errors and to abstain no more than the optimal selective classifier **globally**. But if we don’t make this notion more stringent and inspect it **locally** as well, then we might be getting good error at the expense of unnecessarily over-abstaining on some subgroups. This is why our definition requires our predictor to abstain optimally within *every* subgroup in G (i.e., locally).
> > > >
> > > > In conclusion, while we acknowledge that data bias is a problem, the multigroup fairness framework, as well as our work (which precisely builds on this framework) assumes that the samples $(x, y)$ that we get for learning are correctly generated according to the ground truth predictor $f^{\star}$. This is also what is traditionally the model in agnostic learning and in the usual learning-theoretic models, where we assume that we get labeled examples from the example oracle. What our work (and the multigroup fairness framework) does is require more stringent measures of error and abstentions by requiring them to hold *locally* on every group of a rich and possibly-intersecting collection of groups. Once we know how to build good predictors from good data, we can further add the problem of biased data, and this is an interesting direction for future work. Perhaps “fairness” can be a misleading word that makes the reader think about the problem of data bias: what we are really interested in is in making the FRL notion more robust by inspecting it locally and requiring it to hold within each $g \in G$ (for possibly intersecting $g$'s), which is precisely what we are doing with our notion of $(C, G)$-selective classification.
> > > >
> > > > (4) We will try to add more details to aid with the readability of our paper in the main body, space permitting.
> > > >
> > > > Thank you very much for the questions!

---

### Decision · Program_Chairs · 2025-09-17

**Decision:**

Accept (spotlight)

**Comment:**

The paper studies omniprediction in the setting of selective classification, i.e., where it can abstain from making a prediction. The authors present a post-processing framework that allows a single classifier to optimally learn abstentions and predictions for any pre-specified class of loss functions. They then turn their sights on “fair abstention” by extending multicalibration algorithms to consider abstention. Their main result in this setting is an efficient construction of a (fair) selective classifier from a multiaccurate and calibrated predictor.

Strengths:
- The paper is well-motivated and overall well-structured (but see weakness below).
- The reviewers agree that both the theoretical and empirical results are strong and make a meaningful contribution to the field of omniprediction and selective prediction.

Weaknesses:
- The reviewers identified quite a few typos and areas where exposition was lacking; the authors promised to fix these in the camera-ready version of the paper and reviewers were by and large convinced.